# Riemannian Manifold Learning for Stackelberg Games with Neural Flow Representations

## Abstract

We present a novel framework for online learning in Stackelberg general-sum games, where two agents, the leader and follower, engage in sequential turn-based interactions. At the core of this approach is a learned diffeomorphism that maps the joint action space to a smooth Riemannian manifold, referred to as the *Stackelberg manifold*. This mapping, facilitated by neural normalizing flows, ensures the formation of tractable isoplanar subspaces, enabling efficient techniques for online learning. By assuming linearity between the agents' reward functions on the *Stackelberg manifold*, our construct allows the application of standard bandit algorithms. We then provide a rigorous theoretical basis for regret minimization on convex manifolds and establish finite-time bounds on simple regret for learning Stackelberg equilibria. This integration of manifold learning into game theory uncovers a previously unrecognized potential for neural normalizing flows as an effective tool for multi-agent learning. We present empirical results demonstrating the effectiveness of our approach compared to standard baselines, with applications spanning domains such as cybersecurity and economic supply chain optimization.

## 1 Introduction

A Stackelberg game consists of a sequential decision-making process involving two agents, a leader and a follower. This framework, introduced in Stackelberg 1934 models hierarchical strategic interactions where the leader moves first, anticipating the follower's best response, and then the follower reacts accordingly. These games have become central to understanding interactions in various fields, from economics to societal security, providing a formal method for analyzing situations where one party commits to a strategy before the other, affecting the subsequent decision-making process and reward outcomes. Over time, Stackelberg games have evolved to address more complex environments, incorporating factors like imperfect information and no-regret learning of system parameters. The solution to such a game typically revolves around finding a Stackelberg equilibrium, where the leader optimizes his strategy assuming or knowing the follower type, which affects how she optimizes her utility based on the leader's action. (Kar et al. 2015; Korzhyk, Conitzer, and Parr 2010).

Several challenges arise in the practical applications of Stackelberg games. One key issue is the uncertainty regarding the follower's type or rationality (or sub-rationality). In many real-world scenarios, the follower might not be fully rational or the leader might have incomplete knowledge of the follower's preferences, leading to uncertainty in the leader's decision-making process. Additionally, imperfect information regarding reward outcomes adds another layer of complexity, as the leader may not have accurate knowledge of the payoffs associated with various strategies. These uncertainties have been addressed in domains such as security, where randomized strategies and robust optimization approaches are deployed to mitigate risks arising from incomplete information and unpredictable follower behaviour (Jiang et al. 2013; Kar et al. 2015; Debarun Kar et al. 2017). For instance, in deployed systems like ARMOR at LAX or PROTECT at U.S. ports, leaders must make security decisions under uncertainty, balancing multiple risks (Jain et al. 2011; Shieh et al. 2012). Stackelberg games also feature prominently in supply chain optimization settings, where there exist areas of uncertainty, such as demand manifestation (L. Liu and Rong 2024) (Cesa-Bianchi et al. 2023). Stackelberg game models have also found applications in novel areas like conversational agents using large language models (LLMs), where one agent (the model) anticipates the user's behaviour and adjusts its responses accordingly (Nguyen et al. 2014).

For non-cooperative multi-agent games that exhibit additive noise, sublinear regret can be achieved via gradient based optimization methods, such as AdaGrad (Duchi, Hazan, and Singer 2011), in the face of Gaussian noise but this is often subject to constraints on the magnitude of the noise (Hsieh et al. 2023). Nevertheless, in these games, the problem settings are extended to an unlimited number of players - with regret performance degrading as the number of players increases. We investigate the problem setting of a two-player Stackelberg game, with a tractable best response function - commonplace in economics and adversarial machine learning in general (Wang et al. 2024; Zhou and Kantarcioglu 2016).

**Problem Setting:** We consider a two-player Stackelberg game where player A leads and player B responds. Stackelberg games are sequential, meaning that the players take turns and, the follower can best *respond* to the leader's action, given information available to him. The best response of player B lies on a manifold within a subspace of the joint action space $\mathcal{A} \times \mathcal{B}$. We define this Stackelberg game setting in the framework of optimal transport, where the structure of the best response function $\mathfrak{B}(\cdot)$ gleans simplifications to the solution methodology to obtain Stackelberg regret. This research focuses on applying multi-armed bandit (MAB) methods, particularly in Stackelberg equilibrium settings, to achieve sublinear regret. It explores the game theory, utilizing geometric topologies to better understand agent behaviour and simplify computations.

**Key Contributions:** We introduce a novel algorithm that significantly advances the understanding of Stackelberg learning under imperfect information, akin to the problem settings covered in Balcan et al. 2015 and Haghtalab et al. 2022, presenting a systematic framework for how equilibrium can be efficiently solved in this problem setting. Central to our contribution is the construction of a feature map using neural normalizing flows, which transforms the ambient joint action space into a more tractable embedding, we define as the *Stackelberg manifold*. By leveraging the geodesic properties of this manifold, our approach allows for more efficient computation of Stackelberg equilibria with respect to no-regret learning, particularly in the presence of parameter uncertainty. In addition to this, we offer a rigorous theoretical foundation for optimizing Stackelberg games on spherical manifolds. This framework is validated via empirical simulations, stemming from applications in supply chain management and cybersecurity, demonstrating that our method outperforms standard baselines, offering improvements in both computational efficiency and regret minimization.

## 2 FORMAL DEFINITIONS

In a Stackelberg game, two players take turns executing their actions. Player A is the leader, she acts first with action $\mathbf{a}$ selected from her action space $\mathcal{A}$. Player B is the follower, he acts second with action $\mathbf{b} \in \mathcal{B}$. The follower acts in response to the leader's action, and both players earn a joint payoff as function of their actions.

### 2.1 REPEATED STACKELBERG GAMES

In a repeated Stackelberg game, the leader chooses actions $\mathbf{a}^t \in \mathcal{A}$, and the follower reacts with actions $\mathbf{b}^t \in \mathcal{B}$ at each round $t = 1, 2, \ldots, T$. The leader's strategy $\pi_A(\cdot|\mathcal{H}_t)$ is a probability distribution over the action space $\mathcal{A}$ which selects $\mathbf{a}^t$ based on past joint actions up to time $t$, i.e., $\mathcal{H}_t := \{(\mathbf{a}^\tau, \mathbf{b}^\tau)|\tau < t\}$. Similarly, the follower's strategy $\pi_B(\cdot|\mathcal{H}_t)$ is a conditional probability distribution over $\mathcal{B}$ which determines $\mathbf{b}^t$ given the full history, i.e., $\mathcal{H}_t := \mathcal{H}_t \cup \{\mathbf{a}^t\}$.

**Best Response Strategy of the Follower:** To be specific, the follower selects his best response strategy at round $t$ by maximizing his expected reward function $\mu_B(\mathbf{a}, \mathbf{b}) : \mathcal{A} \times \mathcal{B} \to \mathbb{R}$ given that the leader has played action $\mathbf{a}^t$. Since, we assume that the reward function solely depends on the most recent pairs of actions, the follower's best strategy is first order Markov, i.e., $\pi_B(\cdot|\mathcal{H}_t) = \pi_B(\cdot|\mathbf{a}^t)$. Formally, the follower's best response at round $t$ is given by,

$$\pi_B^*(\mathbf{b}|\mathbf{a}^t) := \operatorname*{argmax}_{\pi_B \in \Pi_B} \mathbb{E}_{\pi_B}[\mu_B(\mathbf{a}, \mathbf{b})|\mathbf{a} = \mathbf{a}^t], \quad (2.1) \qquad \mathfrak{B}(\mathbf{a}^t) := \{\mathbf{b} \in \mathcal{B}|\pi_B^*(\mathbf{b}|\mathbf{a}^t) > 0\}. \quad (2.2)$$

where $\Pi_B$ is the space of probability distributions over the action space $\mathcal{B}$ and the expectation is taken with respecter to the strategy of the follower. In this case, we can define the set of follower's best responses in Eq. (2.2). Analogously, the leader aims at maximizing the expected utility $\mu_A(\mathbf{a}^t, \mathbf{b}^t) : \mathcal{A} \times \mathcal{B} \to \mathbb{R}$ that is a deterministic function solely driven by her action $\mathbf{a}^t$ followed by the reaction of the follower $\mathbf{b}^t$.

**Stackelberg Equilibrium:** Consider a follower whose best response is optimal. We denote this scenario as Stackelberg Oracle (SOC) learning. From the leader's perspective, the uncertainty is not

necessarily over the system, but rather the strategy of the follower $\pi_B(\cdot)$. *Stackelberg equilibrium* $(\pi_A^*, \pi_B^*)$ is achieved when the follower is best responding, according to Eq. (2.2), and the leader acts with an optimal policy given the best response of the follower,

$$\pi_A^* := \arg\max_{\pi_A \in \Pi_A} \mathbb{E}_{\pi_A, \pi_B^*}[\mu_A], \quad (2.3) \qquad \mathbb{E}_{\pi_A, \pi_B^*}[\mu_A] = \int_{\mathcal{A}} \pi_A(\mathbf{a}) \int_{\mathcal{B}} \mu_A(\mathbf{a}, \mathbf{b}) \pi_B(\mathbf{b}|\mathbf{a}) \, d\mathbf{b} \, d\mathbf{a}.$$
$$(2.4)$$

## 2.2 The Stackelberg Manifold

To address the complexity of solving for Stackelberg equilibrium under uncertainty, we propose the idea of mapping actions from the ambient space onto a manifold $\mathbf{\Phi}$ leading to several key advantages. Simplifying the problem by mapping to a geometric structure, such as a unit sphere, allows for significantly faster numerical computation while optimizing directly on an intuitive intrinsic geometry, reducing redundancies and provides ease with respect to enforcing constraints. Additionally, smoothness on such a structure enables computational advantages through methods like Riemannian gradient descent (Bonnabel 2013), which exploits differentiability for efficient optimization.

This concept of mapping the data from the ambient space, in our case defined by the joint action space $\mathcal{A} \times \mathcal{B}$, onto a latent space $\mathbf{\Phi}$ has been explored in several prior works. For a well defined manifold, typically the approach is to learn a diffeomorphism between the ambient data space, and the objective manifold, which is a subspace of the ambient data space (D. J. Rezende, Papamakarios, et al. 2020) (Gemici, D. Rezende, and Mohamed 2016). Suppose the manifold is not given, or there lies flexibility in defining the structure of such a manifold, the certain manifold learning techniques could be devised (Brehmer and Cranmer 2020). These approaches typically define invertible, or pesudo-invertible, probability density maps between the ambient data space, the latent space, and the manifold space.

### 2.2.1 Normalizing Flows for Joint Action Space Projection

We leverage *normalizing flows* to map a joint action space $\mathcal{A} \times \mathcal{B} \subset \mathbb{R}^D$ onto a manifold, $\mathbf{\Phi}$ embedded in $\mathbb{R}^D$ (Dinh, Sohl-Dickstein, and Bengio 2016; Papamakarios et al. 2021; D. J. Rezende and Mohamed 2015). Normalizing flows are a class of generative models that transform a high dimensional simple distribution (i.e., isotropic Gaussian) into a complex one through a series of invertible bijective mappings using neural networks that are computationally tractable. The joint action space consists of actions taken by two agents, denoted as $a \in \mathcal{A}$ and $b \in \mathcal{B}$, modelled via normalizing flows to ensure bijectivity and a tractable density estimate. Let $x \in \mathcal{A} \times \mathcal{B}$, the model density $p_X(x)$ for a data point $x \in \mathbb{R}^D$ is given by,

$$p_X(x) = p_Z(f(x)) \left| \det\left(\frac{\partial f(x)}{\partial x}\right) \right|. \tag{2.5}$$

Here $Z$ represents the latent space with a simple distribution, and $|\det(\partial f(x)/\partial x)|$ is the Jacobian determinant of the transformation $f : \mathbb{R}^D \to \mathbb{R}^D$. Several open-source methodologies and codebases have been developed to address this manifold mapping problem via normalizing flows (Brehmer and Cranmer 2020). We extend the `nflows` package from Durkan et al. 2020 into our approach. The key contribution of our application is the isolation of the input heads into two separate sections, before concatenating the inputs and feeding it through the normalizing flow. This allows us to control the subspace induced by the leader's action $\mathbf{a} \in \mathcal{A}$. (We provide detailed model specifications in Appendix D.)

### 2.2.2 Specifications of the Feature Map $\phi(\mathbf{a}, \mathbf{b})$

**Feature Map $\phi(\cdot)$:** We propose a function $\phi$, which is a *feature map* (Amani, Alizadeh, and Thrampoulidis 2019; Moradipari et al. 2022; Zanette et al. 2021). Let $|\mathcal{A}|$ and $|\mathcal{B}|$ denote the finite dimension of the action space of the leader and follower respectively, the feature map $\phi : \mathcal{A} \times \mathcal{B} \mapsto \mathbb{R}^D$, which effectively maps any $A$ by $B$ combination of vectors to a $D$ dimensional feature representation.

| Definition | Expression | |
|---|---|---|
| **(D1)** $\boldsymbol{\Phi}$ is measurable and reachable w.r.t. a $\sigma$-algebra over $\mathcal{A} \times \mathcal{B}$ (denoted as $\mathfrak{E}_{\mathcal{A} \times \mathcal{B}}$). | $\boldsymbol{\Phi} \subseteq \mathcal{A} \times \mathcal{B} \subseteq \mathbb{R}^D, \quad \boldsymbol{\Phi} \in \mathfrak{E}_{\mathcal{A} \times \mathcal{B}}.$ | (2.7) |
| **(D2)** $\boldsymbol{\Phi}$ is compact and closed. | See Appendix A.1 for detailed definition. | (2.8) |
| **(D3)** $\boldsymbol{\Phi}$ is Lipschitz in the joint $\mathcal{A} \times \mathcal{B}$. | $\left\| \|\nabla_{\mathbf{a}}\phi\|_p + \|\nabla_{\mathbf{b}}\phi\|_p - C \right\| \leq L_c$ | (2.9) |
| **(D4)** $\boldsymbol{\Phi}$ variational sensitivity in $\mathcal{A} \times \mathcal{B}$, with high probability. | $\|\mathbf{a} - \mathbf{a}'\| \leq \epsilon \implies \|\phi(\mathbf{a}', \mathbf{b}) - \phi(\mathbf{a}, \mathbf{b})\| \leq \delta, \forall \mathbf{b}$ | (2.10) |
| | $\|\mathbf{b} - \mathbf{b}'\| \leq \epsilon \implies \|\phi(\mathbf{a}, \mathbf{b}') - \phi(\mathbf{a}, \mathbf{b})\| \leq \delta, \forall \mathbf{a}$ | (2.11) |
| **(D5)** $\boldsymbol{\Phi}$ forms a smooth Riemannian manifold. | See Appendix A.4 for detailed definition. | (2.12) |
| **(D6)** $\boldsymbol{\Phi}$ has an approximate pullback. There exists $\phi^{-1}(\cdot) : \boldsymbol{\Phi} \mapsto \mathcal{A} \times \mathcal{B}$ such that, | $\left\| \phi^{-1}(\phi(\mathbf{a}, \mathbf{b})) - (\mathbf{a}, \mathbf{b}) \right\| \leq \epsilon, \, \forall \mathbf{a}, \mathbf{b}$ | (2.13) |

Table 1: Key assumptions of the Stackelberg Embedding $\boldsymbol{\Phi}$.

Further, we introduce a concept known as the *Stackelberg embedding*, denoted by $\boldsymbol{\Phi}$, which is defined as the image of $\phi$ over the joint action space domain $\mathcal{A} \times \mathcal{B}$,

$$\boldsymbol{\Phi} := Im(\phi) = \{\phi(\mathbf{a}, \mathbf{b}) | \mathbf{a} \in \mathcal{A}, \mathbf{b} \in \mathcal{B}\}. \tag{2.6}$$

The construction of $\phi : \mathcal{A} \times \mathcal{B} \mapsto \mathbb{R}^D$ can be via any means, in our case a normalizing neural flow network (but possibly any other architecture), but should abide by the imposed assumptions in Table 1. To be precise, $\hat{\phi}$ should denote our best learned representation of the *ideal* map $\phi$. Provided that we only have access to $\hat{\phi}$, purely for notational convenience, we will use $\phi$ to represent $\hat{\phi}$ moving forward.

**Definition 2.1.** *Bipartite Spherical Map* $\phi(\mathbf{a}, \mathbf{b})$*: Let $\mathbf{a} \in \mathcal{A}$ and $\mathbf{b} \in \mathcal{B}$, and define a mapping $\phi : \mathcal{A} \times \mathcal{B} \to \mathcal{S}^{(D-1)}$ from Cartesian coordinates to spherical coordinates on the $D$-dimensional unit sphere $\mathcal{S}^{(D-1)}$. The spherical coordinates are partitioned such that, $\mathbf{a}$ parametrizes a subset of the spherical coordinates, and $\mathbf{b}$ parametrizes the remaining coordinates $\nu_{\mathbf{b}}(\mathbf{b})$. Also, $\nu_{\mathbf{a}} \cap \nu_{\mathbf{b}} = \emptyset$, meaning the partitions are disjoint. Thus, the full mapping is given by:*

$$\phi(\mathbf{a}, \mathbf{b}) := (\nu_{\mathbf{a}}(\mathbf{a}), \nu_{\mathbf{b}}(\mathbf{b}))^{\intercal} \in \mathcal{S}^{(D-1)},$$

*where $\nu_{\mathbf{a}}$ and $\nu_{\mathbf{b}}$ represent distinct angular components of the spherical coordinates.*

**Mapping to a Spherical Manifold:** The transformation from spherical coordinates to Cartesian coordinates is used to map input features onto an $D$-dimensional spherical manifold. Therefore, in addition to the properties of our feature map $\phi$ from Table 1, we also enforce $\phi$ as a bipartite spherical map from Def. 2.1. This bipartite spherical map which constructs a disjoint spherical mapping to parameterize two subspaces in $\boldsymbol{\Phi}$. To accomplish this, we define two heads in the neural network input, the head from A specifically controls the azimuthal spherical coordinate and the head from B specifically controls other coordinates. (A visualization of the results, showcasing the learned bipartite mapping to $\boldsymbol{\Phi}$ as a 3D spherical surface, is provided in Appendix E.1. This visualization is generated by varying $\mathbf{a}$ or $\mathbf{b}$ to create longitudinal or latitudinal subspaces.)

Constructing a sufficient map to $\boldsymbol{\Phi}$ involves specifying the architecture and training model parameters such that it satisfies dynamics (**D1**) to (**D6**) as much as possible. This fundamentally requires a trade-off between being well behaved on the manifold, as stipulated by (**D3**) and (**D4**), and having an accurate inverse (**D6**). Thus, we train a neural network to approximate $\phi$, via $\phi$, with the loss

function,

$$\mathcal{L}(\phi) = \alpha_N \mathcal{L}_\phi^N + \alpha_R \mathcal{L}_\phi^R + \alpha_P \underbrace{\mathrm{Var}\Big(\phi(\mathbf{a}, \mathbf{b}) - \phi(\mathfrak{J}_\sigma(\mathbf{a}, \mathbf{b}))\Big)}_{\text{Perturbation Loss: } \mathcal{L}_\phi^P} + \alpha_L \underbrace{\Big|\|\nabla_\mathbf{a}\phi\| + \|\nabla_\mathbf{b}\phi\| - C\Big|}_{\text{Lipschitz Loss: } \mathcal{L}_\phi^L}.$$

$$(2.14)$$

The total loss $\mathcal{L}(\phi)$ is composed of multiple loss functions added together in a linear convex combination. $\mathcal{L}_\phi^N$ represents the negative log-likelihood loss of the normalizing flow $\phi(\cdot)$. To construct the Stackelberg manifold $\Phi$, data is first sampled uniformly the ambient Cartesian space. We then, fit a normalizing flow to $\Phi$ based on the criteria in Table 1, minizing $\mathcal{L}(\phi)$.

**Loss Function Descriptions:** $\mathcal{L}_\phi^N$ ensures transformed data matches the base distribution while adjusting for volume changes from invertible transformations, with respect to Eq. (2.5). Minimizing $\mathcal{L}_\phi^N$ allows the model to efficiently map complex data bijectively to simpler distributions. (A detailed description of $\mathcal{L}_\phi^N$ can be found in Appendix A.7.) $\mathcal{L}_\phi^R$ represents the geodesic repulsion loss of the output, which penalizes the concentration of elements being pairwise close to one another. (A detailed description of $\mathcal{L}_\phi^R$ can be found in Appendix A.6.) $\mathfrak{J}_\sigma(\mathbf{a}, \mathbf{b}) : \mathcal{A} \times \mathcal{B} \mapsto \mathcal{A} \times \mathcal{B}$ is a Gaussian perturbation function on the Cartesian product of the joint action space $\mathcal{A} \times \mathcal{B}$ to itself, subject to standard deviation $\sigma$. (A formal definition is provided in Appendix A.5.) The variance of the difference between $\phi(\mathbf{a}, \mathbf{b})$ and the perturbed $\phi(\mathfrak{J}_\sigma(\mathbf{a}, \mathbf{b}))$ should be kept minimal. This variance is captured over all elements in $\Phi$. The Lipschitz loss penalizes drastic deviations in the gradient with respect to $\mathbf{a}$ and $\mathbf{b}$, provided that the sum of the absolute values of the gradients does not deviate too far from some target $C \in \mathbb{R}$. The aforementioned losses in Eq. (2.14) are linearly combined in a convex combination to form the total loss $\mathcal{L}(\phi)$, denoted as $\alpha_N, \alpha_R, \alpha_P$, and $\alpha_L$ respectively. The hyperparameters were optimized via a selection process, leveraging empirical validation to identify the settings that maximized performance. Experimental hyperparameters and architecture of the normalizing neural flow network can be found in Appendix D.

## 2.3 Reward Function

**Reward Mechanisms:** A Stackelberg game provides two reward functions $\mu_A(\mathbf{a}, \mathbf{b})$ and $\mu_B(\mathbf{a}, \mathbf{b})$. Both of which are linearizable with sub-Gaussian noises, $\epsilon_A$ and $\epsilon_B$, i.e.,

$$\mu_A(\mathbf{a}, \mathbf{b}) = \langle \theta_A^*, \phi(\mathbf{a}, \mathbf{b}) \rangle + \epsilon_A, \quad (2.15) \qquad \mu_B(\mathbf{a}, \mathbf{b}) = \langle \theta_B^*, \phi(\mathbf{a}, \mathbf{b}) \rangle + \epsilon_B. \quad (2.16)$$

We assume zero-mean sub-Gaussian distribution for both $\epsilon_A$ and $\epsilon_B$ but they do not necessarily need to be identical. The objective is to learn the parameters $\theta_A^* \in \mathbb{R}^D$, and possibly as an extension problem $\theta_B^*$. The feature map $\phi(\cdot)$ maps the joint action space $\mathcal{A} \times \mathcal{B}$, to a subspace in $\mathbb{R}^D$. The parameters of the model, can be estimated via parameterized regression,

$$\hat{\theta}_t = (\phi_{1:t}\phi_{1:t}^\top + \lambda_{\text{reg}} I)^{-1} \phi_{1:t}^\top \mu_{1:t}, \qquad \text{for A and B, respectively,} \qquad (2.17)$$

Where $\phi_{1:t}$ represents the sequence of $\phi(\cdot)$ values via the feature map given the action sequences $\mathbf{a}_{1:t}$ and $\mathbf{b}_{1:t}$, $\lambda_{\text{reg}}$ serves as a regularization parameter, $I$ is the identity matrix, and $\mu_{1:t}$ are the historical rewards of players A or B (depending on the subscript). Here, we extend the reward structure of classical linear bandits in (Abbasi-Yadkori, Pál, and Szepesvári 2011; Chu et al. 2011) to a setting where two players jointly decide on the action sequence. We stipulate assumptions to ensure that the covariance matrix $\Sigma_T^{-1}$ is well-conditioned and positive semi-definite (PSD), with a regularization parameter $\lambda_{\text{reg}}$ balancing bias and variance, while the norm $||\phi(\mathbf{a}^t, \mathbf{b}^t)||_{\Sigma_T^{-1}}$ must remain small to facilitate efficient uncertainty reduction. (These assumptions are outlined in detail in Appendix A.2.)

## 3 Optimization of Stackelberg Games

**Optimization under Perfect Information:** We see that regardless of the convexity of $\mathcal{A}$ or $\mathcal{B}$, so long as we are dealing with compact spaces, under perfect information, we can solve the Stackelberg equilibrium by solving a bilevel optimization problem expressed as,

$$\pi_A^* = \arg\max_{\pi_A \in \Pi_A} \mathbb{E}\big[\langle \theta_A^*, \phi(\pi_A, \pi_B^*(\pi_A))\rangle\big], \qquad \pi_B^*(\pi_A) := \arg\max_{\pi_B \in \Pi_B} \mathbb{E}\big[\langle \theta_B^*, \phi(\pi_A, \pi_B)\rangle\big],$$

$$(3.1) \qquad\qquad (3.2)$$

With a slight abuse of notation, we use $\phi(\pi_A, \pi_B)$ and $\pi_B^*(\pi_A)$ to denote $\mathbb{E}_{\pi_A, \pi_B}[\phi]$ and the best response function in response to policy $\pi_A$, respectively. The expectation are taken with respect to the sub-Gaussian noises.

**Optimization under Parameter Uncertainty:** For some no-regret learning algorithm suppose that after observing $t$ samples, the uncertainty among the parameters $\theta$ is characterized by,

$$\texttt{Ball}(\theta^*, \mathcal{C}_\theta(t)) := \Big\{ \theta : \|\theta^* - \theta\| \le \mathcal{C}_\theta(t) \Big\}. \qquad (3.3)$$

with probability at least $1 - \delta$. In this formulation, $\|\cdot\|$ denotes some norm in the space of parameters. Assuming a *pessimistic* leader, the optimization problem under parameter uncertainty at round $t$ can be expressed as,

$$\pi_A^* := \arg\max_{\pi_A \in \Pi_A} \min_{\theta_A} \mathbb{E}\big[\langle \theta_A, \phi(\pi_A, \pi_B^*(\pi_A))\rangle\big], \qquad \text{s.t.} \quad \theta_A \in \texttt{Ball}(\theta_A^*, \mathcal{C}_\theta(t)), \qquad (3.4)$$

$$\text{where} \quad \pi_B^*(\pi_A) := \arg\max_{\pi_B \in \Pi_B} \max_{\theta_B} \mathbb{E}\big[\langle \theta_B, \phi(\pi_A, \pi_B)\rangle\big], \quad \text{s.t.}\, \theta_B \in \texttt{Ball}(\theta_B^*, \mathcal{C}_\theta(t)). \quad (3.5)$$

Given $\pi_B^*(\cdot)$ in Eq. (3.5), let us define,

$$\underline{\mathcal{H}}(\theta_A^*, t) := \max_{\pi_A \in \Pi_A} \min_{\theta_A} \mathbb{E}\big[\langle \theta_A, \phi(\pi_A, \pi_B^*(\pi_A))\rangle\big], \qquad \text{s.t.} \quad \theta_A \in \texttt{Ball}(\theta_A^*, \mathcal{C}_\theta(t)), \qquad (3.6)$$

$$\overline{\mathcal{H}}(\theta_A^*, t) := \max_{\pi_A \in \Pi_A} \max_{\theta_A} \mathbb{E}\big[\langle \theta_A, \phi(\pi_A, \pi_B^*(\pi_A))\rangle\big], \qquad \text{s.t.} \quad \theta_A \in \texttt{Ball}(\theta_A^*, \mathcal{C}_\theta(t)). \qquad (3.7)$$

We can see from the structure of Eq. (3.4) to Eq. (3.7), the resemblance to a bi-level optimization problem, which can be solved both under perfect information and uncertainty. We provide a discussion of such methods in Appendix B.1.

## 4 ONLINE LEARNING ON THE STACKELBERG MANIFOLD

To enable efficient multi-agent online learning on the Stackelberg manifold, $\mathbf{\Phi}$, we enforce $\mathbf{\Phi}$ to be a *convex manifold*. The convex manifold is a manifold where the geodesic between any two points on the manifold is contained within or forms a geodesically convex set (Def. C.1). Essentially, in a convex manifold, every geodesic between two points is contained within the manifold, adhering to the geodesic convexity property. The formal definitions can be found in Appendix C.1.

### 4.1 STACKELBERG OPTIMIZATION UNDER PERFECT INFORMATION

Provided that we can transform data from the joint action space (or ambient data space) onto a spherical manifold, we can leverage the properties of the D-sphere to determine the best response solution for the Stackelberg follower and optimize the corresponding Stackelberg regret. Consider the reward function structure outlined in Section 2.3. In general, for each agent, $\mu = \langle \theta, \phi \rangle$. Here, $\theta$ represents a $D$-dimensional vector in the manifold space, and we must find the element in $\mathbf{\Phi}$ that maximizes this inner product. In the Stackelberg game, since the leader moves first, they define a restricted subspace on the $\mathbf{\Phi}$. The follower must then optimize within this subspace. Moving forward, $\theta_A$ and $\theta_B$ will be referred to as *objective vectors*.

We define the *divergence angle*, $\alpha_{\text{Div}}$ as the angle between the two objective vectors. Further, we can define the *geodesic distance* between two vectors, denoted as $\mathfrak{G}(\theta_A, \theta_B)$, as follows. For a unit-spherical manifold, this has the definition,

$$\cos(\alpha_{\text{Div}}) := \frac{\langle \theta_A, \theta_B \rangle}{\|\theta_A\|\|\theta_B\|}, \qquad \mathfrak{G}(\theta_A, \theta_B) := \arccos\left( \frac{\langle \theta_A, \theta_B \rangle}{\|\theta_A\|\|\theta_B\|} \right). \qquad (4.1)$$

In a $D$-dimensional sphere, for a cooperative game with no divergence angle, the optimal solution that maximizes the inner product is an element in $\mathbf{\Phi}$ that is collinear with $\theta_A$, mutatis mutandis for $\theta_B$. Lemmas 4.1 to 4.2 establishes a link between solving for the follower's best response, from

Eq. (2.2), and minimizing geodesic distance, in a general sum game. Moving forward, we use the convention $\theta'_A$ and $\theta'_B$ to denote the projection of the objective vectors $\theta_A$ and $\theta_B$ onto $\Phi$.

**Lemma 4.1.** *Geodesic Distance and Closeness: Let $\Phi \subset \mathbb{R}^D$ be a manifold serving as a boundary of a convex set in $\mathbb{R}^D$. Given $\theta$, let $\xi_\theta \in \Phi$ be the point on the manifold that maximizes the dot product $\langle \theta, \xi_\theta \rangle$, and is orthogonal to $\Phi$ at the point of intersection. For any two points on the manifold $\theta'_A, \theta'_B \in \Phi$, if the geodesic distance between $\xi_\theta$ and $\theta'_A$ is greater than the geodesic distance between $\xi_\theta$ and $\theta'_B$, $\mathfrak{G}(\xi_\theta, \theta'_A) > \mathfrak{G}(\xi_\theta, \theta'_B)$, then the dot product satisfies $\langle \theta, \theta'_A \rangle < \langle \theta, \theta'_B \rangle$. (Proof in Appendix C.2.)*

**Lemma 4.2.** *Pure Strategy of the Follower: While optimizing over a convex manifold, proposed in Definition C.2, given any objective vector $\theta$, the linear structure of the reward functions from Eq. (2.15) and Eq. (2.16), and that the subspace induced by $\mathbf{a} \in \mathcal{A}$ forms a geodesically convex subset, as defined in Definition C.1, the optimal strategy of the follower will be that of a pure strategy, such that $\pi_A(\mathbf{b}|\mathbf{a}) \in \{0, 1\}$. (Proof provided in Appendix C.4.)*

The intuition behind Lemmas 4.1 and 4.2 is that the maximum the dot product between $\theta'_A$ and $\theta'_B$ on the convex manifold must be collinear with each other, ensuring the optimal reward. In the case of a convex subspace, the follower acting optimally has no viable alternatives other than a single choice.

## 4.2 REGRET DEFINITIONS

**Definition 4.1.** *Stackelberg Regret: We define Stackelberg regret, denoted as $R_A^T$ for the leader, measuring the difference in cumulative rewards between a best responding follower and an optimal leader in a perfect information setting, against best responding follower and leader exhibiting bounded rationality. The leader policy stipulates that the she acts rationally given the estimates of the expected reward function from the data gathered, as in Eq. (3.7) and Eq. (3.6),*

$$R_A^T := \sum_{t=1}^T \mathbb{E}\left[\max_{\mathbf{a}\in\mathcal{A}} \mu_A(\mathbf{a}, \mathfrak{B}(\mathbf{a})) - \mu_A(\mathbf{a}^t, \mathfrak{B}(\mathbf{a}^t))\right] \leq \sum_{t=1}^T \left(\bar{\mathcal{H}}(\theta_A^*, t) - \underline{\mathcal{H}}(\theta_A^*, t)\right). \quad (4.2)$$

*The leader selects $\mathbf{a}^t$ from policy $\pi_A$ according to their best estimate of $\hat{\theta}_A$ and $\hat{\theta}_B$, following the maximization equations in Eq. (3.4) and Eq. (3.5) respectively.*

The leader commits to a strategy $\pi_A$ aimed at maximizing her reward while accounting for the uncertainty in the follower's response. The leader is free to estimate the follower's response rationally, and within the confidence interval. Our algorithm minimizes the *Stackelberg regret*, providing a no-regret learning process for the leader. To compute the Stackelberg regret of the algorithm, which is defined from the leader's perspective, we must derive a closed form expression for the gap over time between the expected reward under the optimal policy and the expected reward under any algorithm.

**Definition 4.2.** *Simple Regret: Let us define the simple regret, where with probability $1-\delta$ at time $t$,*

$$reg(t) := \langle \theta_A^*, \phi(\mathbf{a}^*, \mathfrak{B}(\mathbf{a}^*)) \rangle - \langle \theta_A^*, \phi(\mathbf{a}^t, \mathfrak{B}(\mathbf{a}^t)) \rangle \leq \bar{\mathcal{H}}(\theta_A^*, t) - \underline{\mathcal{H}}(\theta_A^*, t) \quad (4.3)$$

*This assumes that the leader is acting under the bounded rationality assumption.*

## 4.3 QUANTIFYING UNCERTAINTY ON THE STACKELBERG MANIFOLD

We now revisit the parameter uncertainty constraints introduced in Sec. 2.3, which dictate the uncertainty of a given learning algorithm, characterized by an uncertainty radius $\mathcal{C}_\theta(t)$. Given the feature map $\phi(\cdot)$, which adheres to the linear reward assumptions, particularly with respect to the covariance matrix of the regression (as outlined in Sec. 2.3), the learning leader can apply any bandit learning algorithm that imposes a high-probability bound on the parameter estimate. This constraint is formalized in Eq. (3.3) by the uncertainty region $\mathcal{C}_\theta(t)$. Let us define $\overleftrightarrow{\Phi}_{\mathbf{a}}$ and $\overleftrightarrow{\Phi}_{\mathbf{b}}$ as two subspaces, which we will use to analyze the leader's actions under these uncertainty constraints.

$$\overleftrightarrow{\Phi}_{\mathbf{a}} := \{\phi(\mathbf{a}, \mathbf{b}')|\mathbf{b}' \in \mathcal{B}\}, \quad (4.4) \qquad\qquad \overleftrightarrow{\Phi}_{\mathbf{b}} := \{\phi(\mathbf{a}', \mathbf{b})|\mathbf{a}' \in \mathcal{A}\}, \quad (4.5)$$

where $\overleftrightarrow{\Phi}_{\mathbf{a}}$ and $\overleftrightarrow{\Phi}_{\mathbf{b}}$ are the sub-spaces formed when we fix one of the leader or follower's action, and let the other action vary freely.

**Lemma 4.3.** *Intersection of $\overleftrightarrow{\Phi}_{\mathbf{a}}$ and $\overleftrightarrow{\Phi}_{\mathbf{b}}$: Given a bipartite spherical map $\mathcal{Q}(\cdot)$ from Definition 2.1, with $\mathbf{a}$ parameterizing the azimuthal (latitudinal) coordinates, the cardinality of the intersect between $\overleftrightarrow{\Phi}_{\mathbf{a}}$ and $\overleftrightarrow{\Phi}_{\mathbf{b}}$ will be non-empty. That is, $|\overleftrightarrow{\Phi}_{\mathbf{a}} \cap \overleftrightarrow{\Phi}_{\mathbf{b}}| > 0$. (Proof is in Appendix C.5.)*

The derivation of Lemma 4.3 first comes by isolating the subspaces in terms of angular coordinates. Next, due to the *Poincare-Hopf theorem* (Hopf 1927; Poincaré 1885), the compactness of the smooth Riemmanian manifold imposes strong geometric constraints such that the two subspaces cannot avoid eachother.

**Lemma 4.4.** *Orthogonality of Subspaces $\overleftrightarrow{\Phi}_{\mathbf{a}}$ and $\overleftrightarrow{\Phi}_{\mathbf{b}}$: The two submanifolds $\overleftrightarrow{\Phi}_{\mathbf{a}}$ and $\overleftrightarrow{\Phi}_{\mathbf{b}}$, are orthognal to eachother within $\Phi$. (Proof provided in Appendix C.6.)*

Lemma 4.4 is proven by isolating and taking the partial derivatives of the cartesian coordinates with respect to their spherical coordinates to obtain tangent vectors. Afterwards, by computing the dot product between these two tangents and demonstrating that it equates to 0, we establish their orthogonality.

**Geodesic Isoplanar Subspace Alignment (GISA):** The general methodology in which we can compute the optimal leader strategy for a Stackelberg game, for manifold $\Phi$ that forms a convex boundary, is that the leader can anticipate the follower strategy based on knowledge of follower's reward parameters $\theta'_B$ and the isoplane $\overleftrightarrow{\Phi}_{\mathbf{a}}$. We denote this homeomorphism as $f_1(\overleftrightarrow{\Phi}_{\mathbf{a}}, \theta'_B)$ : $\overleftrightarrow{\Phi}_{\mathbf{a}} \mapsto \overleftrightarrow{\Phi}_{\mathbf{b}_{\mathbf{a}}^*}$. Thereafter, we compute the geodesic distance minimizing distance from $\overleftrightarrow{\Phi}_{\mathbf{b}_{\mathbf{a}}^*}$ to $\theta'_A$ via injective map $f_2(\overleftrightarrow{\Phi}_{\mathbf{b}_{\mathbf{a}}^*}, \theta'_A)$ : $\overleftrightarrow{\Phi}_{\mathbf{b}_{\mathbf{a}}^*} \mapsto \mathbb{R}$. Leader's objective is to find $\mathbf{a} \in \mathcal{A}$ such that it minimizes the composition of $f_1 \circ f_2$, giving us the geodesic distance. This composition is abstractly defined as,

$$\overleftrightarrow{\Phi}_{\mathbf{a}} \underset{f_1(\cdot, \theta'_A)}{\longmapsto} \overleftrightarrow{\Phi}_{\mathbf{b}_{\mathbf{a}}^*} \underset{f_2(\cdot, \theta'_B)}{\longmapsto} \mathfrak{G}(\mathbf{a}, \mathbf{b}_{\mathbf{a}}^*) \in \mathbb{R}, \quad \text{where,} \quad \theta' = \frac{\theta}{\|\theta\|}, \quad \text{for A and B.} \quad (4.6)$$

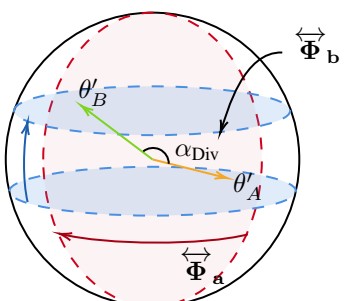

Figure 1: Illustration of isoplanar subspaces for players A and B.

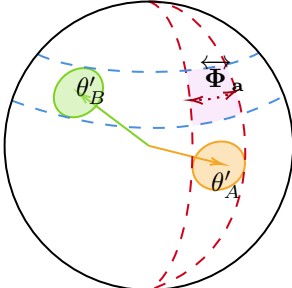

Figure 2: Illustration of geodesic confidence balls for players A and B.

**Diagram Description:** A visualization of the isoplanes $\overleftrightarrow{\Phi}_{\mathbf{a}}$ and $\overleftrightarrow{\Phi}_{\mathbf{b}}$ on a 2-sphere embedded in three dimensions is shown in Fig. 1. The isoplanes are depicted relative to the normalized objective vectors $\theta'_A$ and $\theta'_B$, which lie on the manifold surface, separated by a divergence angle $\alpha_{Div}$. Figure 2 illustrates the geodesic confidence balls, positioned on the surface of the spherical manifold. In three dimensions, it becomes evident that $\overleftrightarrow{\Phi}_{\mathbf{a}}$ and $\overleftrightarrow{\Phi}_{\mathbf{b}}$ are orthogonal at any point of intersection. This intersection, denoted by $\overleftrightarrow{\Phi}_{\mathbf{b}_{\mathbf{a}}}$, is where the joint action emerges, represented by a purple geodesic square indicating the uncertainty region.

**Theorem 1.** *Isoplane Stackelberg Regret: For D-dimensional spherical manifolds embedded in $\mathbb{R}^D$ space, where $\phi(\mathbf{a}, \cdot)$ generates an isoplanes $\overleftrightarrow{\Phi}_{\mathbf{a}}$, and the linear relationship to the reward function in Eq. (2.15) and Eq. (2.16) and Eq. (2.15) and Eq. (2.16), the simple regret, defined in Eq. (4.3), of any learning algorithm with uncertainty parameter uncertainty $\mathcal{C}_\theta(t)$, refer to in Eq. (3.3), is bounded by $\mathcal{O}(\arccos(1 - \mathcal{C}_\theta(t)^2/2))$. (Proof provided in Appendix C.11.)*

The proof of Theorem 1 focuses on analyzing the geodesic distances on $\Phi$ due to uncertainty. First, we argue that any norm-like confidence ball in Cartesian coordinates, $\texttt{Ball}(\cdot)$, can be transformed into a confidence bound into a geodesic distance-based confidence ball, $\texttt{Ball}_{\mathfrak{G}}(\cdot)$, in spherical coordinates (discussed in Lemma C.2 of the Appendix.) Due to orthogonality between $\overleftrightarrow{\Phi}_{\mathbf{a}}$ and

---

**Algorithm 1** Geodesic Isoplanar Subspace Alignment (GISA) Algorithm

---

1: **Input:** Time horizon $T$, and confidence ball $\mathcal{C}_\theta(\cdot)$.
2: **Output:** Estimated optimal leader action $\hat{\mathbf{a}}$.
3: Initialize $\hat{\theta}_A$ and $\hat{\theta}_B$ uniformly at random.
4: Initialize reward and action histories, $\mathcal{U}$ and $\mathcal{H}$ as empty sets $\emptyset$.
5: Construct a Stackelberg embedding $\mathbf{\Phi}$ and feature map $\phi$ per specifications in Sec. 2.2.
6: **for** $t \in 1...T$ **do**:
7:     **if** $\mathfrak{G}(\hat{\theta}_A, \hat{\theta}_B) < 2\mathcal{C}_\theta(t)$ **then**
8:         `Phase 1`: Select uniformly an action on the boundary of $A$'s geodesic confidence ball.
9:         $\mathbf{a} \sim \texttt{Uniform}[\partial\,\texttt{Ball}_\mathfrak{G}(\mathcal{C}_\theta(t))]$ (See Lemma C.2.)
10:     **else**
11:         `Phase 2`: Select $\mathbf{a}$ that minimizes the geodesic distance to $\hat{\theta}_B$ from $\texttt{Ball}_\mathfrak{G}(\mathcal{C}_\theta(t))$.
12:         $\mathbf{a} \leftarrow \underset{\mathbf{a} \in \texttt{Ball}_\mathfrak{G}(\mathcal{C}_\theta(t))}{\arg\min}\ \mathfrak{G}(\mathbf{a}, \hat{\theta}_B)$
13:     **end if**
14:     $\mathbf{b} \leftarrow \underset{\mathbf{b} \in \overleftrightarrow{\mathbf{\Phi}}_\mathbf{a}}{\arg\min}\,\mathfrak{G}(\mathbf{b}, \hat{\theta}_B)$
15:     $\hat{\mathbf{a}}^t, \hat{\mathbf{b}}^t \leftarrow \phi^{-1}(\mathbf{a}, \mathbf{b})$         ▷ Perform an inverse map back to the joint action space.
16:     **yield** $\hat{\mathbf{a}}^t, \hat{\mathbf{b}}^t$, and obtain empirical reward $\mu_A^t, \mu_B^t$.
17:     $\mathcal{H} \leftarrow \mathcal{H} \cup (\hat{\mathbf{a}}^t, \hat{\mathbf{b}}^t), \quad \mathcal{U} \leftarrow \mathcal{U} \cup (\mu_A^t, \mu_B^t)$.
18:     Re-estimate $\hat{\theta}_A$ and $\hat{\theta}_B$ from $\mathcal{H}$ and $\mathcal{U}$, based on Eq. (2.17).
19: **end for**
20: **return** $\hat{\mathbf{a}}^t$

---

$\overleftrightarrow{\mathbf{\Phi}}_\mathbf{b}$, we argue that that the geodesic distance either remains the same or decreases when we projected from any $\texttt{Ball}(\cdot)'$ from $\overleftrightarrow{\mathbf{\Phi}}_\mathbf{a}$ to $\overleftrightarrow{\mathbf{\Phi}}_\mathbf{b}$ (discussed in Lemma C.3 of the Appendix.) This naturally extends to a bound on the maximum diameter of the projected confidence ball on $\overleftrightarrow{\mathbf{\Phi}}_\mathbf{b}$. This constitutes the best and worst possible outcomes due to misspecification in accordance with the formulas in Eq. (3.6) and Eq. (3.7), as expressed in Eq. (4.3), which upper bounds the simple regret.

**Lemma 4.5.** *Pure Strategy of the Leader: Given a spherical manifold, $\mathbf{\Phi}$, and isoplanar subspace, $\overleftrightarrow{\mathbf{\Phi}}_\mathbf{a}$ and $\overleftrightarrow{\mathbf{\Phi}}_\mathbf{b}$ for the longitudinal and lattitudinal subspaces respectively, the optimal strategy of the leader is that of a pure strategy, that is, $\pi_A^*(\mathbf{a}) \in \{0, 1\}$. (Proof is provided in Appendix C.7.)*

Lemma 4.5 argues that the intersection between $\overleftrightarrow{\mathbf{\Phi}}_\mathbf{a}$ and $\overleftrightarrow{\mathbf{\Phi}}_\mathbf{b}$ contains at most one element due to their orthogonality. Consequently, no other actions on the manifold can further maximize the leader's reward. Intuitively, the positive curvature of the manifold ensures that once two non-degenerate isoplanes intersect, the intersection is a unique point that maximizes the dot product between the action and the objective vector.

## 5 EMPIRICAL EXPERIMENTS

We provide three practical instances of Stackelberg games in practice. We benchmark the GISA from Algorithm 1 against a dual-UCB algorithm, where both agents are running a UCB algorithm. Although a simplistic, benchmark, the dual-UCB algorithm does constitute a no-regret learning algorithm (Blum and Mansour 2007).

$\mathbb{R}^1$ **Stackelberg Game:** In this Stackelberg game, the leader selects an action while anticipating the follower's best response. The action spaces of both the leader and the follower are one-dimensional, $\mathbf{a}, \mathbf{b} \in \mathbb{R}^1$. The interaction between nonlinear rewards and penalties requires numerical methods to determine optimal strategies. However, the nonlinear reward functions introduce complexity, resulting in a non-trivial equilibrium. A practical application is energy grid management, where a utility company (leader) sets energy prices or output levels, anticipating the aggregate consumers' (followers) energy usage while accounting for nonlinear feedback such as fluctuating demand or storage limits. (Details and additional experiments are provided in Appendix G.1.)

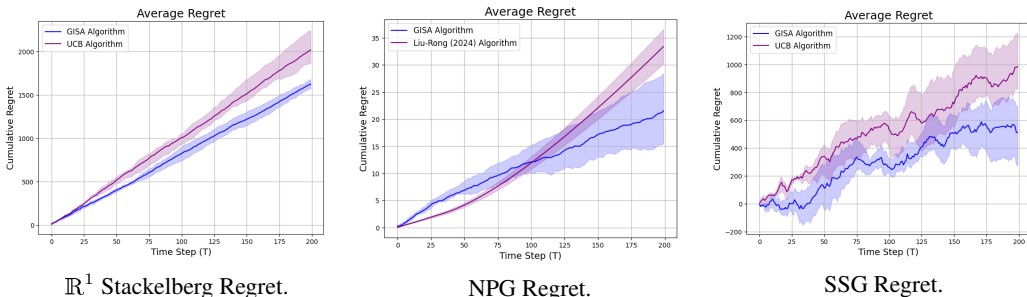

Figure 3: Average cumulative regret performance across three Stackelberg games. Parameters of the simulations outlined in Appendices G.1 - G.3. Uncertainty region denote upper and lower quartile of experimental results.

**The Newsvendor Pricing Game (NPG):** We model two agents in a *Newsvendor pricing game*, with a supplier (leader) and a retailer (follower), inspired by the work of Cesa-Bianchi et al. 2023 and L. Liu and Rong 2024. The action space of the leader is denoted as $\mathbf{a} \in \mathbb{R}^1$, and for the follower as $\mathbf{b} \in \mathbb{R}^2$. The leader and follower interact with each other in a repeated Stackelberg game, modelling a leader-follower supply chain game. The supplier dynamically prices the product, aiming to maximize her reward, while the retailer determines the optimal pricing and order quantity based on the demand distribution according to classical Newsvendor theory (Arrow, Harris, and Marschak 1951; Petruzzi and Dada 1999). The reward function is an abstraction that is a function of stochastic demand, and the reward formats are asymmetric, rendering computation and learning of the Stackelberg equilibrium non-trivial. (We specify the details and additional experiments in Appendix G.2.)

**Stackelberg Security Game (SSG) in $\mathbb{R}^5$:** In this Stackelberg security game (SSG), inspired by the frameworks developed in Balcan et al. 2015 and Zhang and Malacaria 2021, the defender (leader) allocates limited resources across multiple targets, anticipating the attacker's (follower) strategy (i.e. to protect a computer network from malicious intruders). In our example, both players select actions from $\mathbb{R}^5$, where the rewards are governed by the relative difference between their actions (i.e., $\mathbf{a} - \mathbf{b}$) and are subject to quadratic penalties for overextension. Furthermore, resource constraints are modelled via weighted $L_1$-norms, imposing additional limitations on the feasible actions. The Stackelberg equilibrium in this setting is characterized by the leader's optimal resource allocation, taking into account the adversary's best response. The interplay between nonlinear penalties and resource constraints renders the equilibrium computation non-trivial, requiring advanced numerical techniques for tractable solutions. (We specify the details and additional experiments in Appendix G.3.)

## 6 CONCLUSION

This work establishes a foundational connection between Stackelberg games and normalizing neural flows, marking a significant advancement in the study of equilibrium learning and manifold learning. By utilizing normalizing flows to map joint action spaces onto Riemannian manifolds, particularly spherical ones, we offer a novel, theoretically grounded framework with formal guarantees on simple regret. This approach represents the first application of normalizing flows in game-theoretic settings, specifically Stackelberg games, thereby opening new avenues for learning on convex manifolds. Our empirical results, grounded in realistic simulation scenarios, highlight promising improvements in both computational efficiency and regret minimization, underscoring the broad potential of this methodology across multiple domains in economics and engineering. Despite potential challenges related to numerical accuracy for the neural flow network, this integration of manifold learning into game theory nevertheless exhibits strong implications for online learning, positioning neural flows as a promising tool for both machine learning and strategic decision-making.

## ETHICS STATEMENT

We affirm that this research adheres to the ICLR Code of Ethics. All simulations and methodologies were conducted with integrity and transparency, without harm to individuals, groups, or the environment. We ensured that the theoretical and practical contributions of this work are aimed at advancing knowledge in a responsible and ethical manner, with no misuse or malicious application of the techniques proposed. Additionally, no conflicts of interest or external influences have compromised the objectivity or scientific rigour of this work.

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

## A  KEY ASSUMPTIONS AND DEFINITIONS

### A.1  COMPACT AND CLOSED SETS

In this formal definition, $\Phi$ is both compact and closed in the product space $\mathcal{A} \times \mathcal{B}$. A set $\Phi$ is compact if for every open cover $\{U_i\}_{i \in I}$ of $\Phi$, there exists a finite subcover such that $\Phi \subseteq \bigcup_{k=1}^{n} U_{i_k}$, where $U_{i_k}$ are open sets in $\mathcal{A} \times \mathcal{B}$. This ensures that $\Phi$ is "contained" in a finite manner within the space, even if $\mathcal{A} \times \mathcal{B}$ is infinite. Furthermore, $\Phi$ is closed if its complement, $\Phi^c = (\mathcal{A} \times \mathcal{B}) \setminus \Phi$, is open. This implies that $\Phi$ contains all its limit points, making it a complete set within the topological space. Thus, $\Phi$ is a compact and closed subset of $\mathcal{A} \times \mathcal{B}$, meaning that it is both bounded and contains its boundary, providing useful properties for convergence and stability in this space.

$$\forall \{U_i\}_{i \in I}, \quad \Phi \subseteq \bigcup_{i \in I} U_i \implies \exists \{U_{i_1}, U_{i_2}, \ldots, U_{i_n}\} \text{ such that } \Phi \subseteq \bigcup_{k=1}^{n} U_{i_k}, (\mathcal{A} \times \mathcal{B}) \setminus \Phi \text{ is open.} \tag{A.1}$$

### A.2  ASSUMPTIONS ON LINEAR REWARD FUNCTION

1. **Covariance Matrix**:

$$\Sigma_T := \sum_{t=1}^{T} \phi(\mathbf{a}^t, \mathbf{b}^t) \phi(\mathbf{a}^t, \mathbf{b}^t)^\top + \lambda_{\text{reg}} I \tag{A.2}$$

$\phi(\mathbf{a}^t, \mathbf{b}^t)$ must ensure that the covariance matrix $\Sigma_T^{-1}$ (a.k.a. the inverse of the covariance matrix) is sufficiently large for effective learning.

2. **Norm Bounds**:

$$\|\phi(\mathbf{a}^t, \mathbf{b}^t)\|_{\Sigma_T^{-1}} \equiv \sqrt{\phi(\mathbf{a}^t, \mathbf{b}^t) \Sigma_T^{-1} \phi(\mathbf{a}^t, \mathbf{b}^t)^\top} \tag{A.3}$$

$\|\phi(\mathbf{a}^t, \mathbf{b}^t)\|_{\Sigma_T^{-1}}$ must be small to ensure efficient uncertainty reduction.

3. **Regularization Effect**: Regularization parameter $\lambda_{\text{reg}}$ balances bias and variance, affecting sample complexity.

4. **Positive Semi-Definiteness**: $\Sigma_T^{-1}$ is positive semi-definite (PSD).

### A.3  DISCRETE MEASURE INTERPRETATION

Let $\{x_1, x_2, \ldots, x_n\}$ be a set of discrete points in $\mathbb{R}^n$. We define the measure $\alpha$ on these points as,

$$\alpha = \sum_{i=1}^{n} \alpha(\{x_i\}) \delta_{x_i} \tag{A.4}$$

where $\delta_{x_i}$ is the Dirac measure centered at $x_i$. The integral of a function $f : \mathbb{R}^n \to \mathbb{R}$ with respect to the measure $\alpha$ is given by,

$$\int_{\mathbb{R}^n} f(x) \, d\alpha(x) = \sum_{i=1}^{k} \alpha(\{x_i\}) f(x_i) \tag{A.5}$$

### A.4  DEFINITION OF RIEMANN MANIFOLD

A Riemannian manifold, expressed as $\Phi$, consists of a smooth manifold $\Phi$ equipped with a smoothly varying collection of inner products $\omega_p$ on each tangent space $T_p \Phi$ at every point $p \in \Phi$. This assignment $\omega_p : T_p \Phi \times T_p \Phi \to \mathbb{R}$ is positive-definite, meaning it measures angles and lengths in a consistent and non-degenerate manner. Consequently, each vector $\mathbf{v} \in T_p \Phi$ inherits a smoothly

defined norm $\|\mathbf{v}\|_p = \sqrt{\omega_p(\mathbf{v}, \mathbf{v})}$. This structure allows $\boldsymbol{\Phi}$ to possess a locally varying yet smoothly coherent geometric framework.

### A.5 STOCHASTIC PERTURBATION FUNCTION

To model uncertainty in the joint action space, we introduce a stochastic perturbation over the leader and follower actions. Specifically, we define a small, one-step random perturbation function $\mathfrak{J}(\mathbf{a}, \mathbf{b})$, where $\mathbf{a} \in \mathbb{R}^m$ and $\mathbf{b} \in \mathbb{R}^n$ are the actions of the leader and follower, respectively. The perturbed joint action is given by:

$$\mathfrak{J}(\mathbf{a}, \mathbf{b}) = (\mathbf{a}', \mathbf{b}') = (\mathbf{a} + \epsilon_a, \mathbf{b} + \epsilon_b) \tag{A.6}$$

where $\epsilon_a \in \mathbb{R}^m$ and $\epsilon_b \in \mathbb{R}^n$ are independent Gaussian perturbations with zero mean and variance $\sigma_a^2$ and $\sigma_b^2$, respectively:

$$\epsilon_a \sim \mathcal{N}(0, \sigma_a^2 I_m), \quad \epsilon_b \sim \mathcal{N}(0, \sigma_b^2 I_n) \tag{A.7}$$

Here, $\sigma_a$ and $\sigma_b$ are scalar diffusion parameters controlling the magnitude of the perturbation, and $I_m$ and $I_n$ are identity matrices of size $m \times m$ and $n \times n$, ensuring isotropic perturbations in each component of $\mathbf{a}$ and $\mathbf{b}$.

In component form, this perturbation can be written as:

$$a_i' = a_i + \epsilon_{a_i}, \quad \epsilon_{a_i} \sim \mathcal{N}(0, \sigma_a^2) \tag{A.8}$$

$$b_j' = b_j + \epsilon_{b_j}, \quad \epsilon_{b_j} \sim \mathcal{N}(0, \sigma_b^2) \tag{A.9}$$

This formulation introduces small, independent, and isotropic random deviations from the original actions, modeling the stochastic uncertainty in the decision-making process.

### A.6 GEODESIC REPULSION LOSS

To encourage an even distribution of points on the spherical manifold, we employ the *Geodesic repulsion loss*, which penalizes pairs of points that are too close in geodesic distance. This loss function facilitates the spreading out of points uniformly over the sphere, preventing clustering.

**Geodesic Distance:** Let $\mathbf{y}_i, \mathbf{y}_j \in \mathbb{R}^D$ be points on the surface of a Riemannian manifold denoted as $\mathfrak{G}(\mathbf{y}_i, \mathbf{y}_j)$ in the abstract sense. For a unit sphere it would hold that $\|\mathbf{y}_i\| = \|\mathbf{y}_j\| = 1$). The geodesic distance between two points on the sphere is the angle between them, which can be computed from their dot product,

$$\mathfrak{G}(\mathbf{y}_i, \mathbf{y}_j) = \arccos\left(\mathbf{y}_i^\top \mathbf{y}_j\right), \tag{A.10}$$

where $\mathbf{y}_i^\top \mathbf{y}_j$ is the dot product of $\mathbf{y}_i$ and $\mathbf{y}_j$.

**Repulsion Term:** To penalize pairs of points that are close in geodesic distance, we use an exponential decay function, which strongly penalizes small distances:

$$\exp\left(-\frac{\mathfrak{G}(\mathbf{y}_i, \mathbf{y}_j)}{\gamma}\right), \tag{A.11}$$

where $\gamma > 0$ is a sensitivity parameter controlling how strongly the loss reacts to small distances. A smaller $\gamma$ enforces stronger repulsion between nearby points.

**Geodesic Repulsion Loss:** The total Geodesic Repulsion Loss is computed as the sum of repulsion terms over all pairs of points, excluding the diagonal (self-repulsion),

$$\mathcal{L}_{\text{repulsion}} = \sum_{i=1}^{n} \sum_{j=1, j\neq i}^{n} \exp\left(-\frac{\arccos\left(\mathbf{y}_i^\top \mathbf{y}_j\right)}{\gamma}\right), \tag{A.12}$$

where $n$ is the number of points on the manifold. The geodesic distance $\mathfrak{G}(\mathbf{y}_i, \mathbf{y}_j)$ is computed using the angle between $\mathbf{y}_i$ and $\mathbf{y}_j$, ensuring that points are uniformly spaced across the spherical manifold.

To avoid penalizing points for being close to themselves, we exclude the self-repulsion terms by masking the diagonal elements in the pairwise distance computation,

$$\mathfrak{G}(\mathbf{y}_i, \mathbf{y}_i) = 0, \quad \text{for all } i. \tag{A.13}$$

This formulation ensures that points are pushed apart when their geodesic distances are too small, leading to a more uniform distribution on the manifold, which is critical for preserving the geometry of the learned representation.

### A.7 NEGATIVE LOG-LIKELIHOOD LOSS FOR NORMALIZING FLOWS

Let $x \in \mathbb{R}^d$ be an input data point, and let $f : \mathbb{R}^d \to \mathbb{R}^d$ be an invertible transformation defined by the normalizing flow. The transformation $f$ maps the input data $x$ to a latent variable $z = f(x)$ that follows a simple base distribution $p_Z(z)$. Assume that the base distribution is a standard normal distribution, $Z \sim \mathcal{N}(0, I_d)$, with the probability density function (PDF) given by,

$$p_Z(z) = \frac{1}{(2\pi)^{d/2}} \exp\left(-\frac{1}{2}\|z\|^2\right). \tag{A.14}$$

The log probability under this distribution is,

$$\log p_Z(z) = -\frac{1}{2}\|z\|^2 - \frac{d}{2}\log(2\pi). \tag{A.15}$$

Using the change of variables formula, the probability density of $x$ under the model is related to the base distribution via the transformation $f$ as follows,

$$p_X(x) = p_Z(f(x)) \left|\det \frac{\partial f(x)}{\partial x}\right|. \tag{A.16}$$

Where $\frac{\partial f(x)}{\partial x}$ is the Jacobian matrix of $f$ with respect to $x$, and $\left|\det \frac{\partial f(x)}{\partial x}\right|$ is the absolute value of the determinant of the Jacobian.

**NLL Loss:** The negative log-likelihood (NLL) loss for a single data point $x$ is defined as,

$$\mathcal{L}_\phi^N(x) = -\log p_X(x) = -\left[\log p_Z(f(x)) + \log\left|\det \frac{\partial f(x)}{\partial x}\right|\right]. \tag{A.17}$$

Substituting the log probability of $z = f(x)$ under the base distribution:

$$\mathcal{L}_\phi^N(x) = \frac{1}{2}\|f(x)\|^2 + \frac{d}{2}\log(2\pi) - \log\left|\det \frac{\partial f(x)}{\partial x}\right|. \tag{A.18}$$

For a dataset $\{x_i\}_{i=1}^n$, the total NLL loss is the average over all data points:

$$\mathcal{L}_\phi^N = \frac{1}{n}\sum_{i=1}^n \left(\frac{1}{2}\|f(x_i)\|^2 + \frac{d}{2}\log(2\pi) - \log\left|\det \frac{\partial f(x_i)}{\partial x_i}\right|\right). \tag{A.19}$$

The objective of training is to minimize $\mathcal{L}_\phi^N$, ensuring that the transformed latent variables $z = f(x)$ follow the base distribution and the transformation $f$ appropriately adjusts the volume of space via the Jacobian determinant.

# B  OPTIMIZATION ALGORITHMS

## B.1  BI-LEVEL OPTIMIZATION STRUCTURE

**Bi-level Optimization Structure:** The optimization problems represented by Eqs. (3.4) and (3.5) exhibit the structure of a *bi-level optimization* problem (Balling and Sobieszczanski-Sobieski 1995; Beck, Ljubić, and Schmidt 2023; Sinha, Malo, and Deb 2017). Generally, a bilevel optimization problem comprises an upper-level optimization task with an embedded lower-level problem, where the solution to the upper-level problem depends on the solution to the lower-level one. Two conventional methods have been employed to address the bilevel optimization problem. The first leverages the Karush-Kuhn-Tucker (KKT) conditions to exploit the optimality of the lower-level problem (see Appendix B.2). The second employs gradient-based algorithms like gradient ascent (discussed in Appendix B.3). Both approaches, however, have notable limitations. KKT conditions assume strong convexity or pseudo-convexity, making them unsuitable for many non-convex settings, while gradient-based methods, in addition to being computationally inefficient, often struggle or converge poorly when weak-convexity is not guaranteed. Moreover, these methods typically assume optimization under perfect information, whereas we focus on learning-based frameworks with uncertainty due to sampling.

## B.2  KKT REFORMULATION FOR SOLVING STACKELBERG OPTIMIZATION PROBLEMS

The bi-level optimization structure can be solved via reformulating the problem as a bilevel optimization problem via the Karush-Kuhn-Tucker (KKT) conditions. It assumes convexity and differentiability in the embedded space and transforms the original bilevel problem into a single-stage optimization problem via the KKT conditions.

$$
\begin{aligned}
&\max_{\pi_A, \pi_B, \lambda} \ \langle \theta_A, \ \phi(\pi_A, \pi_B) \rangle \\
&\text{s.t. } \pi_A \in \Pi_A \\
&\quad \nabla_{\pi_B} \langle \theta_B, \ \phi(\pi_A, \pi_B) \rangle + \sum_{i=1}^{\ell} \lambda_i \nabla_{\pi_B} g_i(\pi_B) = 0 \\
&\quad g(\pi_B) \geq 0 \\
&\quad \lambda \geq 0 \\
&\quad \lambda^\top g(\pi_B) = 0
\end{aligned}
\tag{B.1}
$$

where $\Pi_B = \{\pi_B | g(\pi_B) \geq 0\}$ and $g_i$ represents the $i$-th constraint of $\Pi_B$. Specifically, it requires the convexity of the lower level problem (3.2). Otherwise, KKT complementarity conditions turns the problem into a nonconvex and nonlinear problem even $\pi_B$ is a set of linear constraints. And the problem is incapable to solve under normal nonconvex and nonlinear algorithm. In addition, Slater's constraint qualification is required to ensure that the solution under KKT reformulation is the solution of original bilevel problem.(Allende and Still 2013)The reformulation involves converting non-linear constraints into a convex hull, thus simplifying the problem into a linear program (LP). Sensitivity analysis can be then performed to understand how changes in constraints impact the solution, with particular attention to the effects of shrinking parameters on the objective function. The approach is utilizes the application of the Weak Duality Theorem to analyze sensitivity.

## B.3  GRADIENT ASCENT APPROACH FOR SOLVING BILEVEL OPTIMIZATION PROBLEMS

Another approaches is transforming Stackelberg game into the the bilevel optimization problem. Namely, we are interested in the following problem,

$$
\begin{aligned}
&\min_{x \in \mathbb{R}^d, y \in y^*(x)} \ f(x, y), \ \text{(Upper-Level)} \\
&\text{s.t. } y^*(x) \equiv \arg \min_{y \in Y} g(x, y). \ \text{(Lower-Level)}
\end{aligned}
\tag{B.2}
$$

The gradient-based algorithms have seen a growing interest in the bilevel problem (Huang 2024; Huang et al. 2022; Ji, J. Yang, and Liang 2021; R. Liu et al. 2021; Sato, Tanaka, and Takeda 2021; Xiao, Lu, and Chen 2023). To measure the stationarity of the lower-level problem, Polyak-Lojasiewicz(PL) condition on $g(x, \cdot)$ is widely applied to show the last-iterate convergence of $\|\nabla_x f(x^t, y^\star(x^t))\|$, i.e,

$$\|\nabla_y g(x,y)\|^2 \geq 2\mu(g(x,y) - \min_z g(x,z)). \tag{B.3}$$

where $\mu$ is a positive constant. This condition relaxes the strong convexity but is still not satisfied for the polynomial function $g(x,y) = y^4$. Also, the lower level function $g(x, \cdot)$ needs to be differentiable in $R^d$. For the Stackelberg game, this is not the case since the follower's strategy $\pi_B \in \Pi_B$.

Interestingly, should the objective functions be differentiable, one strategy to do this optimization is via gradient descent. Of course the gradient descent algorithm would have to be reformulated to accommodate to a finite amount of traversals based on the gradient update (Sato, Tanaka, and Takeda 2021) (Franceschi et al. 2017) (Naveiro and Insua 2019).

### B.4 TECHNICAL NOTE: CONVERSION OF ABSOLUTE VALUE CONSTRAINTS INTO REGULAR LP CONSTRAINTS

Suppose there exists $D$ dimensions on the L1 norm. Wnd we have the constraint,

$$\|\mathbf{x} - \mathbf{c}\|_1 \leq D, \quad \text{expressed as,} \quad \sum_{i=1}^{D} |x_i - c_i| \leq C \tag{B.4}$$

This can be expressed as,

$$\begin{align}
z_i &\geq x_i - c_d & \text{for } i = 1, 2, \ldots, D \tag{B.5} \\
z_i &\geq -(x_i - c_d) & \text{for } i = 1, 2, \ldots, D \tag{B.6}
\end{align}$$

$$\sum_{i=1}^{D} z_i \leq C \tag{B.7}$$

$$z_i \geq 0 \qquad \text{for } i = 1, 2, \ldots, D \tag{B.8}$$

By introducing a new dummy variable $z_i$, we and adding $2D + 1$ additional constraints, we can express this now as a standard linear program.

## C  Topology & Geodesy

### C.1  Convex Manifold Definitions:

**Definition C.1.** *Geodesically Convex Sets: Let* $(\mathbf{\Phi}, h)$ *be a Riemannian manifold, where* $\mathbf{\Phi}$ *is a smooth manifold and* $h(\mathbf{a}, \mathbf{b})$ *is a Riemannian metric on* $\mathbf{\Phi}$ *(i.e. innner product). A subset* $\mathcal{S}_{\mathbf{\Phi}} \subseteq \mathbf{\Phi}$ *is said to be geodesically convex if for any two points* $\mathbf{a}, \mathbf{b} \in \mathcal{S}_{\mathbf{\Phi}}$*, there exists a geodesic* $\tau : d \in [0,1] \to \mathbf{\Phi}$ *parameterized by* $d$ *such that,*

$$\tau(0) = \mathbf{a}, \ \tau(1) = \mathbf{b}, \qquad and, \ \tau(d) \in \mathcal{S}_{\mathbf{\Phi}}, \quad \forall d \in [0,1]. \tag{C.1}$$

Where $d$ can be viewed as a parameter proportional to the distance travelled along the geodesic. In other words, a set $\mathcal{S}_{\mathbf{\Phi}}$ is geodesically convex if for any two points in $\mathcal{S}_{\mathbf{\Phi}}$, there exists a geodesic between these points that lies entirely within $\mathcal{S}_{\mathbf{\Phi}}$.

**Definition C.2.** *Convex Manifolds: A convex manifold is a manifold where the geodesic between any two points on the manifold falls within, or constitutes, a geodesically convex set* $\mathcal{S}_{\mathbf{\Phi}}$*, as per Definition C.1.*

### C.2  Proof of Lemma 4.1

**Geodesic Distance and Closeness to** $\xi_\theta$**:** Let $\Phi \subset \mathbb{R}^D$ be a manifold serving as a boundary of a convex set in $\mathbb{R}^D$. Given $\theta$, let $\xi_\theta \in \Phi$ be the point on the manifold that maximizes the dot product $\langle \theta, \xi_\theta \rangle$, and is orthogonal to $\Phi$ at the point of intersection. For any two points on the manifold $\theta'_A, \theta'_B \in \Phi$, if the geodesic distance between $\xi_\theta$ and $\theta'_A$ is greater than the geodesic distance between $\xi_\theta$ and $\theta'_B$, $\mathfrak{G}(\xi_\theta, \theta'_A) > \mathfrak{G}(\xi_\theta, \theta'_B)$, then the dot product satisfies $\langle \theta, \theta'_A \rangle < \langle \theta, \theta'_B \rangle$.

*Proof.* **Geodesic Distance and Closeness to** $\xi_\theta$**:** Since $\mathcal{M}$ is a smooth, compact manifold bounding a convex region, the geodesic distance between two points on $\mathcal{M}$, say $\xi_1, \xi_2 \in \mathcal{M}$, is defined as the shortest path along the manifold $\mathfrak{G}(\xi_1, \xi_2)$ between $\xi_1$ and $\xi_2$. For convex manifolds, the geodesic distance behaves similarly to the distance on the surface of a sphere: an increase in the geodesic distance from $\xi_\theta$ to another point on the manifold corresponds to an increase in the angle between the tangent vector at $\xi_\theta$ and the vectors corresponding to points on the manifold. Hence, if $\mathfrak{G}(\xi_\theta, \theta'_A) > \mathfrak{G}(\xi_\theta, \theta'_B)$, the angle between $\xi_\theta$ and $\theta'_A$ is larger than the angle between $\xi_\theta$ and $\theta'_B$.

**Dot Product and Angle:** The dot product $\langle \theta, \xi \rangle$ between a normal vector $\theta$ at $\xi_\theta$ and a point $\xi$ on the manifold is given by:

$$\langle \theta, \xi \rangle = \|\theta\| \|\xi\| \cos(\alpha) \tag{C.2}$$

where $\alpha$ is the angle between the vectors $\theta$ and $\xi$. Since $\theta = \frac{\xi_\theta}{\|\xi_\theta\|}$ (as $\xi_\theta$ is a unit vector), the angle between $\theta$ and any point $\xi$ on the manifold depends only on the angle between $\xi_\theta$ and $\xi$. Since $\mathfrak{G}(\xi_\theta, \theta'_A) > \mathfrak{G}(\xi_\theta, \theta'_B)$ implies that the angle between $\xi_\theta$ and $\theta'_A$ is larger than the angle between $\xi_\theta$ and $\theta'_B$, we have:

$$\cos(\alpha_{\theta'_A}) < \cos(\alpha_{\theta'_B}), \tag{C.3}$$

where $\alpha_{\theta'_A}$ is the angle between $\theta$ and $\theta'_A$, and $\alpha_{\theta'_B}$ is the angle between $\theta$ and $\theta'_B$.

**Conclusion on Dot Products:** Since the dot product is proportional to the cosine of the angle between the vectors, and $\cos(\alpha_{\theta'_A}) < \cos(\alpha_{\theta'_B})$, it follows that:

$$\langle \theta, \theta'_A \rangle = \|\theta\| \|\xi_{\theta'_A}\| \cos(\alpha_{\theta'_A}) < \langle \theta, \theta'_B \rangle = \|\theta\| \|\xi_{\theta'_B}\| \cos(\alpha_{\theta'_B}). \tag{C.4}$$

Therefore,

$$\langle \theta, \theta'_A \rangle < \langle \theta, \theta'_B \rangle. \tag{C.5}$$

$\square$

### C.3  LEMMA C.1

**Lemma C.1.** *Maximization on a Manifold: Given a smooth manifold* $\Phi$, *and objective vector* $\theta$, *the element on a manifold which optimizes* $\langle \phi, \theta \rangle$ *is the element whose normal vector's tangent plane* $\overrightarrow{\mathbf{n}}_{\Phi}$ *is collinear with* $\theta$. *(Proof in Appendix C.3.)*

*Proof.* Let $\Phi \subset \mathbb{R}^D$ be the unit sphere, defined as:

$$\Phi = \{\phi \in \mathbb{R}^D \mid \|\phi\| = 1\}.$$

Given a vector $\theta_A \in \mathbb{R}^D$, we aim to find the point $\phi^*$ on the sphere that maximizes the inner product $\langle \phi, \theta_A \rangle$. This can be formally stated as the following optimization problem:

$$\begin{aligned} \underset{\phi \in \mathbb{R}^D}{\text{maximize}} \quad & \langle \phi, \theta_A \rangle \\ \text{subject to} \quad & \|\phi\| = 1. \end{aligned}$$

**Optimization Formulation:** The problem is a constrained optimization problem where the objective is to maximize the dot product $\langle \phi, \theta_A \rangle$ and the constraint ensures that $\phi$ lies on the unit sphere. Mathematically:

$$\begin{aligned} \underset{\phi \in \mathbb{R}^D}{\text{maximize}} \quad & \langle \phi, \theta_A \rangle \\ \text{subject to} \quad & g(\phi) = \|\phi\|^2 - 1 = 0. \end{aligned}$$

Here, $g(\phi)$ represents the constraint that $\phi$ lies on the unit sphere. $\square$

### C.4  PROOF OF LEMMA 4.2

**Pure Strategy of the Follower:** While optimizing over a convex manifold, proposed in Definition C.2, given any objective vector $\theta$, the linear structure of the reward functions from Eq. (2.15) and Eq. (2.16), and that the subspace induced by $\mathbf{a} \in \mathcal{A}$ forms a geodesically convex subset, as defined in Definition C.1, the optimal strategy of the follower, will be that of a pure strategy, such that $\pi_A(\mathbf{b}|\mathbf{a}) \in \{0, 1\}$.

*Proof.* The goal is to show that the follower's optimal strategy $\pi_A(\mathbf{b}|\mathbf{a})$ is a pure strategy, such that $\pi_A(\mathbf{b}|\mathbf{a}) \in \{0, 1\}$. Let the objective vector $\theta \in \mathbb{R}^D$ define the direction of optimization, with the reward function given by,

$$\mu(\mathbf{a}, \mathbf{b}) = \langle \phi(\mathbf{a}, \mathbf{b}), \theta \rangle, \tag{C.6}$$

where $\phi : \mathcal{A} \times \mathcal{B} \to \mathbb{R}^D$ is a feature map.

Since $\mathbf{\Phi}$ is geodesically convex, for any point $\mathbf{a} \in \mathcal{A}$, there exists a unique geodesic that connects the subspace formed by fixing $\mathbf{a}$, denoted as $\mathbf{\Phi}_{\mathbf{a}} \equiv \phi(\mathbf{a}, \cdot)$ to any other point $g \in \mathbf{\Phi}$. By Lemma 4.1 in order to maximize the follower's reward $\mu_B$, we must find the shortest geodesic distance, $\mathfrak{G}(\cdot)$, to $\theta'_A$ within $\mathcal{S}_{\mathbf{\Phi}}$. We express this as,

$$\phi(\mathbf{a}, \mathbf{b}^*) = \arg\min_{g \in \mathcal{S}_{\mathbf{\Phi}}} \mathfrak{G}(\mathbf{a}, \mathbf{b}), \tag{C.7}$$

Since $\mathbf{\Phi}$ is convex, this minimizer is unique. The reward function $\mu_B(\mathbf{a}, \mathbf{b})$ depends on the inner product $\langle \phi(\mathbf{a}, \mathbf{b}), \theta_B \rangle$. As this structure is linear with respect to $\phi(\mathbf{a}, \mathbf{b})$, maximizing the reward is equivalent to minimizing the geodesic distance from $\phi(\mathbf{a}, \mathbf{b})$ to the objective vector $\theta$. Since this minimizer is unique by geodesic convexity, the follower's optimal strategy will correspond to this unique solution $\mathbf{b}^*$ given $\mathbf{a}$. As there are no alternative solutions for $\phi(\mathbf{b}^*, \cdot)$ given $\mathbf{a}$. Because $\phi(\cdot)$ is a bijective mapping, we conclude that any probablistic mapping function must adhere to $\pi_A(\mathbf{b}|\mathbf{a}) \in \{0, 1\}$.

$\square$

## C.5    Proof of Lemma 4.3

**Intersection of $\overleftrightarrow{\Phi}_{\mathbf{a}}$ and $\overleftrightarrow{\Phi}_{\mathbf{b}}$:** Given a bipartite spherical map $\mathcal{Q}(\cdot)$ from Definition 2.1, with $\mathbf{a}$ parameterizing the azimuthal (latitudinal) coordinates, the cardinality of the intersect between $\overleftrightarrow{\Phi}_{\mathbf{a}}$ and $\overleftrightarrow{\Phi}_{\mathbf{b}}$ will be non-empty. That is, $|\overleftrightarrow{\Phi}_{\mathbf{a}} \cap \overleftrightarrow{\Phi}_{\mathbf{b}}| > 0$.

*Proof.* Given two distinct points $\theta'_A$ and $\theta'_B$, we define the *isoplane*, $\overleftrightarrow{\Phi}_{\mathbf{a}}$, as the submanifold formed by fixing a subset of spherical coordinates $(\gamma_1^{(A)}, \ldots, \gamma_k^{(A)})$, including the azimuthal angle $\nu^{(A)}$, and allowing the remaining coordinates to vary. Similarly, the isoplane at $\theta'_B$ is formed by fixing a different subset of spherical coordinates $(\gamma_{k+1}^{(B)}, \ldots, \gamma_{D-2}^{(B)})$, while allowing the rest to vary.

If $\mathbf{\Phi}$ is a compact, orientable, smooth manifold without boundary, and $\vec{X}$ is a smooth vector field on $\mathbf{\Phi}$ with isolated zeros, the *Poincaré-Hopf theorem* states that,

$$\sum_{\mathbf{P} \in \text{Zeroes}(\vec{X})} \text{Index}(\vec{X}, \mathbf{P}) = \chi(\mathbf{\Phi}), \tag{C.8}$$

where $\chi(\mathbf{\Phi})$ is the Euler characteristic of the manifold, and $\text{Index}(\vec{X}, \mathbf{P})$ denotes the index of the vector field at point $\mathbf{P}$. The compactness of $\mathcal{S}^{D-1}$ imposes strong geometric constraints: subspaces or submanifolds (such as isoplanes) embedded within $\mathcal{S}^{D-1}$ must intersect unless they are specifically configured to avoid each other (e.g., in certain degenerate cases of orthogonality). To dive deeper, and provide a more fundamental and intuitive analysis, let $\Psi_\theta$ represent the intersection of isoplanar subspaces,

$$\Psi_\theta = \overleftrightarrow{\Phi}_{\mathbf{a}} \cap \overleftrightarrow{\Phi}_{\mathbf{b}}. \tag{C.9}$$

First, the compactness of the unit sphere $\mathcal{S}^{D-1}$ implies that any sufficiently dimensional subspaces embedded in the manifold cannot be disjoint. The intersection may be a single point or a higher-dimensional subset, depending on the number of coordinates fixed and the degrees of freedom allowed for the remaining coordinates. Secondly, even in the case where the isoplanes at $\overleftrightarrow{\Phi}_{\mathbf{a}}$ and $\overleftrightarrow{\Phi}_{\mathbf{b}}$ are orthogonal, the fact that the subspaces are embedded in a compact, orientable manifold forces them to intersect. This intersection result is a consequence of the general principles of intersection theory in compact manifolds, which asserts that two subspaces of sufficient dimension within a compact manifold must intersect unless they are orthogonal in all directions. However, since we are working with constrained isoplanes that do not span the entire manifold, even orthogonal subspaces are forced to intersect due to the lack of space for complete disjointness. Therefore,

$$|\Psi_\theta| > 0. \tag{C.10}$$

$\square$

### C.6 PROOF OF LEMMA 4.4

**Orthogonality of Subspaces $\overleftrightarrow{\Phi}_{\mathbf{a}}$ and $\overleftrightarrow{\Phi}_{\mathbf{b}}$:** The two submanifolds $\overleftrightarrow{\Phi}_{\mathbf{a}}$ and $\overleftrightarrow{\Phi}_{\mathbf{b}}$, are orthognal to eachother within $\Phi$.

We consider the spherical manifold $S^{D-1}$, embedded in $\mathbb{R}^D$, where points are parameterized using $D-1$ angular coordinates. These coordinates are composed of latitude-like angles $\nu_1, \ldots, \nu_{D-2}$ and a longitude-like angle $\gamma$. The Cartesian coordinates, $\mathbf{x} = [x_1, x_2, \ldots, x_D]^\intercal$, of a point on $S^{D-1}$ are expressed as:

$$x_1 = \prod_{i=1}^{D-2} \sin(\nu_i) \cos(\gamma),$$

$$x_2 = \prod_{i=1}^{D-2} \sin(\nu_i) \sin(\gamma),$$

$$x_3 = \prod_{i=1}^{D-3} \sin(\nu_i) \cos(\nu_{D-2}),$$

$$x_4 = \prod_{i=1}^{D-4} \sin(\nu_i) \cos(\nu_{D-3}),$$

$$\vdots$$

$$x_{D-1} = \sin(\nu_1) \cos(\nu_2),$$
$$x_D = \cos(\nu_1).$$

We aim to show that the subspaces generated by fixing $\theta'_A$, the set of latitude-like angles, and fixing $\theta'_B$, the longitude-like angle, are orthogonal. To this end, we compute the tangent vectors of the manifold in the directions of these angular coordinates.

First, we compute the partial derivative of each coordinate with respect to $\gamma$. The coordinates $x_1$ and $x_2$ explicitly depend on $\gamma$, while the other coordinates $x_3, \ldots, x_D$ do not. Therefore, we have,

$$\frac{\partial x_1}{\partial \gamma} = \frac{\partial}{\partial \gamma} \left( \prod_{i=1}^{D-2} \sin(\nu_i) \cos(\gamma) \right) = - \prod_{i=1}^{D-2} \sin(\nu_i) \sin(\gamma),$$

$$\frac{\partial x_2}{\partial \gamma} = \frac{\partial}{\partial \gamma} \left( \prod_{i=1}^{D-2} \sin(\nu_i) \sin(\gamma) \right) = \prod_{i=1}^{D-2} \sin(\nu_i) \cos(\gamma),$$

$$\frac{\partial x_j}{\partial \gamma} = 0, \quad \forall j \geq 3.$$

Thus, the complete partial derivative with respect to $\gamma$ is,

$$\frac{\partial}{\partial \gamma} (x_1, x_2, \ldots, x_D) = \left( - \prod_{i=1}^{D-2} \sin(\nu_i) \sin(\gamma), \ \prod_{i=1}^{D-2} \sin(\nu_i) \cos(\gamma), \ 0, \ldots, 0 \right).$$

Next, we compute the partial derivative of the coordinates with respect to $\nu_1$. This affects all coordinates $x_1, x_2, \ldots, x_D$. Specifically:

$$\frac{\partial x_1}{\partial \nu_1} = \frac{\partial}{\partial \nu_1} \left( \prod_{i=1}^{D-2} \sin(\nu_i) \cos(\gamma) \right) = \cos(\nu_1) \prod_{i=2}^{D-2} \sin(\nu_i) \cos(\gamma),$$

$$\frac{\partial x_2}{\partial \nu_1} = \frac{\partial}{\partial \nu_1} \left( \prod_{i=1}^{D-2} \sin(\nu_i) \sin(\gamma) \right) = \cos(\nu_1) \prod_{i=2}^{D-2} \sin(\nu_i) \sin(\gamma),$$

$$\frac{\partial x_3}{\partial \nu_1} = \frac{\partial}{\partial \nu_1}\left(\prod_{i=1}^{D-3}\sin(\nu_i)\cos(\nu_{D-2})\right) = \cos(\nu_1)\prod_{i=2}^{D-3}\sin(\nu_i)\cos(\nu_{D-2}),$$

$$\frac{\partial x_4}{\partial \nu_1} = \cdots = \frac{\partial x_D}{\partial \nu_1} = -\sin(\nu_1).$$

Thus, the complete partial derivative with respect to $\nu_1$ is:

$$\frac{\partial}{\partial \nu_1}(x_1, x_2, \ldots, x_D) = \left(\cos(\nu_1)\prod_{i=2}^{D-2}\sin(\nu_i)\cos(\gamma),\ \cos(\nu_1)\prod_{i=2}^{D-2}\sin(\nu_i)\sin(\gamma),\ -\sin(\nu_1),\ 0,\ldots,0\right).$$

**Dot Product of Tangent Vectors:** To prove orthogonality of the subspaces spanned by these vectors, we compute the dot product of the tangent vectors $\frac{\partial}{\partial \gamma}$ and $\frac{\partial}{\partial \nu_1}$. The dot product is given by,

$$\frac{\partial}{\partial \gamma}\cdot\frac{\partial}{\partial \nu_1} = \left(-\prod_{i=1}^{D-2}\sin(\nu_i)\sin(\gamma)\right)\cdot\left(\cos(\nu_1)\prod_{i=2}^{D-2}\sin(\nu_i)\cos(\gamma)\right) + \ldots,$$

which simplifies to zero, as the terms corresponding to the components in $x_1$, $x_2$, and $x_3$ do not align. Consequently, we have,

$$\frac{\partial}{\partial \gamma}\cdot\frac{\partial}{\partial \nu_1} = 0.$$

Since the dot product of the tangent vectors is zero, the subspaces spanned by fixing $A$ and fixing $B$ are orthogonal at every point on $S^{D-1}$. This orthogonality arises from the fact that the angular coordinates for latitude and longitude parameterize independent directions in the tangent space of the spherical manifold. Thus, we conclude that the subspaces resulting from fixing $A$ and $B$ are mutually orthogonal.

C.7 PROOF OF LEMMA 4.5

**Pure Strategy of the Leader:** *Given a spherical manifold, $\boldsymbol{\Phi}$, and isoplanar subspace, $\overleftrightarrow{\boldsymbol{\Phi}}_{\mathbf{a}}$ and $\overleftrightarrow{\boldsymbol{\Phi}}_{\mathbf{b}}$ for the longitudinal and lattitudinal subspaces respectively, the optimal strategy of the leader is that of a pure strategy, that is, $\pi_A^*(\mathbf{a}) \in \{0, 1\}$.*

*Proof.* Let $\mathcal{S}^{D-1} \subset \mathbb{R}^D$ be the unit sphere embedded in $D$-dimensional Euclidean space. Consider two distinct points $\theta'_A$ and $\theta'_B$ on the manifold, each with spherical coordinates $(\gamma_1^{(A)}, \gamma_2^{(A)}, \ldots, \gamma_{D-2}^{(A)}, \nu^{(A)})$ and $(\gamma_1^{(B)}, \gamma_2^{(B)}, \ldots, \gamma_{D-2}^{(B)}, \nu^{(B)})$, respectively. We aim to demonstrate that the isoplanes formed by fixing half of the spherical coordinates at $\theta'_A$ and $\theta'_B$ must intersect, and this intersection $\Psi_\theta$ is a singleton. By Lemma 4.3 we infer that $\overleftrightarrow{\boldsymbol{\Phi}}_{\mathbf{a}}$ and $\overleftrightarrow{\boldsymbol{\Phi}}_{\mathbf{b}}$ must form a non-empty intersect in $\boldsymbol{\Phi}$. Follower by Lemma 4.4, $\overleftrightarrow{\boldsymbol{\Phi}}_{\mathbf{a}}$ and $\overleftrightarrow{\boldsymbol{\Phi}}_{\mathbf{b}}$ are orthognal to eachother in $\boldsymbol{\Phi}$.

**Singleton Intersection due to Orthogonality:** Consider the isoplanes formed by fixing the angular coordinates $\theta'_A$ (latitude-like) and $\theta'_B$ (longitude-like) on the unit sphere $S^{D-1}$. These isoplanes correspond to submanifolds of the sphere, which are defined by holding certain angular coordinates constant while allowing others to vary. In the special case where the isoplanes at $\theta'_A$ and $\theta'_B$ are orthogonal, we argue that the intersection set of these submanifolds is reduced to a single element (singleton). Let $\mathbf{P}$ be the point where the isoplanes associated with fixed $\theta'_A$ and $\theta'_B$ intersect. The tangent space at $\mathbf{P}$, denoted as $T_{\mathbf{P}}S^{D-1}$, consists of vectors tangent to the sphere at $\mathbf{P}$.

The isoplane formed by fixing $\theta'_A$ corresponds to a submanifold $\overleftrightarrow{\boldsymbol{\Phi}}_{\mathbf{a}}$ whose tangent space at $\mathbf{p}$, denoted $T_{\mathbf{p}}\overleftrightarrow{\boldsymbol{\Phi}}_{\mathbf{a}}$, is spanned by the partial derivatives with respect to the longitude-like angular coordinates $\gamma_i$. Similarly, the isoplane formed by fixing $\theta'_B$ corresponds to a submanifold $\overleftrightarrow{\boldsymbol{\Phi}}_{\mathbf{b}}$, and the tangent space $T_{\mathbf{p}}\overleftrightarrow{\boldsymbol{\Phi}}_{\mathbf{b}}$ is spanned by the partial derivatives with respect to the latitude-like angular

coordinates $\nu_j$. Orthogonality between the isoplanes at $\theta'_A$ and $\theta'_B$ implies that the tangent spaces $T_{\mathbf{p}} \overleftrightarrow{\mathbf{\Phi}}_{\mathbf{a}}$ and $T_{\mathbf{p}} \overleftrightarrow{\mathbf{\Phi}}_{\mathbf{b}}$ are mutually orthogonal. This means that the dot product of any vector from $T_{\mathbf{p}} \overleftrightarrow{\mathbf{\Phi}}_{\mathbf{a}}$ with any vector from $T_{\mathbf{p}} \overleftrightarrow{\mathbf{\Phi}}_{\mathbf{b}}$ is zero:

$$\mathbf{v}_A \cdot \mathbf{v}_B = 0, \quad \forall \mathbf{v}_A \in T_{\mathbf{p}} \overleftrightarrow{\mathbf{\Phi}}_{\mathbf{a}}, \quad \mathbf{v}_B \in T_{\mathbf{p}} \overleftrightarrow{\mathbf{\Phi}}_{\mathbf{b}}.$$

Geometrically, this implies that the submanifolds $\overleftrightarrow{\mathbf{\Phi}}_{\mathbf{a}}$ and $\overleftrightarrow{\mathbf{\Phi}}_{\mathbf{b}}$ intersect at a right angle at $\mathbf{P}$. Since the submanifolds are orthogonal, no other points of intersection can occur, and the intersection set is reduced to the single point $\mathbf{P}$. Therfore,

$$|\Psi_\theta| = 1. \tag{C.11}$$

**Minimal Geodesic Distance from $\Psi_\theta$:** Let $\Psi_\gamma = (x_1^{(\text{int})}, x_2^{(\text{int})}, \ldots, x_D^{(\text{int})})$ be the unique intersection point of the two isoplaness. Now, we consider the geodesic distance from this intersection point to any other point on the sphere. The geodesic distance between two points $\mathbf{P}_1 = (x_1^{(1)}, x_2^{(1)}, \ldots, x_D^{(1)})$ and $\mathbf{P}_2 = (x_1^{(2)}, x_2^{(2)}, \ldots, x_D^{(2)})$ on the unit sphere is given by,

$$\mathfrak{G}(\mathbf{P}_1, \mathbf{P}_2) = \arccos(\mathbf{P}_1 \cdot \mathbf{P}_2).$$

At the intersection point $\Psi_\theta$, the geodesic distance is minimized, thus,

$$\mathbf{P}_1 = \Psi_\theta \implies \mathfrak{G}(\mathbf{P}_1, \Psi_\theta) = 0.$$

Suppose we move away from $\Psi_\gamma$ along either the longitude isoplanes (by changing $x_1$) or the latitude isoplanes (by changing $x_2, x_3, \ldots, x_D$). Any such deviation implies a change in the dot product $\mathbf{P}_1 \cdot \mathbf{P}_2$, which results in an increase in the geodesic distance. Specifically, if we move along the longitude isoplanes, we are changing $x_1$, while the other coordinates remain constant, resulting in a decrease in the dot product. Similarly, if we move along the latitude isoplanes, we are changing $x_2, x_3, \ldots, x_D$, again causing a decrease in the dot product. Since the geodesic distance is a monotonically increasing function of the angular separation, any deviation from $\Psi_\gamma$ leads to an increase in the geodesic distance,

$$\mathfrak{G}(\mathbf{P}_2, \mathbf{P}_1) > \mathfrak{G}(\mathbf{P}_1, \Psi_\gamma) = 0.$$

Thus, any deviation from the intersection point of the longitude and latitude isoplaness must result in an increase in the geodesic distance, $\mathfrak{G}(\cdot)$. By Lemma 4.1, this increase in the geodesic distance will decrease the expected reward $\mu_A$. As the cardinality of $\Psi$ is $|\Psi_\gamma| = 1$ from Eq. (C.11), this implies no optimal mixed strategies exist for the leader, and thus, $\pi_A^*(\mathbf{a}) \in \{0, 1\}$.

$\square$

## C.8 Conversion of Cartesian Uncertainty to Spherical

**Lemma C.2.** *Given two points $\theta_A, \tilde{\theta}_A \in \mathbb{R}^D$, denoting points on the surface of a unit spherical manifold, the uncertainty in Cartesian coordinates expressed as $\|\theta_A - \tilde{\theta}_A\| < \mathcal{C}_\theta(t)$ can be expressed as uncertainty in geodesic distance as $\mathfrak{G}(A, \tilde{\theta}_A) < \cos^{-1}\left(1 - \frac{\mathcal{C}_\theta(t)^2}{2}\right).$*

*Proof.* Given two points $\theta_A, \tilde{\theta}_A \in \mathbb{R}^D$, with $\|A\| = \|\tilde{\theta}_A\| = 1$, denoting points on the surface of a unit sphere, the uncertainty in Cartesian coordinates is expressed as:

$$\|\theta_A - \tilde{\theta}_A\| < \mathcal{C}_\theta(t)$$

where $\mathcal{C}_\theta(t) \in \mathbb{R}^+$ is the uncertainty bound. We aim to translate this uncertainty into spherical coordinates.

**Cartesian Coordinates on the Unit Sphere:**  In $\mathbb{R}^D$, the spherical coordinates of a point $\theta_A$ on the surface of the unit sphere can be represented as:

$$\theta_A^{(1)} = \cos(\nu_1),$$

$$\theta_A^{(2)} = \sin(\nu_1)\cos(\nu_2),$$

$$\theta_A^{(3)} = \sin(\nu_1)\sin(\nu_2)\cos(\nu_3),$$

$$\vdots$$

$$\theta_A^{(D-1)} = \sin(\nu_1)\sin(\nu_2)\ldots\sin(\nu_{D-2})\cos(\gamma),$$

$$\theta_A^{(D)} = \sin(\nu_1)\sin(\nu_2)\ldots\sin(\nu_{D-2})\sin(\gamma),$$

where $\nu_1, \nu_2, \ldots, \nu_{D-2}$ represent the latitude angles, and $\gamma$ represents the longitude angle. Similarly, the point $\tilde{\theta}_A$ can be written in terms of spherical angles $\nu_1', \nu_2', \ldots, \gamma'$.

**Uncertainty in Cartesian Coordinates:**  The uncertainty in Cartesian space is given by:

$$\|\theta_A - \tilde{\theta}_A\|^2 = (\theta_A^{(1)} - \tilde{\theta}_A^{(1)})^2 + (\theta_A^{(2)} - \tilde{\theta}_A^{(2)})^2 + \ldots + (\theta_A^{(D)} - \tilde{\theta}_A^{(D)})^2 < \mathcal{C}_\theta(t)^2.$$

However, it is more efficient to relate this uncertainty directly to spherical angular distance.

**Spherical Angular Distance:**  The squared Euclidean distance between two points $\theta_A$ and $\tilde{\theta}_A$ on the surface of the unit sphere is related to their angular distance $\nu$ by the spherical law of cosines:

$$\|\theta_A - \tilde{\theta}_A\|^2 = 2(1 - \cos(\nu)),$$

where $\nu$ is the angular distance between the two points, and $\cos(\nu)$ is given by:

$$\cos(\nu) = \cos(\nu_1)\cos(\nu_1') + \sin(\nu_1)\sin(\nu_1')\Big(\cos(\nu_2)\cos(\nu_2') + \sin(\nu_2)\sin(\nu_2')\cdots\Big).$$

This expression provides the exact angular distance between points $\theta_A$ and $\tilde{\theta}_A$ on the unit sphere.

**Uncertainty in Spherical Coordinates:**  The inequality $\|\theta_A - \tilde{\theta}_A\| < \mathcal{C}_\theta(t)$ implies that the angular distance $\nu$ between the two points satisfies:

$$2(1 - \cos(\nu)) < \mathcal{C}_\theta(t)^2,$$

which simplifies to:

$$\cos(\nu) > 1 - \frac{\mathcal{C}_\theta(t)^2}{2}.$$

Since $\cos(\nu)$ ranges from 1 (when $\theta_A = \tilde{\theta}_A$) to -1 (for antipodal points), the angular distance $\nu$ is bounded by:

$$\nu < \cos^{-1}\left(1 - \frac{\mathcal{C}_\theta(t)^2}{2}\right).$$

This inequality describes the exact spherical uncertainty region. Thus, the uncertainty $\|\theta_A - \tilde{\theta}_A\| < \mathcal{C}_\theta(t)$ in Cartesian space corresponds to an angular uncertainty $\nu < \cos^{-1}\left(1 - \frac{\mathcal{C}_\theta(t)^2}{2}\right)$ on the unit sphere. $\qquad\square$

## C.9  Distance Preserving Orthogonal Projection:

**Lemma C.3.** *Consider a unit sphere $S^{D-1} \subset \mathbb{R}^D$. Given a point $\theta_A \in S^{D-1}$ and a geodesic ball $B_J \subset S^{D-1}$ centered at $\theta_A$, we are interested in the behaviour of this ball under orthogonal projection onto a subspace of $\mathbb{R}^D$. Specifically, we aim to rigorously show that the diameter of the orthogonally projected ball does not exceed the diameter of the original geodesic ball.*

*Proof.* **Geodesic Uncertainty Balls:** Let $\theta_A, \tilde{\theta}_A \in \mathbb{R}^D$ be two points on the unit sphere, i.e., $\|\theta_A\| = \|\tilde{\theta}_A\| = 1$, and let the geodesic distance between $\theta_A$ and $\tilde{\theta}_A$ be denoted by $\gamma(\theta_A, \tilde{\theta}_A)$. The geodesic distance between any two points on $S^{D-1}$ is given by,

$$\gamma(\theta_A, \tilde{\theta}_A) = \arccos\left(\theta_A \cdot \tilde{\theta}_A\right),$$

where $\theta_A \cdot \tilde{\theta}_A$ is the Euclidean dot product between $\theta_A$ and $\tilde{\theta}_A$. A geodesic ball $B_J(\theta_A)$ centered at $\theta_A$ with radius $J$ is defined as the set of points on the unit sphere such that their geodesic distance from $\theta_A$ is less than or equal to $J$:

$$B_J(\theta_A) = \{\theta'_A \in S^{D-1} \mid \gamma(\theta_A, \tilde{\theta}_A) \leq J\}.$$

We are particularly interested in the case where $J \leq \arccos\left(1 - \frac{\mathcal{C}_\theta(t)^2}{2}\right)$, where $\mathcal{C}_\theta(t)$ is a positive value corresponding to the uncertainty radius in the Euclidean distance.

**Orthogonal Projection and Geodesic Distance:** Given a subspace $V \subset \mathbb{R}^D$, let $P_V : \mathbb{R}^D \to V$ denote the orthogonal projection onto $V$. For any points $\theta_A, \tilde{\theta}_A \in \mathbb{R}^D$, the Euclidean distance between their projections is bounded by:

$$\|P_V(\theta_A) - P_V(\tilde{\theta}_A)\| \leq \|\theta_A - \tilde{\theta}_A\|.$$

Since the geodesic distance on the unit sphere is a measure of arc length between points, it follows that the geodesic distance between two points is non-increasing under orthogonal projection. We aim to show that the diameter of the projected geodesic ball onto the subspace $V$ does not exceed the diameter of the original ball.

**Diameter of a Geodesic Ball:** The diameter of a set $S \subset S^{D-1}$ is defined as the greatest geodesic distance between any two points in $S$:

$$\text{diam}(S) = \sup_{x,y \in S} \gamma(x, y).$$

For a geodesic ball $B_J(\theta_A)$, the maximum geodesic distance occurs between two antipodal points on the boundary of the ball. Therefore, the diameter of the geodesic ball is:

$$\text{diam}(B_J(\theta_A)) = 2J.$$

In particular, for $J = \arccos\left(1 - \frac{\mathcal{C}_\theta(t)^2}{2}\right)$, we have:

$$\text{diam}(B_J(\theta_A)) = 2\arccos\left(1 - \frac{\mathcal{C}_\theta(t)^2}{2}\right).$$

$\square$

### C.10 DIAMETER PRESERVING ORTHOGONAL PROJECTION

We now formalize the behaviour of the geodesic ball under orthogonal projection.

**Lemma C.4.** *Let $B_J(\theta_A)$ be a geodesic ball of radius $J \leq \arccos\left(1 - \frac{\mathcal{C}_\theta(t)^2}{2}\right)$ on the unit sphere $S^{D-1} \subset \mathbb{R}^D$. Let $V \subset \mathbb{R}^D$ be a subspace, and let $P_V : \mathbb{R}^D \to V$ be the orthogonal projection onto $V$. Then, the diameter of the orthogonally projected ball $P_V(B_J(\theta_A))$ satisfies:*

$$diam(P_V(B_J(\theta_A))) \leq diam(B_J(\theta_A)) = 2J.$$

*Proof.* Consider two points $\theta_A, \tilde{\theta}_A \in B_J(\theta_A)$. By the definition of a geodesic ball, we know that:

$$\gamma(\theta_A, \tilde{\theta}_A) \leq 2J.$$

Next, project $\theta_A$ and $\tilde{\theta}_A$ orthogonally onto the subspace $V$, yielding the points $P_V(\theta_A)$ and $P_V(\tilde{\theta}_A)$. Since orthogonal projection reduces or preserves Euclidean distances, we have:

$$\|P_V(\theta_A) - P_V(\tilde{\theta}_A)\| \leq \|\theta_A - \tilde{\theta}_A\|.$$

Moreover, since the geodesic distance between points on the sphere is a function of their Euclidean distance, it follows that the geodesic distance between the projected points $P_V(\theta_A)$ and $P_V(\tilde{\theta}_A)$ is also bounded by:

$$\gamma(P_V(\theta_A), P_V(\tilde{\theta}_A)) \leq \gamma(\theta_A, \tilde{\theta}_A).$$

Thus, for all pairs $\theta_A, \tilde{\theta}_A \in B_J(\theta_A)$, we have:

$$\gamma(P_V(\theta_A), P_V(\tilde{\theta}_A)) \leq 2J.$$

This shows that the diameter of the projected geodesic ball $P_V(B_J(\theta_A))$ is at most $2J$, i.e.,

$$\text{diam}(P_V(B_J(\theta_A))) \leq \text{diam}(B_J(\theta_A)) = 2J.$$

$\square$

### C.11 Proof of Theorem 1

**Isoplane Stackelberg Regret:** For D-dimensional spherical manifolds embedded in $\mathbb{R}^D$ space, where $\phi(\mathbf{a}, \cdot)$ generates an isoplanes $\overleftrightarrow{\mathbf{\Phi}}_{\mathbf{a}}$, and the linear relationship to the reward function in Eq. (2.15) and Eq. (2.16) holds, the simple regret, defined in Eq. (4.3), of any learning algorithm with uncertainty parameter uncertainty $\mathcal{C}_\theta(t)$, refer to in Eq. (3.3), is bounded by $\mathcal{O}(2\arccos(1 - \mathcal{C}_\theta(t)^2/2))$.

*Proof.* The proof of Theorem 1 hinges on the aforementioned arguments in Lemma C.2, Lemma C.3, and Lemma C.4 sequentially, but in the context of parameter estimation.

First, Lemma C.2 argues that one can transform a confidence bound $|\theta_A - \hat{\theta}_A| \leq \mathcal{C}_\theta(t)$ into a confidence bound on geodesic distance $\mathfrak{G}(\theta_A, \hat{\theta}_A) \leq \cos^{-1}\left(1 - \frac{\mathcal{C}_\theta(t)^2}{2}\right)$. Let us denote this as the geodesic confidence ball $\texttt{Ball}_{\mathfrak{G}}(\theta^*, \mathcal{C}_\theta(t))$. Nevertheless, due to the separation of subspaces $\overleftrightarrow{\mathbf{\Phi}}_{\mathbf{a}}$ and $\overleftrightarrow{\mathbf{\Phi}}_{\mathbf{b}}$, we must find the projection of $\texttt{Ball}_{\mathfrak{G}}(\theta^*, \mathcal{C}_\theta(t))$ onto $\overleftrightarrow{\mathbf{\Phi}}_{\mathbf{b}}$ such that we can obtain a diameter measure on the new intersecting subspace $\overleftrightarrow{\mathbf{\Phi}}_{\mathbf{a}} \cap \overleftrightarrow{\mathbf{\Phi}}_{\mathbf{b}}$. Next, Lemma C.3 argues that geodesic distances will either be preserved or reduced when making a projection to an orthogonal subspace $\overleftrightarrow{\mathbf{\Phi}}_{\mathbf{b}}$, the orthogonality of this subspace was previously established in Lemma 4.4. Thereafter, Lemma C.4 specifies that the maximum diameter of this new confidence ball $\texttt{Ball}'_{\mathfrak{G}}(\theta^*, \mathcal{C}_\theta(t))$ that is projected onto $\overleftrightarrow{\mathbf{\Phi}}_{\mathbf{b}}$ is confined to a maximum diameter of $2\cos^{-1}\left(1 - \frac{\mathcal{C}_\theta(t)^2}{2}\right)$.

Thus, this constitutes the best and worst possible outcomes due to misspecification in accordance with the formulation in Eq. (3.6) and Eq. (3.7), denoted as $\bar{\mathcal{H}}(\theta_A^*, t) - \underline{\mathcal{H}}(\theta_A^*, t)$, also expressed in Eq. (4.3), which upper bounds the simple regret.

$\square$

# D    Neural Flow Architectural Specifications

We present the mathematical foundations of the normalizing flow architecture used to model spherical mappings. Our method combines a spherical coordinate transformation with normalizing flows to provide an invertible mapping between input features and a latent space, with applications to tasks requiring smooth transformations on a manifold.

**Mapping to a Spherical Manifold:** The transformation from Cartesian coordinates to spherical coordinates is used to map input features onto an $D$-dimensional spherical manifold. We define two heads in the neural network input, the head from A specifically controls the azimuthal spherical coordinate and additional coordinates, and the head from B specifically controls other coordinates. The output sizes of the neural network that transforms the inputs are $\lfloor \frac{D-1}{2} \rfloor + 1$ for A and $\lfloor \frac{D-1}{2} \rfloor$ for B. The conversion from spherical coordinates to Cartesian coordinates, $\mathbf{x} \in \mathbb{R}^D$, is defined in Appendix F.1.

**Affine Coupling Layers:** A normalizing flow consists of a series of invertible transformations, including affine coupling layers, which divide the input into two parts and transform one part conditioned on the other. Let the input be $\mathbf{x} = [\mathbf{x}_1, \mathbf{x}_2]$, where $\mathbf{x}_1$ and $\mathbf{x}_2$ are disjoint subsets of the input. The affine coupling transformation is defined as,

$$\mathbf{y}_1 = \mathbf{x}_1, \tag{D.1}$$

$$\mathbf{y}_2 = \mathbf{x}_2 \odot \exp(s(\mathbf{x}_1)) + t(\mathbf{x}_1), \tag{D.2}$$

where $\odot$ denotes element-wise multiplication, and $s(\mathbf{x}_1)$ and $t(\mathbf{x}_1)$ are the scaling and translation functions, respectively, parameterized by a neural network. The inverse of this transformation is straightforward:

$$\mathbf{x}_1 = \mathbf{y}_1, \tag{D.3}$$

$$\mathbf{x}_2 = (\mathbf{y}_2 - t(\mathbf{y}_1)) \odot \exp(-s(\mathbf{y}_1)). \tag{D.4}$$

This transformation is invertible by design, making it suitable for use in flow-based models.

**Log Determinant of the Jacobian:** The log-likelihood calculation requires computing the log determinant of the Jacobian matrix for the transformation. For the affine coupling layer, the Jacobian matrix is triangular, and the log determinant is simply the sum of the scaling terms:

$$\log \left| \det \frac{\partial \mathbf{y}}{\partial \mathbf{x}} \right| = \sum_i s(\mathbf{x}_1). \tag{D.5}$$

This term contributes to the overall log probability during training.

**Normalizing Flow Forward Transform:** A normalizing flow is constructed by stacking several affine coupling layers and random permutation layers. Let $\mathbf{x} \in \mathbb{R}^d$ be the input, and $\mathbf{z} \in \mathbb{R}^d$ be the transformed latent variable after $L$ layers of flow. Each layer applies a transformation $f_l$ such that:

$$\mathbf{z}^{(l+1)} = f_l(\mathbf{z}^{(l)}), \tag{D.6}$$

where $f_l$ represents either an affine coupling transformation or a random permutation. After $L$ layers, the final output is denoted as $\mathbf{z} = \mathbf{z}^{(L)}$. The forward transformation can thus be written as:

$$\mathbf{z}, \log \det J = f_{\text{flow}}(\mathbf{x}), \tag{D.7}$$

where $\log \det J$ is the log determinant of the Jacobian matrix for the entire flow.

To compute the log-likelihood of the input $\mathbf{x}$, we map it to the latent space $\mathbf{z}$ under the flow transformation. The probability of $\mathbf{x}$ is computed as:

$$p(\mathbf{x}) = p(\mathbf{z}) \left| \det \frac{\partial \mathbf{z}}{\partial \mathbf{x}} \right|, \tag{D.8}$$

where $p(\mathbf{z})$ is the probability of $\mathbf{z}$ under the base distribution (typically a standard normal distribution):

$$p(\mathbf{z}) = \mathcal{N}(\mathbf{z}; 0, I). \tag{D.9}$$

The log probability is then given by:

$$\log p(\mathbf{x}) = \log p(\mathbf{z}) + \log \left| \det \frac{\partial \mathbf{z}}{\partial \mathbf{x}} \right|. \tag{D.10}$$

**Inverse Transform:** The invertibility of the flow allows for both density estimation and sampling. To sample from the model, we draw samples $\mathbf{z} \sim \mathcal{N}(0, I)$ from the base distribution and apply the inverse transformation:

$$\mathbf{x} = f_{\text{flow}}^{-1}(\mathbf{z}). \tag{D.11}$$

Each affine coupling layer and random permutation is applied in reverse order to recover the original inputs.

**Random Permutation Layer:** The random permutation layer permutes the features of the input vector to ensure that different parts of the input are transformed at each layer. Let $\mathbf{x} \in \mathbb{R}^d$ be the input, and let $P$ be a permutation matrix. The permutation transformation is defined as:

$$\mathbf{x}' = P\mathbf{x}. \tag{D.12}$$

Since permutation matrices are orthogonal, the Jacobian determinant of this transformation is always 1, and it does not contribute to the log determinant calculation.

| Layer | Description | Output Size |
|---|---|---|
| Input Head A | Input head A | $N_B \times |\mathcal{A}|$ |
| Input Head B | Input head B | $N_B \times |\mathcal{B}|$ |
| Input Features | Input features | $N_B \times D$ |
| Affine Coupling Layer | No. of Affine Coupling layers | $N_B \times 64$ |
| fc_A1 Hidden Dim. | Number of hidden dimensions in first fully connected layer A. | $B \times 1024$ |
| fc_B1 Hidden Dim. | Number of hidden dimensions in first fully connected layer B. | $B \times 1024$ |
| Hidden Dim. | No.of hidden layers for A and B. | $N_B \times 16$ |
| fc_A1 Final Layer Dim. | Number of hidden dimensions in final layer A | $N_B \times \left( \lfloor \frac{D-1}{2} \rfloor + 1 \right)$ |
| fc_B1 Final Layer Dim. | Number of hidden dimensions in final layer B | $N_B \times \left( \lfloor \frac{D-1}{2} \rfloor \right)$ |
| Output | Output features after flow transformation | $N_B \times D$ |

Table 2: Normalizing Flows Neural Architecture Specifications.

**Overview:** In summary, the normalizing flow architecture combines spherical mapping, affine coupling transformations, and random permutations to form a powerful framework for invertible transformations. The model leverages the flexibility of normalizing flows to map inputs to a spherical manifold, enabling efficient density estimation and sampling from a base Gaussian distribution.

| Parameter | Value |
|---|---|
| $N_B$ Batch Size | 2048 |
| $\alpha_N$ (Negative Log Liklihood Loss Coef.) | 0.5 |
| $\alpha_R$ (Repulsion Loss Coef.) | 1.0 |
| $\alpha_P$ (Perturb. Loss Coef.) | 0.5 |
| $\alpha_L$ (Lipschitz Loss Coef.) | 1.5 |
| No. Epochs | 20,000 |
| $\alpha_{LR}$ (Learning Rate) | 0.05 |
| $C_L$ (Lipschitz Constant) | 0.5 |

Table 3: Hyper parameters used for normalizing neural flow network training.

# E    VISUALIZATIONS

## E.1    COMPUTATIONAL RESULTS OF ISOPLANE BEHAVIOUR

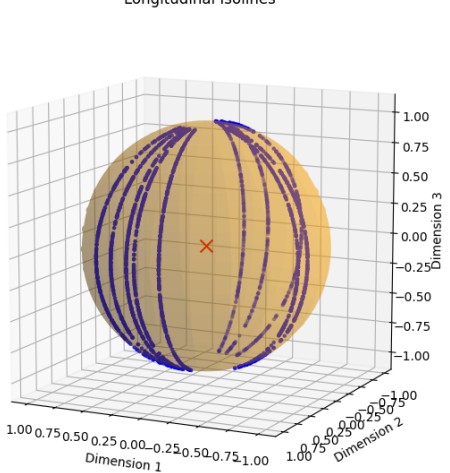
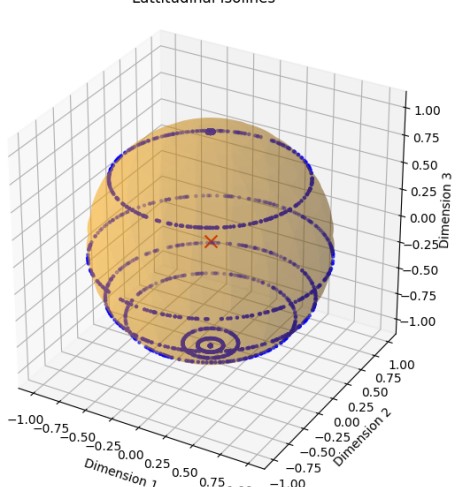

**Longitudinal Isolines:**    Visualization of longitudinal isolines generated by the normalizing neural flow network.

**Latitudinal Isoplanes:** Visualization of lattitudinal isolines generated by the normalizing neural flow network.

Figure 4: Formation of isolines (or isoplanes in higher dimensions) forming on the spherical manifold $\mathbf{\Phi}$ as we fix $\mathbf{a}$ and vary $\mathbf{b}$ (longitudinal), and fix $\mathbf{b}$ and vary $\mathbf{a}$ (lattitudinal).

# F   ALGORITHMS

## F.1   MAPPING BETWEEN SPHERICAL AND CARTESIAN COORDINATES

---

**Algorithm 2** Spherical to Cartesian Conversion in $n$-Dimensions

---

1: **function** SPHERICAL_TO_CARTESIAN($r, \nu$)
2:     **Input:** $r$ (radius), $\nu$ (Spherical coordinates $D - 1$ dimensions.)
3:     **Output:** Cartesian coordinates $p = [x_1, x_2, \ldots, x_D]$
4:     $x_1 \leftarrow r \cdot \cos(\nu_1)$
5:     **for** $i = 2$ to $D - 1$ **do**
6:         $x_i \leftarrow r \cdot \sin(\nu_1) \cdot \sin(\nu_2) \cdot \cdots \cdot \sin(\nu_{i-1}) \cdot \cos(\nu_i)$
7:     **end for**
8:     $x_n \leftarrow r \cdot \sin(\nu_1) \cdot \cdots \cdot \sin(\nu_{D-1})$
9:     **return** $[x_1, x_2, \ldots, x_D]$
10: **end function**

---

**Algorithm 3** Cartesian to Spherical Conversion in $n$-Dimensions

---

1: **function** CARTESIAN_TO_SPHERICAL($p$)
2:     **Input:** Cartesian coordinates $p = [x_1, x_2, \ldots, x_D]$
3:     **Output:** $r$ (radius), $\nu = [\nu_1, \nu_2, \ldots, \nu_{D-1}]$ (Spherical coordinates $D - 1$ dimensions.)
4:     $r \leftarrow \sqrt{x_1^2 + x_2^2 + \cdots + x_D^2}$                                    ▷ Compute the radius
5:     $\nu_1 \leftarrow \arccos\left(\frac{x_1}{r}\right)$                                    ▷ First spherical angle
6:     **for** $i = 2$ to $n - 1$ **do**
7:         $\nu_i \leftarrow \arctan 2\left(\sqrt{x_1^2 + x_2^2 + \cdots + x_i^2}, x_{i+1}\right)$     ▷ Spherical angles for $i = 2$ to $D - 1$
8:     **end for**
9:     **return** $r, \nu = [\nu_1, \nu_2, \ldots, \nu_{D-1}]$
10: **end function**

---

# G EXPERIMENTAL RESULTS

## G.1 $\mathbb{R}^1$ STACKELBERG GAME

**Problem Setup:** We consider a Stackelberg game with a leader $\theta_A$ and a follower $B$, both operating in continuous action spaces $a, b \in \mathbb{R}^1$. The leader chooses an action $\theta_A$, and the follower responds by choosing an action $b$ based on the leader's decision. The reward functions for both players are linear in structure but include nonlinear components to model real-world constraints and interactions.

**Leader's Reward Function:** The leader's reward function $\mu_A(a, b)$ is defined as follows:

$$\mu_A(a, b) = \theta_1 a + \theta_2 \log(1 + b^2) - \frac{\theta_3}{2} a^2 + \epsilon, \quad \epsilon \in \mathcal{N}(0, \sigma) \tag{G.1}$$

where,

- $\theta_1, \theta_2 > 0$ are weight parameters that control the trade-off between the leader's direct action $\theta_A$ and the follower's response $b$.

- $\log(1 + b^2)$ introduces nonlinearity with respect to the follower's action $b$.

- $-\frac{\theta_3}{2} a^2$ is a quadratic penalty on large leader actions to avoid extreme behaviour by the leader.

**Follower's Reward Function:** The follower's reward function $\mu_B(\mathbf{a}, \mathbf{b})$ is given by:

$$\mu_B(\mathbf{a}, \mathbf{b}) = \alpha_1(-b^2) + \alpha_2 ab + \epsilon, \quad \epsilon \in \mathcal{N}(0, \sigma) \tag{G.2}$$

where,

- $\alpha_1, \alpha_2 > 0$ are parameters that determine the influence of the follower's own action $b$ and the leader's action $\theta_A$ on the follower's reward.

- $-b^2$ represents a concave cost function for the follower, preferring smaller values of $b$.

- $ab$ introduces an interaction term between the leader's action and the follower's action.

**Follower's Best Response:** The follower maximizes their reward function $\mu_B(\mathbf{a}, \mathbf{b})$ by choosing $b$ given $\theta_A$. To determine the follower's best response $\mathfrak{B}(a)$, we compute the first-order condition with respect to $b$:

$$\frac{\partial \mathbb{E}[\mu_B(\mathbf{a}, \mathbf{b})]}{\partial b} = -2\alpha_1 b + \alpha_2 a = 0 \tag{G.3}$$

Solving for $b$, the follower's best response is:

$$\mathfrak{B}(a) = \frac{\alpha_2 a}{2\alpha_1} \tag{G.4}$$

**Leader's Optimization Problem:** Given that the follower's best response is $\mathfrak{B}(a) = \frac{\alpha_2 a}{2\alpha_1}$, the leader maximizes their reward function $\mu_A(a, \mathfrak{B}(a))$ as,

$$\mathbb{E}[\mu_A(a, \mathfrak{B}(a))] = \theta_1 a + \theta_2 \log\left(1 + \left(\frac{\alpha_2 a}{2\alpha_1}\right)^2\right) - \frac{\theta_3}{2} a^2. \tag{G.5}$$

This results in the following optimization problem for the leader,

$$\max_a \left( \theta_1 a + \theta_2 \log \left( 1 + \frac{\alpha_2^2 a^2}{4\alpha_1^2} \right) - \frac{\theta_3}{2} a^2 \right). \tag{G.6}$$

**Non-Trivial Solution for the Leader:** To solve for the leader's optimal action $a^*$, we take the derivative of the leader's reward function with respect to $\theta_A$ and set it equal to zero,

$$\frac{d}{da} \left( \theta_1 a + \theta_2 \log \left( 1 + \frac{\alpha_2^2 a^2}{4\alpha_1^2} \right) - \frac{\theta_3}{2} a^2 \right) = 0 \tag{G.7}$$

$$\theta_1 - \theta_3 a + \theta_2 \cdot \frac{2 \cdot \left( \frac{\alpha_2 a}{2\alpha_1} \right) \cdot \left( \frac{\alpha_2}{2\alpha_1} \right)}{1 + \frac{\alpha_2^2 a^2}{4\alpha_1^2}} = 0 \tag{G.8}$$

Which simplifies to,

$$\theta_1 - \theta_3 a + \frac{\theta_2 \cdot \frac{\alpha_2^2 a}{\alpha_1^2}}{1 + \frac{\alpha_2^2 a^2}{4\alpha_1^2}} = 0. \tag{G.9}$$

This equation has no simple closed-form solution and must be solved numerically. The interplay between the nonlinear logarithmic term and the quadratic penalty introduces complexity into the leader's optimization, making the optimal value of $a^*$ non-trivial.

### G.1.1 $\mathbb{R}^2$ STACKELBERG GAME

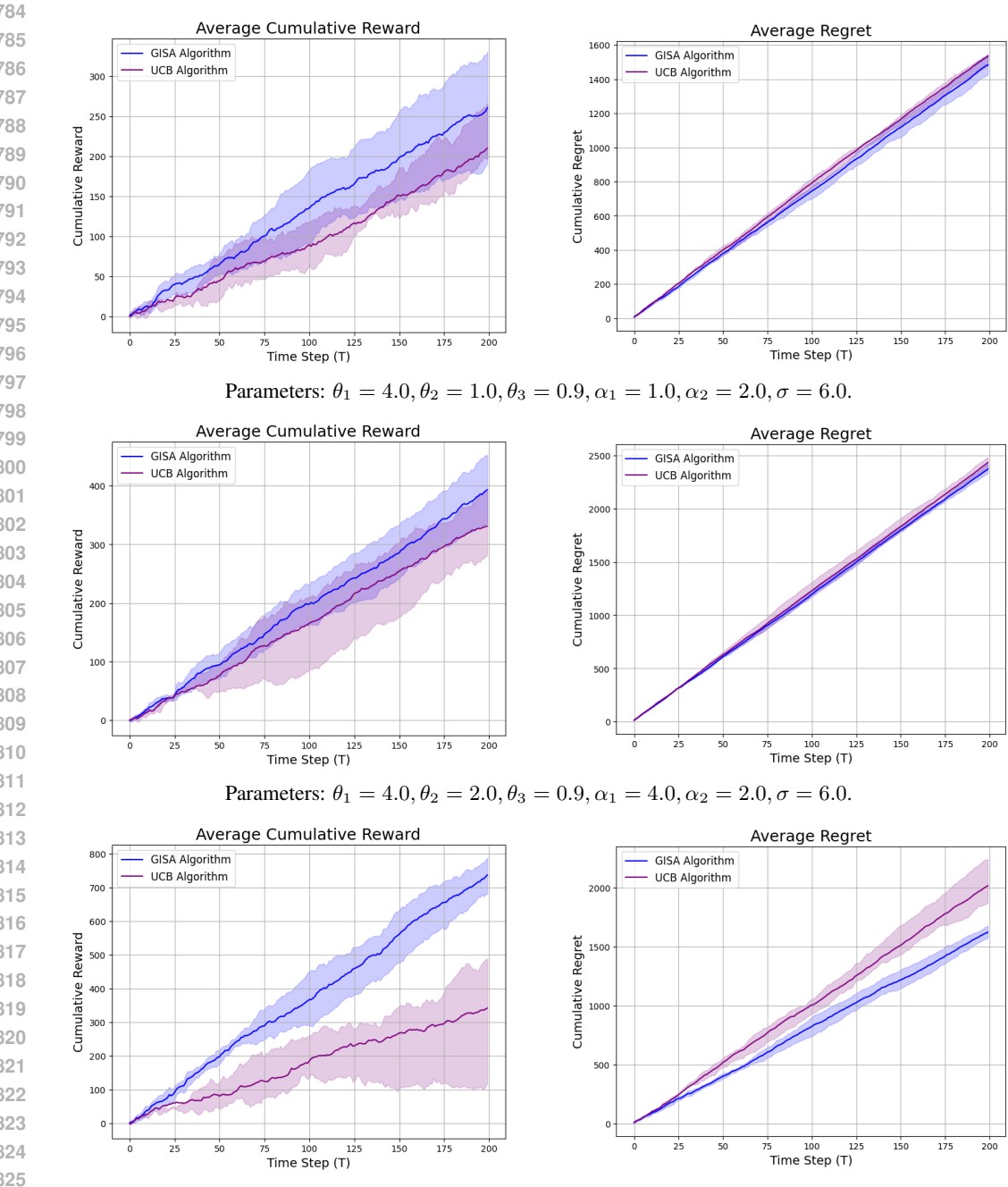

Figure 5: Mean values are calculated over 1,000 trials, with shaded regions representing confidence intervals, all of which fall within the first quartile.

## G.2 THE NEWSVENDOR PRICING GAME SPECIFCATIONS (NPG)

We model the two learning agents in a *Newsvendor pricing game*, involving a supplier $A$ and a retailer $B$. The leader, a supplier, is learning to dynamically price the product for the follower, a retailer, aiming to maximize her reward. To achieve this, the follower adheres to classical Newsvendor theory, which involves finding the optimal order quantity given a known demand distribution before the realization of the demand.

**Rules of the Newsvendor Pricing Game:** We explicitly denote $a \equiv \mathbf{a} \in \mathbb{R}^1$, and $\mathbf{b} \equiv [b, p]^\intercal \in \mathbb{R}^2$. Where $a$ denotes wholesale price from the supplier firm, $p$ and $b$ denote the retail price and order amount of the retail firm.

1. The supplier selects wholesale price $a$, and provides it to the retailer.
2. Given wholesale cost $a$, the retailer reacts with his best response $[b, p]^\intercal$, consisting of retail price $p$, and order amount $b$.
3. As the retailer determines the optimal order amount $b$, he pays $\mathcal{G}_A(a, b) = ab$ to the supplier.
4. At time $t$, nature draws demand $d^t \sim d_\rho(p)$, and it is revealed to the retailer.
5. The retailer makes a profit of $\mathcal{G}_B(a, b) = p \min\{d^t, b\} - ab$.
6. Steps 1 to 5 are repeated for $t \in 1...T$ iterations.

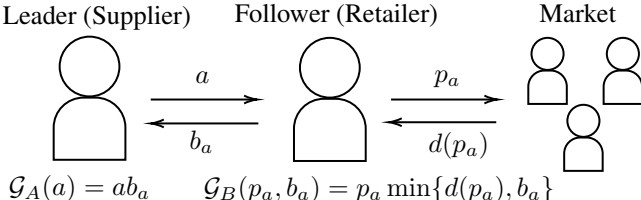

$$\mathcal{G}_A(a) = ab_a \qquad \mathcal{G}_B(p_a, b_a) = p_a \min\{d(p_a), b_a\}$$

Figure 6: **The Newsvendor Pricing Game.** From (L. Liu and Rong 2024), in this Stackelberg game, there a logistics network between a supplier (leader) and retailer (follower), where utility functions are not necessarily supermodular, the supplier issues a wholesale price $a$, and the retailer issues a purchase quantity $b$, and a retail price $p$ in response.

**Demand Function:** Stochastic demand is represented in Eq. G.11, which is governed by a linear additive demand function $\Gamma_\rho(p)$ representing the expected demand, $\mathbb{E}[d(p)]$, as a function of $p$ in Eq. G.11. The demand function is governed by parameters $\rho$.

$$\Gamma_\rho(p) = \max\{0, \rho_0 - \rho_1 p\}, \quad \rho_0 \geq 0, \ \rho_1 \geq 0 \tag{G.10}$$

$$d_\rho(p) = \Gamma_\rho(p) + \epsilon, \quad \epsilon \in \mathcal{N}(0, \sigma) \tag{G.11}$$

This problem combines the problem of the *price-setting Newsvendor* (Petruzzi and Dada 1999) (Arrow, Harris, and Marschak 1951), with that of a bilateral Stackelberg game under imperfect information. Even in the scenario of perfect information, the *price-setting Newsvendor* has no closed-form solution, therefore no exact solution to the Stackelberg equilibrium. We apply the algorithm from (L. Liu and Rong 2024) to learn a Stackelberg equilibrium under a *risk-free pricing* strategy assumption, and apply Algorithm 4 from (L. Liu and Rong 2024) as a baseline against Algorithm 1 (GISA).

---

**Algorithm 4** Learning Algorithm for Newsvendor Pricing Game from (L. Liu and Rong 2024)

---

1: **for** $t \in 1...T$ **do**:
2:     Leader and follower estimates a confidence interval $\mathcal{C}_\theta(t)$ from available data.
3:     $\mathcal{H}(\rho) = \hat{\rho}_0/\hat{\rho}_1$.
4:     Leader plays action $a$, where $a = \underset{a \in \mathcal{A}, \rho \in \mathcal{C}^t}{\operatorname{argmax}} \ a F_{\bar{\rho}_a}^{-1}\left(1 - \frac{2a}{\mathcal{H}(\rho)+a}\right)$ from Eq. (3.8) in (L. Liu and Rong 2024).
5:     Follower sets price $p = (\mathcal{H}(\rho) + a)/2$.
6:     Follower estimates their optimistic parameters $\bar{\rho}_a$, and best response $\bar{b}_a$ from from Eq. (3.4) and (3.5a) respectively in (L. Liu and Rong 2024).
7:     Leader obtains reward, $\mathcal{G}_A = ab$.
8:     Follower obtains reward, $\mathcal{G}_B = p \min\{b, d(p)\}$.
9: **end for**

---

### G.2.1 NPG RESULTS

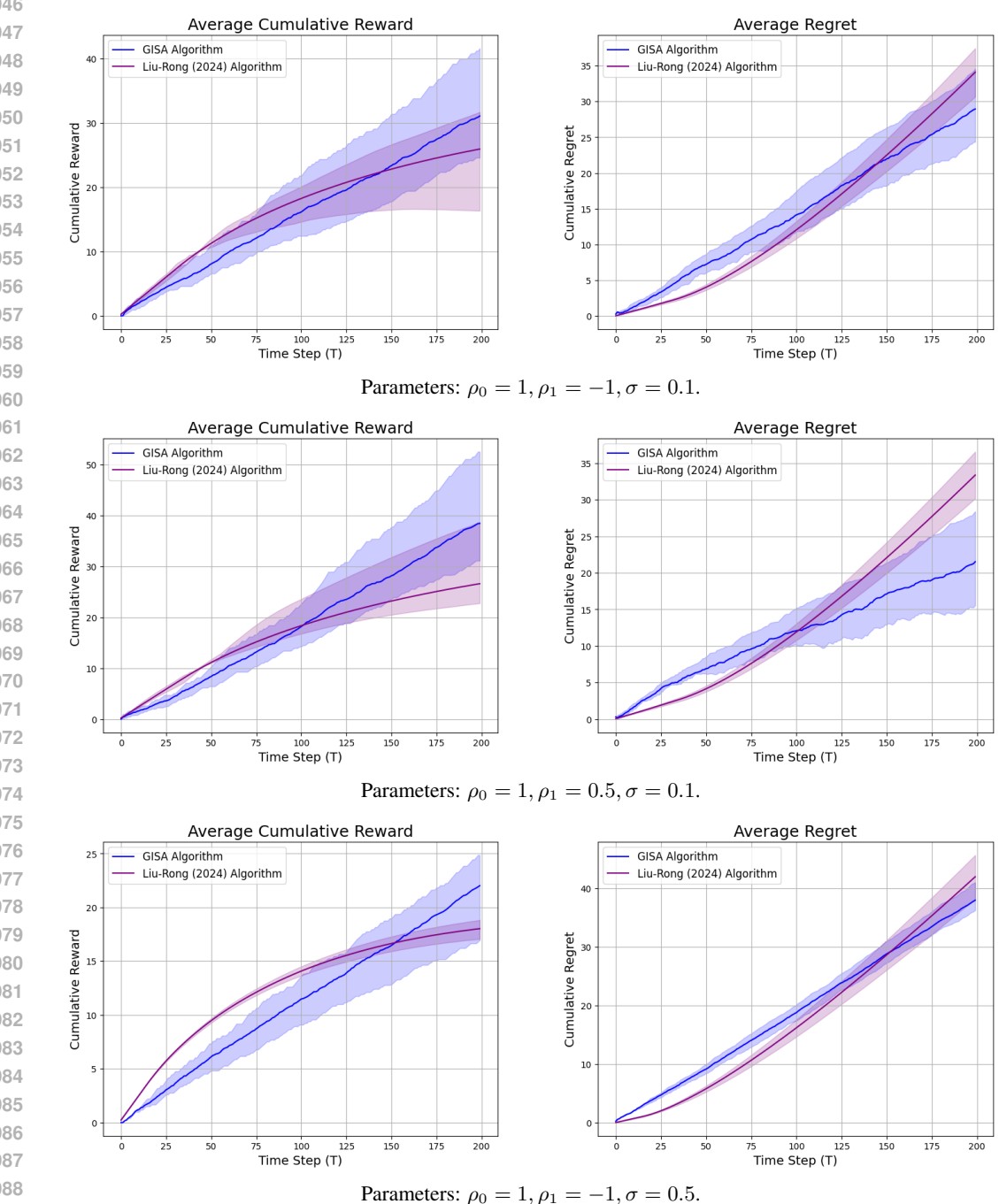

Figure 7: Mean values are calculated over 1,000 trials, with shaded regions representing confidence intervals, all of which fall within the first quartile.

### G.3 MULTI-DIMENSIONAL STACKELBERG GAME (SSG)

We consider a two-player Stackelberg game where the leader $A$ and the follower $B$ choose their actions from a shared action space $\mathbb{R}^n$. The leader chooses an action $\mathbf{a} \in \mathbb{R}^n$, anticipating the follower's response $\mathbf{b} \in \mathbb{R}^n$, where $n = 5$. Both players' rewards are influenced by a combination of the difference in their actions and quadratic penalties on their individual actions. The problem

is constrained by weighted $L_1$-norm bounds on both $\mathbf{a}$ and $\mathbf{b}$, which limit the magnitude of their respective actions.

The leader's reward function $\mu_A$ is defined as:

$$\mu_A(\mathbf{a}, \mathbf{b}) = \theta_A^\top(\mathbf{a} - \mathbf{b}) - \theta_A^\top f(\mathbf{a}) + \epsilon, \quad \epsilon \in \mathcal{N}(0, \sigma) \tag{G.12}$$

where:

- $\mathbf{a} \in \mathbb{R}^n$ is the leader's action,

- $\mathbf{b} \in \mathbb{R}^n$ is the follower's action,

- $\theta_A \in \mathbb{R}^n$ is a weight vector for the leader,

- $f(\mathbf{a})$ is the quadratic penalty function applied elementwise, such that $f(\mathbf{a}) = [\mathbf{a}_1^2, \mathbf{a}_2^2, \ldots, \mathbf{a}_n^2]$.

The leader seeks to maximize $\mu_A(a, b)$ by selecting $\mathbf{a}$, knowing that the follower will respond optimally.

The follower's reward function $\mu_B$ is defined as:

$$\mu_B(\mathbf{a}, \mathbf{b}) = \theta_B^\top(\mathbf{a} - \mathbf{b}) - \theta_B^\top g(\mathbf{b}) \tag{G.13}$$

where:

- $\mathbf{a} \in \mathbb{R}^n$ is the leader's action,

- $\mathbf{b} \in \mathbb{R}^n$ is the follower's action,

- $\theta_B \in \mathbb{R}^n$ is a weight vector for the follower,

- $g(\mathbf{b})$ is the quadratic penalty function applied elementwise, such that $g(\mathbf{b}) = [\mathbf{b}_1^2, \mathbf{b}_2^2, \ldots, b_n^2]$.

The follower seeks to maximize $\mu_B(\mathbf{a}, \mathbf{b})$ by choosing $\mathbf{b}$, given the leader's action $\mathbf{a}$.

Both players are subject to weighted $L_1$-norm constraints on their actions:

$$\sum_{i=1}^{n} |\theta_{A,i} a_i| \leq C_A \quad \text{for the leader} \tag{G.14}$$

$$\sum_{i=1}^{n} |\theta_{B,i} b_i| \leq C_B \quad \text{for the follower} \tag{G.15}$$

where $C_A$ and $C_B$ are constants that limit the magnitude of the actions $\mathbf{a}$ and $\mathbf{b}$, respectively, and $\theta_{A,i}$, $\theta_{B,i}$ are the elements of $\theta_A$ and $\theta_B$.

**Follower's Optimization Problem (Best Response):** Given the leader's action $\mathbf{a}$, the follower solves the following optimization problem:

$$b^*(\mathbf{a}) = \arg\max_b \left(\theta_B^\top(\mathbf{a} - \mathbf{b}) - \theta_B^\top g(\mathbf{b})\right) \tag{G.16}$$

subject to:

$$\sum_{i=1}^{n} |\theta_{B,i} b_i| \leq C_B \tag{G.17}$$

This is a quadratic optimization problem due to the quadratic penalty $g(\mathbf{b})$, and the constraint enforces that the weighted $L_1$-norm of the follower's action does not exceed $C_B$.

**Leader's Optimization Problem:** Given the follower's best response $\mathbf{b}^*(\mathbf{a})$, the leader solves the following optimization problem:

$$a^* = \arg\max_a \left(\theta_A^\top(\mathbf{a} - \mathbf{b}^*(\mathbf{a})) - \theta_A^\top f(\mathbf{a})\right) \tag{G.18}$$

subject to:

$$\sum_{i=1}^{n} |\theta_{A,i} a_i| \leq C_A \tag{G.19}$$

This is also a quadratic optimization problem due to the quadratic penalty $f(a)$, and the constraint enforces that the weighted $L_1$-norm of the leader's action does not exceed $C_A$.

**Stackelberg equilibrium:** The Stackelberg equilibrium is reached when:

$$a^* = \arg\max_{\mathbf{a}} \left( \theta_A^\top (\mathbf{a} - \mathbf{b}^*(\mathbf{a})) - \theta_A^\top f(\mathbf{a}) \right), \quad b^*(\mathbf{a}) = \arg\max_{\mathbf{b}} \left( \theta_B^\top (\mathbf{a} - \mathbf{b}) - \theta_B^\top g(\mathbf{b}) \right) \tag{G.20}$$

subject to the respective $L_1$-norm constraints. At equilibrium, the leader chooses $\mathbf{a}^*$ that maximizes their reward given the follower's optimal response $\mathbf{b}^*(\mathbf{a})$, and the follower chooses $\mathbf{b}^*(\mathbf{a})$ that maximizes their reward given the leader's action.

### G.3.1 SSG Empirical Results

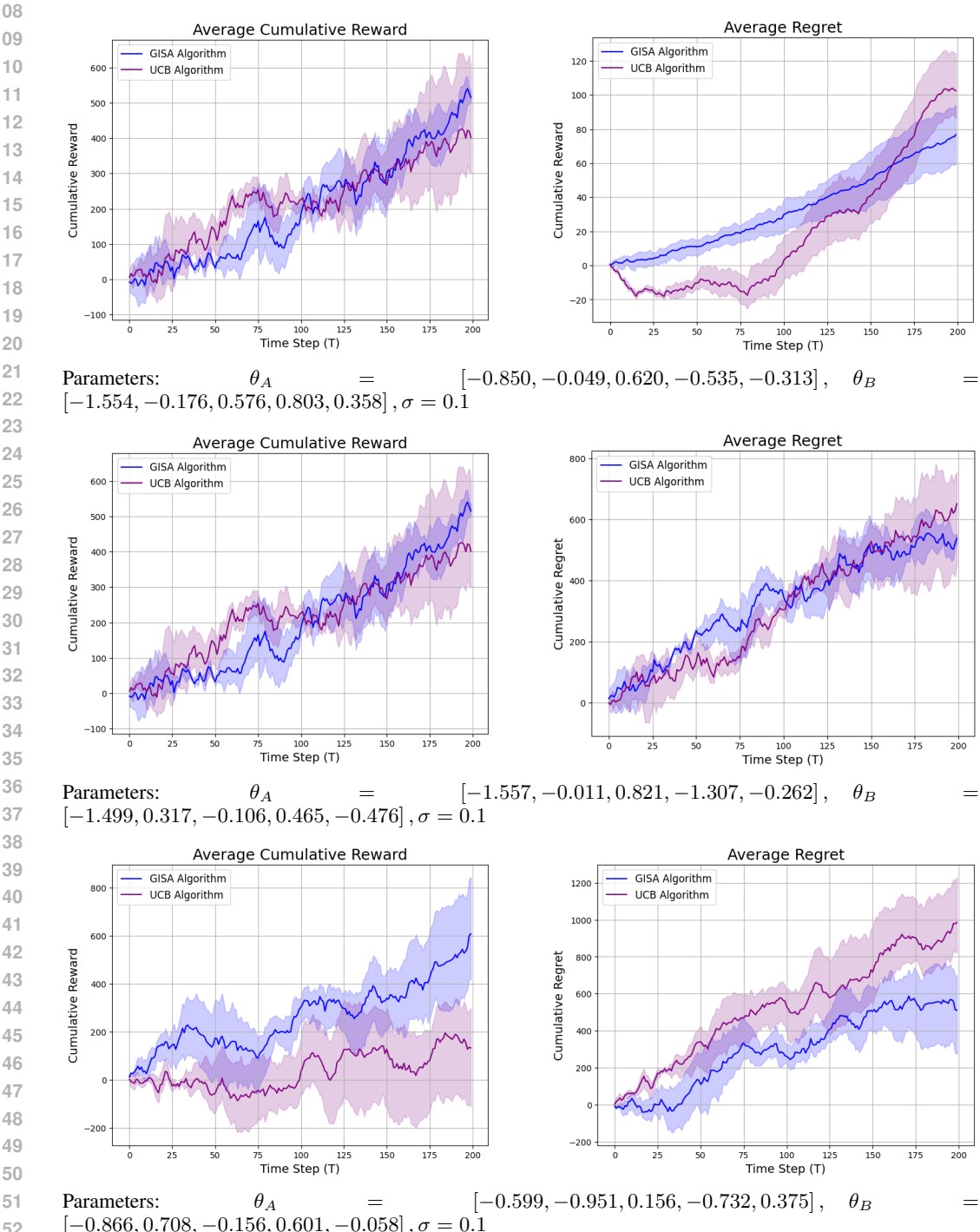

Parameters: $\theta_A = [-0.850, -0.049, 0.620, -0.535, -0.313]$, $\theta_B = [-1.554, -0.176, 0.576, 0.803, 0.358]$, $\sigma = 0.1$

Parameters: $\theta_A = [-1.557, -0.011, 0.821, -1.307, -0.262]$, $\theta_B = [-1.499, 0.317, -0.106, 0.465, -0.476]$, $\sigma = 0.1$

Parameters: $\theta_A = [-0.599, -0.951, 0.156, -0.732, 0.375]$, $\theta_B = [-0.866, 0.708, -0.156, 0.601, -0.058]$, $\sigma = 0.1$

Figure 8: Mean values are computed over 1,000 trials. All shaded areas, denoting confidence intervals, are within a quarter quantile. UCB arms were discretized to increments of 200, with an exploration constant $\alpha_{UCB} = 0.01$.

