# OpenReview forum: "Riemannian Manifold Learning for Stackelberg Games with Neural Flow Representations"
_ICLR.cc/2025/Conference — Submitted to ICLR 2025_

### Official Review · Reviewer_NGCU · 2024-10-20

**Soundness:** 3
**Presentation:** 3
**Contribution:** 3
**Rating:** 6
**Confidence:** 2

**Summary:**

1. **Summary:** This paper presents a novel framework for online learning in Stackelberg general-sum games, which involves two agents: a leader and a follower. The key innovation is the use of a learned diffeomorphism that maps the joint action space to a smooth Riemannian manifold, termed the Stackelberg manifold. This approach, facilitated by neural normalizing flows, creates isoplanar subspaces that simplify the application of standard bandit algorithms for online learning. The authors assume linearity of the agents' reward functions on the Stackelberg manifold, providing a theoretical foundation for regret minimization on convex manifolds and establishing finite-time bounds on simple regret for learning Stackelberg equilibria. The paper demonstrates the effectiveness of this manifold learning integration into game theory, showing its potential for neural normalizing flows as a tool for multi-agent learning. Empirical results are presented, comparing the approach to standard baselines and showing improvements in computational efficiency and regret minimization across applications in cybersecurity and economic supply chain optimization.

2. **Contribution:** The paper's contributions are significant in advancing the understanding of Stackelberg learning under imperfect information and providing a rigorous framework for optimizing Stackelberg games on spherical manifolds.

**Strengths:**

The primary strengths of this paper can be summarized in the following two aspects:

1. The paper presents a novel approach of mapping actions to a Riemannian manifold to solve the Stackelberg equilibrium.
2. The paper provides both rigorous theoretical analysis and comprehensive empirical validation.

**Weaknesses:**

Here are the major weaknesses from my perspective:

1. The authors have not compared their method to many current state-of-the-art approaches for solving Stackelberg equilibrium. The only comparison provided is in the experimental section, where they benchmark their algorithm against the dual-UCB method. The lack of broader comparisons with other recent algorithms, both from an empirical and theoretical standpoint, limits the contribution.

2. As a minor suggestion, the author should provide additional background information on Riemannian manifolds, as the concept may not be familiar to the broader deep learning community.


Additionally, I have a minor suggestion: I recommend that the authors switch to the official template for the conference to ensure a more formal presentation.

**Questions:**

Based on the discussion of the paper's strengths and weaknesses, I have the following questions for the authors:

1. What is the intuition behind mapping the action space to a Riemannian manifold? Does this approach stem from leveraging methods such as Riemannian gradient descent?
2. Is it necessary to consider a general Riemannian manifold, or is it sufficient to focus only on the unit sphere?
3. Can the authors provide a theoretical explanation of why their algorithm has advantages over previously proposed methods?

---

> ### Author Response · Authors · 2024-11-19
>
> We sincerely thank Reviewer NGCU for their thoughtful and positive review. We greatly appreciate the strengths you highlighted in our work and the constructive feedback you offered. We have worked respond to your concerns and update our document. Below is our detailed response:
>
> ---
>
> *What is the intuition behind mapping the action space to a Riemannian manifold? Does this approach stem from leveraging methods such as Riemannian gradient descent?*
>
> - The intuition behind mapping to a Riemannian manifold is that the game-theoretic equilibrium solution becomes easier to compute once we can characterize the follower’s best response. This effectively transforms a complex game-theoretic equilibrium computation problem into a much simpler problem of finding equilibrium points on convex subspaces induced by the leader. Deeper commentary addressing this concern is provided in **General Response 3**.
>
> *Is it necessary to consider a general Riemannian manifold, or is it sufficient to focus only on the unit sphere?*
>
> - The application of a Riemannian manifold is imposed more as a formality, enabling a rigorous extension of our work. Our study focuses on the D-dimensional sphere, and we propose a rigorous framework that could naturally extend to other Riemannian manifolds. For a deeper discourse, please refer to **General Responses 3 & 4**.
>
> *Can the authors provide a theoretical explanation of why their algorithm has advantages over previously proposed methods?*
>
> - Previous solution algorithms are often problem-specific—meaning the algorithm works only within a particular problem setting [1-B, 2-B, 3-B]—or are unable to construct a suitable feature map for a shared reward space in general. Additionally, our work addresses the challenge of building a solution framework for the multi-dimensional continuous action space, a problem with technical challenges distinct from those in previous discrete analyses.
>
> - A key issue is the equilibrium solution. In many cases, this solution does not always have a closed form in the multi-dimensional continuous joint action space, as illustrated by Eq. (2.3) in our paper. Computationally, solving for $\pi^*_A$ is not always tractable via standard methods, which typically involve iterating over the complete best response region of the follower.
>
> - Our work introduces the idea of bridging two seemingly disparate areas of research: multi-agent learning and differential geometry. This is achieved through normalizing flows, an enabling link introduced in recent ML literature [4-B, 5-B]. This allows us to map any ambient joint action space into a shared, tractable feature space $\mathbf{\Phi}$ with desirable properties (as outlined in Table 1, Pg. 4). For further discussion, please refer to **General Responses 1 & 3**.
>
> PS we have updated our draft to the ICLR 2025 template, thanks for pointing this out.
>
> ---
>
> We appreciate your encouraging and positive review of our work. If you have further questions or concerns, please do not hesitate to follow up. We would be happy to respond accordingly during the discussion period.
>
> ---
>
> [1-B] Zhao, Geng, et al. "Online learning in stackelberg games with an omniscient follower." International Conference on Machine Learning. PMLR, 2023.
>
> [2-B] Liu, Larkin, and Yuming Rong. "No-Regret Learning for Stackelberg Equilibrium Computation in Newsvendor Pricing Games." arXiv preprint arXiv:2404.00203 (2024).
>
> [3-B] Chen, Yiling, Yang Liu, and Chara Podimata. "Learning strategy-aware linear classifiers." Advances in Neural Information Processing Systems 33 (2020): 15265-15276.
>
> [4-B] Brehmer, Johann and Kyle Cranmer (2020). “Flows for simultaneous manifold learning and density estimation”. In: Advances in neural information processing systems 33, pp. 442–453.
>
> [5-B] Durkan, Conor et al. (2020). nflows: normalizing flows in PyTorch. Version v0.14. DOI: 10.5281/zenodo. 4296287. URL: https://doi.org/10.5281/zenodo.4296287.

---

> > ### Author Response · Authors · 2024-11-24
> >
> > Thank you once again for your positive review. With the reviewer-author discussion phase concluding soon, we would greatly appreciate it if you could let us know whether our responses have sufficiently addressed your concerns. If you have any additional questions or feedback, we would be happy to address them before the discussion period concludes.

---

> ### Comment · Reviewer_NGCU · 2024-11-25
> **Thank You**
>
> Thanks the authors for the response, and I will keep my score (which is positive).

---

> > ### Author Response · Authors · 2024-11-27
> >
> > Thank you again much for your efforts in reviewing our paper, and for the positive review! We shall continue refining our paper as we gather more feedback.

---

> > > ### Author Response · Authors · 2024-12-03
> > >
> > > As the discussion period comes to an end, we sincerely thank you for your valuable review and hope our responses highlight the significance our paper's contribution.

---

### Official Review · Reviewer_D4pN · 2024-10-22

**Soundness:** 3
**Presentation:** 2
**Contribution:** 2
**Rating:** 6
**Confidence:** 3

**Summary:**

This work proposes an approach to no-regret learning in Stackelberg games that first embeds the original joint actions in a Riemmanian manifold, in particular the sphere. The authors deploy techniques from normalizing flows to learn this invertible mapping using a loss function they designed to approximately satisfy several desiderata (D1-D6) that they argue a *Stackelberg Embedding* should satisfy. They then propose an algorithm that takes advantage of this embedding to estimate optimal *embedding* actions that they then map back to the original action space. Experiments show the regret of their algorithm (GISA) grows more slowly than the baseline UCB algorithm.

**Strengths:**

The idea of learning an embedding of a Stackelberg game such that the game can be reasoned about and solved in that space more easily is novel and interesting. To accomplish this, the authors leverage normalizing flows; in general, this work requires aggregating ideas and techniques from several typically disparate research areas which makes it original. While the paper focuses on the spherical manifold, the proposed loss function in equation 2.14 provides a way of learning the embedding, not just for spherical manifolds.

**Weaknesses:**

As the paper is currently presented, there was little discussion or motivation early on for why a spherical embedding (or any embedding) would be helpful for constructing an improved no-regret algorithm. The learned mapping is invertible so there's no dimensionality reduction, which is a typical motivation in other work making similar assumptions. Section 2.3 is introduced eventually, but it would seem the assumption of reward functions that are linear in the embedding space is critical for this approach to be strong. For example, in the linear MDP literature, this reward assumption is stated early on as part of the problem definition [1].

It's unclear how the precise experimental settings illustrate the importance of the embeddings. Do the reward functions in this game satisfy the linearity assumption in the chosen embedding space?

[1] Jin, Chi, et al. "Provably efficient reinforcement learning with linear function approximation." Conference on learning theory. PMLR, 2020.

**Questions:**

In general, I like the idea of the paper but I feel the writing needs to be substantially improved (e.g., motivation and layout of paper mentioned in weaknesses above). I also feel the experimental domains could be supplemented with a few toy domains that are tailored to the specific reward assumption for a given embedding. Including additional manifolds beyond the sphere manifold would strengthen the paper as well if you can construct meaningful motivations for them.

- Figure 1: I originally found Figure 1 to be saying something contradictory to Lemmas 4.3 and 4.4. The red arrow corresponding to $\\Phi_a$ appeared to be suggesting that $\\Phi_a$ runs horizontally. Similarly, $\\Phi_b$ points to a horizontal surface. Can you change $\\Phi_a$ to simply point to a single longitudinal line? Also, maybe make $\\Phi_b$'s arrow blue to be consistent?
- Algorithm 1: line 12 shows the algorithm returning an action before the for-loop completes. Is there an indent typo here? Do you mean *yield* rather than *return*? Also, line 12 states "return $a^*$" whereas I assume you mean $\\hat{a}^*$.
- Figure 2 / Algorithm 1: Presumably, these two can help the reader better understand the algorithm. Figure 2 uses $\\theta'\_B$ while Algorithm 1 uses $\\hat{\\theta}\_B$. Can these be made consistent? Also, if the green circle represents $\\mathcal{B}(\\theta, \\hat{\\theta}\_B)$, can you point out $\\tilde{\\theta}\_A$ specifically in Figure 2?
- Theorem 1: Is there a typo? "and Eq. (2.15) and Eq. (2.16)" is repeated. I'm not sure the sentence currently makes sense either. Do you mean to say "and the linear relationship to the reward function.... holds"?
- Section 4.1, near end of first paragraph: "In the Stackelberg game, since the follower moves first...". You mean the leader moves first.
- Paragraph "Mapping to a Spherical Manifold": The sentence "The transformation from spherical coordinates to Cartesian coordinates is used to map input features onto a $D$-dimensional spherical manifold" seems to read backwards. It should be "transformation from Cartesian to spherical..." or the "inverse of the transformation from...".
- How might you handle Stackelberg games with discrete action spaces?

---

> ### Author Response · Authors · 2024-11-19
>
> We deeply thank Reviewer D4pN for their thoughtful and meticulous review. We greatly appreciate the positive aspects you highlighted and the constructive feedback you provided. We've strived to rectify the typos indicated, and we are optimistic that the concerns and questions have been addressed comprehensively. Here is our detailed response:
>
> ---
>
> *Why a spherical embedding (or any embedding) would be helpful for constructing an improved no-regret algorithm (The learned mapping is invertible so there's no dimensionality reduction)?*
>
> - The geometric properties of a spherical embedding enable us to solve for anticipative best responses without enumerating the entire action space for the follower. This significantly simplifies equilibrium computation. A similar argument could be made for any manifold with a well-defined geometry. For a more detailed commentary, please refer to **General Responses 1 & 4**.
>
> *It would seem the assumption of reward functions that are linear in the embedding space is critical for this approach to be strong.*
>
> - Within the framework of linear bandits and multi-agent learning, this assumption is frequently applied and perfectly valid in most learning scenarios. A detailed explanation is provided in **General Response 1**.
>
> - As a minor point, [1-A] focuses on modelling MDPs, whereas our paper studies online learning in a repeated game, similar to [1, 2, 3], with extensions to multiple agents.
>
> *How might you handle Stackelberg games with discrete action spaces?*
>
> - A discrete action space implies that the image of the feature map $\mathbf{\Phi}$ is non-compact. Initially, this presents some limitations. If the degree of discretization is large enough, the discretization error would be sufficiently less than the inverse error (Definition D6 - Eq. (2.13)). This applies when we can freely select the granularity of discretization, typically used to approximate a continuous action space.
>
> - However, if the problem is inherently discrete (e.g., each player selects from $K$ discrete actions), our framework may not be well-suited, as it is designed for high-dimensional continuous action spaces. For such problems, other solution methods, like [2-A, 3-A], are better suited. Moreover, for small discrete action spaces, standard bandit algorithms, such as independently running UCB for each agent, are computationally tractable and converge to equilibrium, as cited in [4-A]. We focus primarily on the more complex higher dimensional continuous action space problem, rather than the smaller scale discrete problem.
>
> *Additional manifolds beyond the sphere manifold would strengthen the paper as well if you can construct meaningful motivations for them.*
> - A deeper discussion on this point is provided in **General Responses 3 & 4**.
>
> *Do the reward functions in this game satisfy the linearity assumption in the chosen embedding space?*
>
> - Exploring additional manifolds is a valid extension. As the first work bridging manifold learning with multi-agent learning, we begin with a rigorous examination of the spherical manifold, which naturally extends to additional convex Riemannian manifolds. Further discussion is available in **General Response 1A**.
>
> *The red arrow corresponding to $\overleftrightarrow{\mathbf{\Phi}}_a$ appeared to be suggesting that $\overleftrightarrow{\mathbf{\Phi}}_b$ runs horizontally.*
>
> - Thank you for this observation. In the updated draft, we adjusted Fig. 1 to clarify that $\overleftrightarrow{\mathbf{\Phi}}_{\mathbf{a}}$ refers to a single longitudinal line, which can move freely across the sphere's surface (directed by the red arrow).
>
> - The updated diagram demonstrates that for any given value of $\mathbf{a}$, $\overleftrightarrow{\mathbf{\Phi}}_a$ forms a red circumnavigating azimuthal great circle by varying $\mathbf{b}$. The same logic applies to $\overleftrightarrow{\mathbf{\Phi}}_b$, which varies for the small circles. We hope this update clarifies the diagram, and we welcome further suggestions.

---

> ### Author Response · Authors · 2024-11-19
>
> *Figure 2 / Algorithm 1: $\theta_B$ and $\theta'_B$.*
>
> - **$\hat{\theta}_B$ and $\theta'_B$:** To clarify, $\hat{\theta}_B$ represents the approximation of $\theta'_B$, estimated from sampling actions as the agents play $\mathbf{a}$ and $\mathbf{b}$. $\theta'_B$ represents the projection of the true objective vector onto the manifold $\mathbf{\Phi}$.
>
> - **Location of $\tilde{\theta}_A$:** In Fig. 2, $\tilde{\theta}_A$ lies on the border of the orange confidence region, representing where the leader's reward outcome $\mu_A$ likely lies (with high probability) when projected onto $\mathbf{\Phi}$. Specifically, this would be the left-most intersection of the red dotted line with the orange border.
>
> #### Minor Typos:
> ---
>
> *Algorithm 1: Line 12 shows the algorithm returning an action before the for-loop completes... *
>
> - Indeed, it should be $\hat{\mathbf{a}}^*$ and not $\mathbf{a}^*$.  Regarding the "**return** $\hat{\mathbf{a}}^*$," the algorithm should return the optimal estimate of the action at the end of $T$ samples. We have updated Algorithm 1 (Lines 11–14) to adopt the **yield** operator and placed the return at the end of the loop. Thank you for highlighting this point. We hope the algorithm is now clearer.
>
> *Is there a typo? "and Eq. (2.15) and Eq. (2.16)"*
> - Yes, this was a typo. We intended to write:  "...the linear relationship to the reward function in Eq. (2.15) and Eq. (2.16) holds..." This correction is reflected in the updated draft.
>
> *"In the Stackelberg game, since the follower moves first..."*
> - We corrected this typo to read: "...the leader moves first..." Thank you for pointing this out.
>
> *"Mapping to a Spherical Manifold": The sentence "The transformation from spherical coordinates to..."*
> - This typo was corrected to read:  "...from Cartesian coordinates to spherical..." Thank you for identifying this error.
>
> ---
>
> Thank you once again for taking the time to review our work. We hope the response clarifies most of the questions and concerns raised. Should you have further items for discussion, feel free to let us know and we will respond within the review period as timely as possible.
>
> ---
>
> [1-A] Jin, Chi, et al. "Provably efficient reinforcement learning with linear function approximation." Conference on learning theory. PMLR, 2020.
>
> [2-A] Colman, Andrew M., and Michael Bacharach. "Payoff dominance and the Stackelberg heuristic." Theory and Decision 43 (1997): 1-19.
>
> [3-A] Vasal, Deepanshu. "Stochastic stackelberg games." arXiv preprint arXiv:2005.01997 (2020).
>
> [4-A] Blum, Avrim and Yishay Mansour (2007). “From external to internal regret.” In: Journal of Machine Learning Research 8.6.

---

> > ### Author Response · Authors · 2024-11-24
> >
> > Thank you for your thoughtful and detailed review. As the reviewer-author discussion phase approaches its conclusion, we would greatly appreciate it if you could let us know whether our responses and revisions have sufficiently addressed your concerns which could affect the score. If you have any additional questions or suggestions, we would be happy to address them during this time.

---

> > > ### Comment · Reviewer_D4pN · 2024-11-24
> > >
> > > Thank you for your responses. I have follow-up questions.
> > >
> > > In line 11 of Algorithm 1, are you using your learned normalizing flow? Is this $\phi^{-1}$ actually $\hat{\phi}^{-1}$? In all experiments, do you learn this mapping in a separate step before running Algorithm 1?
> > >
> > > How do you compute the uncertainty bound $C_{\theta^*}(t)$? Do you actually estimate the objective vectors as in equation 2.17 in experiments?

---

> > > > ### Author Response · Authors · 2024-11-25
> > > >
> > > > **$\hat{\phi}^{-1}$ vs $\phi^{-1}$:**
> > > >
> > > > Correct, $\hat{\phi}^{-1}$ and $\phi^{-1}$ are used interchangeably.
> > > >
> > > > In our experiment, $\hat{\phi}$ is learned via normalizing flows, independently of the problem setting. For example, it could map a bounded 2D Euclidean space to a 2-sphere, or bounded $\mathbf{R}^3$ to a 3-sphere, etc. This is achieved by generating data uniformly in the ambient space and fitting a normalizing flow to a D-sphere based on the criteria in Table 1. Consequently, we only have access to $\hat{\phi}^{-1}$, not $\phi^{-1}$, which is why they are used interchangeably in the paper. However, we are open to standardizing the notation in a quick revision if needed. It is important to emphasize that $\hat{\phi}^{-1}$ is a learned normalizing flow, constructed independently of the problem setting, before running Algorithm 1.
> > > >
> > > > **Computing**  $\mathcal{C}_{\theta^*}(t)$:
> > > >
> > > > $\mathcal{C}_{\theta^*}(t)$ represents the radius of an uncertainty ball as a scalar function, based on the linear relationships between $\hat{\phi}$ and $\mu_A$ and $\mu_B$, respectively. Specifically, $\hat{\mu}_A = \langle \hat{\phi}, \hat{\theta}_A \rangle$, with a similar expression for $\mu_B$.
> > > >
> > > > This scalar function decreases with the number of samples. Assuming sub-Gaussian noise, a conservative upper bound on the confidence radius could be $1 / \sqrt{t}$ due to the central limit theorem.
> > > >
> > > > In our experiments, we use a tighter $\mathcal{C}_{\theta^*}(t) = \sqrt{\log(t) / t}$, as our sampling strategy optimistically explores under uncertainty. This is consistent with the theory presented in Theorem 2 from [1-C], further analyzed for the finite norm in Lemma 3.1 from [2-C].
> > > >
> > > > Nevertheless, a more conservative bound, such as $1 / \sqrt{t}$, would also work in this setting, although it may lead to slower convergence due to its looser nature. But this would be reflected as a consequence of the bound from Thm. 1 (our paper).
> > > >
> > > > **Estimating $\hat{\theta}$ according to Eq. 2.17:**
> > > >
> > > > Yes, we continuously re-estimate our approximations of $\hat{\theta}_A$ and $\hat{\theta}_B$ as new actions are taken (samples are drawn), during the course of Algorithm 1.
> > > >
> > > > ---
> > > > [1-C] Abbasi-Yadkori, Yasin, Dávid Pál, and Csaba Szepesvári. "Improved algorithms for linear stochastic bandits." Advances in neural information processing systems 24 (2011).
> > > >
> > > > [2-C] Liu, Larkin, and Yuming Rong. "No-Regret Learning for Stackelberg Equilibrium Computation in Newsvendor Pricing Games." arXiv preprint arXiv:2404.00203 (2024).
> > > >
> > > > ---
> > > > We hope this addresses your additional questions and concerns. If there's anything else we can clarify or improve, please let us know.

---

> > > > > ### Comment · Reviewer_D4pN · 2024-11-25
> > > > >
> > > > > Thanks for those clarifications. In particular, I wasn't sure if you were using $\hat{\phi}$ or the closed form expressions in Appendix G.1 "MAPPING BETWEEN SPHERICAL AND CARTESIAN COORDINATES". It seems like you could use the mappings in G.1 instead of the learned normalizing flow, which seems like an important ablation test to run to me.
> > > > >
> > > > > I'm having a hard time absorbing this paper and thus a hard timing providing a concise review. I think there are two main areas for improvement: 1) constructing evidence for the importance of learning a normalizing flow, 2) the writing.
> > > > >
> > > > > 1) The reader is introduced to normalizing flows which is highlighted as a key contribution of the paper (in the abstract and introduction) but normalizing flows (nor the notation) ever appear again after Section 2.2. Moreover, it’s not yet clear to me how critical $\hat{\phi}$ is to the approach given the mappings G.1 are available. If you run your same experiments but with the mappings in G.1, what performance do you see? If the performance is similar, what is the importance of normalizing flows?
> > > > >
> > > > > 2) At a high level, I think there is currently too much distracting information, and it is hard to "see the forest through the trees".
> > > > > - The paragraph on Problem Setting mentions optimal transport and causal game theory. We never see those words again in the paper. Why are you prompting the reader to think about those two areas?
> > > > > - Notation: We see 5 versions of $\theta$: $\theta$, $\theta’$, $\hat{\theta}$, $\tilde{\theta}$, $\theta^*$. Is that necessary? That's only one variable that the reader needs to keep track of among the many in this paper. Are all the variations on notation kept consistent throughout the paper? Are their meanings introduced, i.e., $\hat{\cdot}$ means..., $\tilde{\cdot}$ means ..., etc? Somewhat similarly, what is $Q$ versus $\phi$? Are these not essentially the same maps? Do you need separate names for them?
> > > > > - Algorithm: This is probably the most important IMO. Algorithm 1 only accounts for a subset of the components of your algorithmic pipeline (action embedding, objective parameter estimation, strategy learning). It would help me to better digest your proposed approach if you presented all the components together in one place (e.g., a diagram, more complete pseudocode, etc). This should probably go early in the paper (e.g., page 2). Moreover, the algorithm pseudocode should not contain "global" variables. For example, where are $\hat{\theta}_A$ and $\hat{\theta}_B$ estimated in the course of Algorithm 1? They appear as inputs the $\mathcal{B}$, but their estimation does not appear to be represented anywhere nor are they inputs to the algorithm.
> > > > >
> > > > > I understand the challenge in presenting interdisciplinary material requiring several areas of background. That being said, I'll point out a few areas that you could IMO remove to make space to add details explaining further explaining your approach:
> > > > > - equation 2.5 is standard in normalizing flows and doesn’t need to be highlighted as a stand alone equation
> > > > > - bi-level optimization structure seems like a detour that could be moved to the appendix
> > > > > - geodesically convex sets and geodesically convex manifolds are intuitive to someone vaguely familiar with convex sets and riemannian manifolds (if the reader isn't vaguely familiar, they'd need to brush up before reading this paper which I think is fair)

---

> ### Author Response · Authors · 2024-11-26
>
> Thank you for following up on our responses and providing suggestions to add additional polish to our paper. We also appreciate your acknowledgment that our work serves as an interdisciplinary bridge across three communities (topological learning, online learning, and game theory), which presents some challenges in communicating our work across different vocabularies and definitions. Nevertheless, we believe our work is impactful precisely because it is difficult, as it bridges previously undiscovered connections between different technological domains.
>
> We have provided a modestly revised paper, with pertinent changes highlighted in blue. To further clarify the issues you raised,
>
> ---
>
> ### Major Components:
>
> **$\hat{\phi}^{-1}$ vs $\phi^{-1}$ given the mappings G.1:**
>
> This is a very good point. In principle, although the tooling from G.1 already provides a transform from Cartesian to spherical coordinates, it does not guarantee the properties over the manifold as enforced by the criteria from Table 1 (i.e., Lipschitzness, sensitivity, and even spreading). Therefore, without normalizing flows (NF), this causes the mapped data on the spherical manifold to be highly localized or grouped into highly sporadic clusters (we previously confirmed this in tests comparing a non-NF spherical map to an NF one). As the embedding does not cover the entire manifold as intended, this renders our solution method (from Sec. 4.3) ineffective. To reiterate, the role of the NF tooling is to make the data in the feature space evenly spread across the manifold, in addition to ensuring several ideal properties as specified in Table 1.
>
> **Algorithm 1 only accounts for a subset of the components:**
>
> Allows us to clarify each of the 3 components to our algorithm as you have identified:
>
> - **Action embedding:** There is indeed an offline training mechanism that is data-independent and constructed before Algorithm 1 proceeds to run. (We updated the description in Algorithm 1 to state this before proceeding with the iterative online learning section.)
>
> - **Objective parameter estimation:** Typically, the re-estimation procedure for $\hat{\theta}_A$ and $\hat{\theta}_B$ is implied as new data is observed. (Nevertheless, we updated Algorithm 1 to explicitly state a step in parameter re-estimation based on collected data history.)
>
> - **Strategy Learning:** The main goal of mapping to the Stackelberg manifold is that we obviate the need for strategic learning, as the best response of the follower lies in a confined subspace that is convenient for the follower to derive (as explained in Section 4.3 and illustrated in Fig. 2). Therefore, strategy learning under uncertainty resolves to geodesic distance minimization, which is far more computationally tractable than other approaches from the game theory (such as bi-level optimization  for example).
>
> **Algorithm 1 Changes:** We have updated the paper, specifically *Algorithm 1*, to present a more comprehensive exposition of our methodology in its entirety:
>
> We overhauled the exposition of Algorithm 1 to be more expressive, adhere to coding conventions, and explicitly add in previously missing steps (which we had assumed were obvious to the reader) in order to communicate it more clearly. Specifically:
>
> - We explicitly state when the actions $\mathbf{a}$ and $\mathbf{b}$ are taken and rewards received ($\mu_A$ and $\mu_B$). *Line 16*.
>
> - Added an initialization step for $\hat{\theta}_A$ and $\hat{\theta}_B$, and empty arrays $\mathcal{U}$ and $\mathcal{D}$. *Line 4*.
>
> - We explicitly state how the estimates for the actions $\mathbf{a}$ and $\mathbf{b}$ are obtained. *Lines 9, 12, and 14*.
>
> - We explicitly state the procedure for constructing $\mathbf{\Phi}$ before the online algorithm begins (Line 5), and also the procedure for re-estimating $\hat{\theta}$ as new online data arrives (Line 18).

---

> > ### Author Response · Authors · 2024-11-26
> >
> > ### Minor Points / Changes in Notation:
> >
> > **Terminology Optimal Transport and Causal Game Theory:**
> >
> > - We removed the term *optimal transport* from the revised paper (initially, we meant that Eq. 2.4 can be formulated as an OT problem, but this was indeed not further investigated in the paper).
> >
> > - We changed the sentence regarding "... causal game theory ..." to "It explores game theory, utilizing geometric topologies to better understand agent behavior and simplify computations." in Line 68 of the revised paper.
> >
> > **Notation:**
> >
> > - For convenience and notation reduction, we removed $\hat{\phi}$ from the paper and replaced it with only $\phi$, noting after Line 188 that it represents $\hat{\phi}$ hereafter, for convenience.
> >
> > - **Q and $\phi$:** Q and $\phi$ are not exactly the same, as Q takes in Cartesian coordinates and maps them to spherical coordinates, without any NF component nor does it constitute a *feature map* (as in Eq. 2.16), whereas $\phi$ does. We use Q only as a tool to state our Def. 2.1. However, since the distinguishing factor is minor, to reduce excess notation, we removed $Q$ from the revised paper and replaced it with $\phi$. We explain further that we imbue $\phi$ with the additional property of Def. 2.1 such that it forms a Bipartite spherical map.
> >
> > - In addition to removing the placeholder notation $\tilde{\theta}$, we summarize (also stated in the paper) the definitions of $\theta$:
> >
> >     * $\theta$ is a general representation of $\theta$.
> >
> >     * We use $\tilde{a}$ instead of $\tilde{\theta}$ to reduce the notation conflation around $\theta$.
> >
> >     * $\theta^*$ represents the true parameters of $\theta$.
> >
> >     * $\theta'$ is defined in Eq. 4.7; it is the projection of $\theta$ onto $\mathbf{\Phi}$, in this case, we normalize w.r.t. the magnitude $||\theta ||$.
> >
> > **Additional Definitions:**
> >
> > - **Def. Convex Manifold:** W.r.t. relegating the description of the definitions for *geodesically convex sets*, we agree with your suggestion and moved the definitions to the appendix accordingly.
> >
> > - **Bi-level optimization section:** We agree with you that the bi-level optimization section, although it enriches the appreciation and understanding of our work, could be viewed as supplemental. Nevertheless, some of the items are used subsequently, for example, when it comes to defining simple regret in Eq. 4.3, we use the term $\mathcal{H}$. To compromise, we've shortened this section and moved a chunk of supplemental text of it into the Appendix.
> >
> > - **Eq. Norm. Flows** W.r.t. the *normalizing flows equation in Eq. 2.5*, we believe that there could be other readers who are not so familiar with this topic, and we should keep it in the main paper to have it as a self-contained and self-explanatory paper without flipping to the Appendix. This paper is intended for a cross-disciplinary audience, so we do not want to make too many assumptions regarding the reader's knowledge across domains. Either way, if reviewer D4pN believes relegating it to the appendix still makes more sense, we are also happy to do so as well.
> >
> > ---
> > We thank you very much for the suggestions and comments you have given us - they've definitely impacted the quality of our paper. Should there be any additional concerns or if you feel further considerations could improve our work, please don't hesitate to share them with us.

---

> > > ### Comment · Reviewer_D4pN · 2024-11-26
> > >
> > > Thank you for making all of those changes, especially Algorithm 1. I have increased my score.
> > >
> > > Regarding G.1, that's an interesting point. I actually remember I've observed that phenomena of non-uniform coverage when using those transformations myself and hadn't thought to try normalizing flows.
> > >
> > > Regarding equation 2.5, I actually only meant that you could display the equation *inline* so it takes up less room.

---

> > > > ### Author Response · Authors · 2024-11-27
> > > >
> > > > Thank you again much for your efforts in reviewing our paper and increasing the score! We will certainly further improve and revise our paper as we continue to gather feedback.

---

> > > > > ### Author Response · Authors · 2024-12-03
> > > > >
> > > > > As the discussion period comes to an end, we thank you for carefully reviewing our work. We appreciate your openness to our revisions leading to a strengthened assessment of our paper.
> > > > >
> > > > > We hope that the current improvements further highlight the potential significance of our contribution.

---

### Official Review · Reviewer_R8Sr · 2024-11-04

**Soundness:** 2
**Presentation:** 2
**Contribution:** 2
**Rating:** 5
**Confidence:** 3

**Summary:**

The paper studies Stackelberg games where a leader and follower take actions sequentially. The framework presented in the paper maps the joint actions pace to a smooth Riemannian manifold for efficient methods for online learning. It assumes the agents' rewards function are linear on the Stackelberg manifold and then uses bandit algorithms.

**Strengths:**

The paper tries to provide a framework to map the Stackelberg games to a bandit problem by using neural normalizing flows and by assuming the reward function is linear on the mapped manifold, allowing the use of the bandit results to study the game. The paper could provide regret analysis and empirical results.

However, the reviewer is very confused by the motivation of the approach. More details of weakness is shown below.

**Weaknesses:**

The paper is not very well structured and not very motivated. The contributions are not clear. The problem is statement is not clearly stated. It seems like the paper considers a 2-player Stackelberg game and considers bounding a form of regret as a measure of performance. However, Section 2.2 describes a procedure to obtain an embedding on the D-sphere where the reward functions are linear as in (2.16). First of all, it is unclear why people should do this embedding (motivation is not clear). Secondly, It is not clear how such an embedding is possible. The loss in 2.14 does not reflect that.

There are several lemmas and theoretical results in the paper that do not seem to hold as currently stated and/or the provided proof doesn’t make sense. Examples include:

1) Definition 2: it seems like such a manifold does not exist unless it is a singleton, because any two distinct points as a subset is certainly not geodetically convex.
2) Lemma 4.1 where the statement does not make sense for any non-compact domain.
3) Lemma 4.3: the justification of this lemma does not make sense other than restating the Poincare-Hopf Theorem.

In the simulations, the rewards models seem to be linear in \theta parameters automatically, then it is not clear what is the purpose of Section 2.2. regarding the embedding.

**Questions:**

see comments under the weakness section.

---

> ### Author Response · Authors · 2024-11-13
>
> Hello R8Sr,
>
> We appreciate the time you’ve taken to review our work, and give invaluable feedback. However, we noticed some ambiguity in your comments that we’d like to clarify to ensure we’re responding as accurately as possible.
>
> Firstly, could you kindly clarify what you’re referring to as “Definition 2”? It seems there may have been a slight typo, as Def. 2 does not appear in our current draft. Were you perhaps referring to Def. 2.1 on pg. 4, item D2 in Table 1 pg. 4, or Def. 4.1 on Page 6? Unfortunately, without further specification, it’s challenging for us to provide a precise response, and we want to ensure we fully address your point.
>
> Additionally, regarding your comment on Lemma 4.1, we would be grateful if you could elaborate on what specific aspects do not make sense. It would be great if you could provide a bit more specificity here.
>
> Feel free to share any clarifications at your earliest convenience, it would greatly help us to provide a more comprehensive response as timely as possible.

---

> > ### Comment · Reviewer_R8Sr · 2024-11-14
> > **Clarifications regarding some of the original comments**
> >
> > Regarding the first question about the "Definition 2", it was meant to refer to Definition 4.2, which concerns the definition of ``convex manifolds.'' The original comment essentially provides a counterexample showing that such a manifold cannot exist, because any set of two distinct points can be considered as a subset of any such convex manifold, which obviously cannot be a convex subset.
> >
> > Regarding the second question about Lemma 4.1, the original comment points out that the non-compactness of the convex region obstructs the validity of this lemma. But to make it more concrete why this lemma does not seem to hold in its current form, below is a simple counterexample:
> >
> > Consider the convex region to be the triangle with vertices (0,0), (0,1), and (1,0) in R^2, with the obvious boundary. Let \theta =(-1,1), then we will obtain \xi_\theta =(0,1). Now take \theta_A = (0,0.1) and note that the geodesic distant d(\xi_\theta, \theta_A) = 0.9. Finally, let \theta_B = (0.5,0.5) and observe that the geodesic distance d(\xi_\theta,\theta_B) = 0.5 \sqrt(2) ~= 0.7 and <\theta, \theta_B> = 0; however, <\theta, \theta_A> = 0.1. This contradicts the claim of this lemma that  <\theta, \theta_A> = 0.9 must be smaller than <\theta, \theta_B> = 0.
> >
> > The reviewer believes this should suffice to clarify the original comments.

---

> ### Author Response · Authors · 2024-11-19
>
> We deeply appreciate Reviewer R8Sr's insightful and detailed commentary. For certain points of contention, we have made subtle yet vital updates to our theoretical arguments. For others, we have provided a deeper complementary explanation to clarify the argumentation and justification for specific theorems and lemmas. Overall, we aim to strengthen our claims and address areas of contention. Below are our detailed responses:
>
> ---
>
> *It seems like the paper considers a ... bounding a form of regret as a measure of performance. However, Section 2.2 describes a procedure to obtain an embedding on the D-sphere .... It is unclear why people should do this embedding.*
>
> - Naturally, when agents know the parameters of $\theta$ and act rationally, the simple regret, as defined in Eq. (4.4) 0, is 0, as there would be no deviation from the equilibrium solution. For further clarification regarding regret and its connection to the embedding $\mathbf{\Phi}$ within the framework of linear bandits, please refer to **General Responses 1 & 3**.
>
> *In the simulations, the reward models seem to be linear in $\theta$ parameters automatically.*
>
> - For more clarification on the structure of the reward functions, please refer to **General Response 1A**. If further clarification is still required, could Reviewer R8Sr kindly specify this point further?
>
> *It is not clear how such an embedding is possible. The loss in 2.14 does not reflect that.*
>
> - The embedding $\mathbf{\Phi}$ is learned via normalizing flows [5, 6, 7] (also cited in the paper), which construct a bijective map between ambient space and the manifold. The loss component $\mathcal{L}_\phi^N$ is the loss function for the normalizing flow. Additional loss functions, outlined in Table 1 (Pg. 4), ensure uniform spreading, Lipschitz continuity, and bijective properties across the manifold. Once we map our ambient data to a well-behaved embedding space $\mathbf{\Phi}$, **General Response 1** provides an argument for why $\mathbf{\Phi}$ is suitable for the linear bandit setting.
>
> - If further clarification is necessary, the reviewer is kindly invited to specify which aspects in Lines 193–224 (in the updated draft) might still be unclear, and we would be happy to provide additional discussion.
>
> *Definition 4.2: It seems like such a manifold does not exist unless it is a singleton because any two distinct points as a subset are certainly not geodesically convex.*
>
> - We thank Reviewer R8Sr for pointing out the inconsistency in Definition 4.2. We have clarified our original intent and updated the definition. We believe this change has no impact on downstream results.
>
> - The confusion arises from the definition of "possible." We do not mean that any subset of $\mathbf{\Phi}$ is possible; rather, we refer to all *permissible* subsets that geodesically convex-combine to form $\mathbf{\Phi}$. In other words, the manifold is contained within a union of convex subsets $\mathcal{S}_\mathbf{\Phi}$. Therefore, given reviewer R8Sr's feedback, we have changed the wording in our definition to:
>
> **Definition 4.2 (New - Now Def. C.2 in Appendix). Convex Manifolds:** A convex manifold is a manifold where the geodesic between any two points on the manifold fall within, or constitutes, a geodesically convex set $\mathcal{S}_\mathbf{\Phi}$ (as per Definition 4.1).

---

> ### Author Response · Authors · 2024-11-19
>
> *Lemma 4.1: The statement does not make sense for any non-compact domain, & presented counter-example.*
>
> - The counter-example provided by Reviewer R8Sr is valid but does not impact our theoretical results, as it lies outside our intended use case. Nevertheless, we have strengthened the assumptions of Lemma 4.1 in our updated draft by adding the additional constraint of orthogonality of $\xi_\theta$ with respect to $\mathbf{\Phi}$. To restate the counter-example, arbitrarily selecting a point in $\mathbb{R}^2$ Euclidean space and projecting it onto an arbitrary convex manifold in $\mathbb{R}^2$ does not guarantee that a larger divergence angle corresponds to a larger geodesic distance.
>
> - Thus, for Lemma 4.1 to hold, we must enforce that $\xi_\theta$ forms a vector intersecting the manifold *orthogonally*. This is already implied when using a unit D-sphere as the manifold of choice. We will explicitly state this in the updated draft to ensure rigour, and we have updated our draft with an updated Lemma 4.1:
>
> **Lemma 4.1 (New). Geodesic Distance and Closeness to $\xi_\theta$:** Let $\Phi \subset \mathbb{R}^D$ be a manifold serving as a boundary of a convex set in $\mathbb{R}^D$. Given $\theta$, let $\xi_\theta \in \Phi$ be the point on the manifold that maximizes the dot product $\langle \theta, \xi_\theta \rangle$, **and is orthogonal to** $\mathbf{\Phi}$ **at the point of intersection**. For any two points on the manifold $\theta_A', \theta_B' \in \Phi$, if the geodesic distance between $\xi_\theta$ and $\theta_A'$ is greater than that between $\xi_\theta$ and $\theta_B'$, $\mathfrak{G}(\xi_\theta, \theta_A') > \mathfrak{G}(\xi_\theta, \theta_B')$, then the dot product satisfies $\langle \theta, \theta_A' \rangle < \langle \theta, \theta_B' \rangle$.
>
> - We thank Reviewer R8Sr for highlighting this matter.
>
> *Lemma 4.3: The justification of this lemma does not make sense other than restating the Poincaré-Hopf Theorem.*
>
> - Reviewer R8Sr is correct in asserting that we restate the Poincaré-Hopf (PH) theorem in Lemma 4.3. However, there is more to the argument than mere restatement. We must ensure the conditions for applying the PH theorem are met and provide additional arguments regarding the cardinality of the intersecting set.
>
> - First, we ensure that the criteria on the manifold subspace are met so the PH theorem can apply. Suppose we have a spherical manifold that is compact. The azimuthal subspaces $\overleftrightarrow{\mathbf{\Phi}}_a$ can be represented by a vector field with a fixed index, while the latitudinal subspaces $\overleftrightarrow{\mathbf{\Phi}}_b$ can be represented by trajectories of a vector field circumnavigating small circles around the sphere. For the D-sphere, the fixed index equals 2.
>
> - By the PH theorem, the index remains constant when elevating to higher dimensions. Since we work with constrained isoplanes that do not span the entire manifold, even for orthogonal subspaces, the cardinality of the intersecting subspace must be greater than 0. We believe it is valid to merge Lemma 4.3 with Lemma 4.4, as they are part of Lemma 4.4’s constructive argument. However, we intend to maintain the current structure for this draft. Please refer to **General Response 2** for additional insights into Lemma 4.3.
>
> ---
>
> Once again, we thank Reviewer R8Sr for their time and effort in helping improve our paper. Should further clarification be required, feel free to ask, and we would be happy to respond.

---

> > ### Author Response · Authors · 2024-11-24
> >
> > Thank you for your detailed and critical review. As the reviewer-author discussion phase nears its end, we would greatly appreciate it if you could review our responses and let us know whether they adequately address the theoretical concerns you raised which could affect the score. If you have any remaining questions or require further clarification, we would be happy to provide it during this period.

---

> ### Author Response · Authors · 2024-11-27
>
> We wanted to kindly remind you that the deadline for authors to make revisions to the manuscript is approaching soon (Nov. 27 AoE). While the discussion period has been generously extended by the ICLR committee until Dec. 2 AoE, revisions to the manuscript can only be made before Nov. 27 AoE.
>
> We believe your initial feedback has been very beneficial in clarifying and improving upon our paper. We have already made some revisions under the suggestions from your review and the other reviewers alike, **if there are any remaining concerns or suggestions that you feel would benefit from additional updates to the manuscript, we would be very grateful to hear from you before Nov. 27 AoE.**
>
> After that date, we are happy to patiently continue the discussion until the end of the discussion period to clarify questions and concerns. Thank you so much for your time and effort in reviewing our work, we appreciate it.
>
> ---
> PS. Upon the latest revision, Definitions 4.1 and 4.2 have been moved to the Appendix, now Definition C.1 and C.2 respectively.

---

> > ### Author Response · Authors · 2024-12-03
> >
> > As the discussion period comes to an end, thank you for your initial review and the time you dedicated to evaluating our work. We hope our revisions addressed the key concerns raised and clarified the contributions of our paper. Thank you for increasing our score in light of our response!

---

### Official Review · Reviewer_CJwA · 2024-11-05

**Soundness:** 3
**Presentation:** 3
**Contribution:** 4
**Rating:** 6
**Confidence:** 4

**Summary:**

This paper studies the Stackelberg general-sum games.  They propose a learned diffeomorphism that maps the joint action space to the so-called Stackelber Manifold using neural normalizing flows. This way, they convert the equilibrium learning problem into an online learning problem on manifolds. Applying multi-armed bandit methods, this proposed framework is shown to achieve efficient computation of the equilibrium. Simulation results are provided to validate the proposed methods.

**Strengths:**

The proposed framework is highly novel to me. It effectively bridges equilibrium learning, manifold learning, and online learning paradigms by establishing theoretical and practical connections. The application of normalizing neural flows shows particular promise for converting equilibrium learning problems into online manifold learning, and optimization problems, opening valuable new research directions.

**Weaknesses:**

The paper would benefit from a clearer organizational structure, particularly in Section 4. The progression of ideas through Lemmas 4.1-4.4 needs stronger motivation and explicit statements of their individual contributions to the overall narrative. Additionally, the technical foundation could be strengthened by providing clearer connections between consecutive lemmas and their role in building the theoretical framework and by including fundamental definitions from Riemannian geometry, especially the concept of geodesics, which is central to the framework. In summary, I hope the authors make fewer assumptions about the reader's background knowledge in differential geometry.

**Questions:**

- Could the authors discuss more on the roles of Lemmas 4.1-4.4?
- Could the authors explain what structures of Riemannian manifolds are essential in the proposed framework? Can this framework be applied to space with fewer structures, say a smooth differential manifold?

---

> ### Author Response · Authors · 2024-11-19
>
> We thank Reviewer CJwA for their positive and appreciative review of our work. We understand that the major concerns revolve around intuitively clarifying the theoretical arguments in our paper. We believe that most of these concerns are addressed in the General Responses. To address the specific concerns:
>
> ---
> *Could the authors discuss more on the roles of Lemmas 4.1–4.4?*
>
> - The Lemmas 4.1–4.3 aim to establish rigorous properties useful for online learning in the Stackelberg embedding $\mathbf{\Phi}$. Please refer to **General Response 2** for a detailed discussion of their purpose.
>
> *Could the authors explain which structures of Riemannian manifolds are essential in the proposed framework? Can this framework be applied to a space with fewer structures, such as a smooth differential manifold?*
>
> - The swapping of a Riemannian manifold for a generic smooth manifold has no direct consequences in our framework; our simple regret guarantees are derived from the spherical manifold. Nevertheless, extending our work could pose some challenges if measures of angles and distances are not well-defined. A more detailed discussion is provided in **General Response 4**.
>
> ---
>
> We are grateful for your positive review and invite you to reach out with any further questions, and we will do our best to respond promptly.

---

> > ### Author Response · Authors · 2024-11-24
> >
> > Thank you again for your encouraging review. As the reviewer-author discussion phase is coming to a close, we’d be grateful if you could let us know whether our responses have addressed your concerns. If you have any further questions or feedback, we’d be happy to address them during this discussion period.

---

> ### Author Response · Authors · 2024-11-27
>
> A warm thank you once again for your positive review.
>
> We wanted to kindly remind you that the deadline for authors to make revisions to the manuscript is approaching soon (Nov. 27 AoE). While the discussion period has been generously extended by the ICLR committee until Dec. 2 AoE, revisions to the manuscript can only be made before Nov. 27 AoE.
>
> We’ve already made some relevant updates and clarifications addressing all of the reviewers' feedback thus far. **If you have any further suggestions to revise our manuscript, we’d be grateful to hear them before the Nov. 27 AoE deadline.**
>
> We hope the updated manuscript and rebuttal will strengthen your assessment in the paper, and we'd be happy to continue the discussion until the end of the discussion period Dec 2 AoE.

---

> > ### Comment · Reviewer_CJwA · 2024-11-29
> >
> > I appreciate the authors' detailed responses to my questions. I tend to maintain my original score as it reflects my overall evaluation of the work.

---

> > > ### Author Response · Authors · 2024-12-03
> > >
> > > As the discussion period comes to an end, we thank you for your valuable feedback; we hope our revisions demonstrate the potential impact of this work further.

---

### Author Response · Authors · 2024-11-19
**General Response (cont.)**

### **General Response 4:** Significance of the Riemannian and Spherical Manifold

**Riemannian vs. Smooth Manifold:**  The selection of a Riemannian manifold (RM) for our study offers several key advantages. Notably, the RM enforces a consistent distance metric on any smooth manifold, allowing for a rigorous definition of geodesic distances. Operators such as the gradient are also well-defined on the topology. While this formalism adds security, Riemannian gradient descent [7] could be applied when less convenient manifold geometries (e.g., non-spherical) are imposed.

If we were to work with smooth manifolds alone, this would not pose severe problems for Lemmas 4.1–4.4 and Theorem 1 when dealing with a D-sphere. However, challenges could arise when extending the theory to non-spherical manifolds (e.g., hyperbolic spheres, ellipsoids). For such extensions, it is crucial to maintain a Euclidean metric for geodesic distances. Nevertheless, in this paper, we focus on establishing guarantees for the D-dimensional spherical manifold. The use of RM tooling ensures robustness against potential issues in future extensions of our framework.

**Non-Spherical Manifolds:** Some of the basic theory in our work could, in principle, extend to other types of manifolds, such as hyperbolic spheres or ellipsoids. However, this is not the primary focus of our paper. The following discussion regarding applications to non-spherical manifolds serves to enrich the framework and encourage future exploration. Specifically, Lemma 4.2 will hold for any convex manifold. Lemmas 4.3–4.4 will hold due to the Poincaré-Hopf theorem and manifold compactness.

However, Lemma 4.1 and Theorem 1 may require adjustments for non-spherical manifolds. For Lemma 4.1, we must ensure the geodesic distance between any two projected points on the manifold is proportional to the divergence angle formed by the objective vectors $\theta_A'$ and $\theta_B'$. If this is not strictly the case, Lemma 4.1 should be modified to include an alternative geodesic distance measure or orientation.

Theorem 1 is designed for the spherical manifold, as it is convenient to draw azimuthal lines tangent to the confidence projected onto $\mathbf{\Phi}$ as convex submanifolds (illustrated in Fig. 2). While a similar argument could be made for non-spherical manifolds, it would not follow directly from Theorem 1 and would require adjustments to how confidence regions are projected and subspaces are formed with respect to these regions. Nevertheless, the key contribution of our work is demonstrating how a complex game-theoretic equilibrium computation problem can be transformed into a simpler topological geometry problem, with the D-sphere serving as the pioneering example.

---
[1] Amani, Sanae, Mahnoosh Alizadeh, and Christos Thrampoulidis (2019). “Linear stochastic bandits under safety constraints”. In: Advances in Neural Information Processing Systems 32.

[2] Moradipari, Ahmadreza et al. (2022). “Feature and parameter selection in stochastic linear bandits”. In: Inter- national Conference on Machine Learning. PMLR, pp. 15927–15958.

[3] Zanette, Andrea et al. (2021). “Design of experiments for stochastic contextual linear bandits”. In: Advances in Neural Information Processing Systems 34, pp. 22720–22731.

[4] Brehmer, Johann and Kyle Cranmer (2020). “Flows for simultaneous manifold learning and density estimation”. In: Advances in neural information processing systems 33, pp. 442–453.

[5] Durkan, Conor et al. (2020). nflows: normalizing flows in PyTorch. Version v0.14. DOI: 10.5281/zenodo. 4296287. URL: https://doi.org/10.5281/zenodo.4296287.

[6] Dinh, Laurent, Jascha Sohl-Dickstein, and Samy Bengio (2016). “Density estimation using real nvp”. In: arXiv preprint arXiv:1605.08803.

[7] Bonnabel, Silvere. "Stochastic gradient descent on Riemannian manifolds." IEEE Transactions on Automatic Control 58.9 (2013): 2217-2229.

[8] Liu, Larkin, and Yuming Rong. "No-Regret Learning for Stackelberg Equilibrium Computation in Newsvendor Pricing Games." arXiv preprint arXiv:2404.00203 (2024).

[9] Roughgarden, Tim. "Algorithmic game theory." Communications of the ACM 53.7 (2010): 78-86.

[10] Lattimore, Tor, and Csaba Szepesvári. Bandit algorithms - Chapter 19. Cambridge University Press, 2020.

---

### Author Response · Authors · 2024-11-19
**General Response (cont.)**

### **General Response 2:** Intuition & Justification of Lemmas 4.1–4.3

**Intuition of Lemma 4.1:**  The divergence angle between any two objective vectors can be made proportional to the geodesic distance between the extension of the objective vector onto $\mathbf{\Phi}$. The use case here is quantifying the best response (BR) under a constrained space. The smaller the divergence angle between $\theta_A$ and $\theta_B$, the more preferable it is for Agent B (assuming $\theta_A$ is fixed). Given a constrained space $\overleftrightarrow{\mathbf{\Phi}}_{\mathbf{a}}$, Agent A (the leader) can anticipate what a rational follower would do by examining the geometry of the embedding space and the uncertainty surrounding $\theta_B$. Both agents will try to position the joint action on the manifold closest to their respective objective vectors $\theta_A$ and $\theta_B$. This new method for computing the BR under uncertainty allows us to greatly simplify the BR computation, which is crucial for calculating Stackelberg equilibria.

**Intuition of Lemma 4.2:** We argue that on any strictly convex manifold, there could exist only one unique reward-maximizing solution for each objective vector and convex subspace. This simplifies the search for the equilibrium location, as there is a unique point. This is termed a pure strategy in the existing game theory literature [8], as opposed to a mixed strategy (which arises in situations where multiple equilibria could exist).

**Intuition of Lemma 4.3:** The purpose of Lemma 4.3 is to highlight that, given the bipartite map in Definition 2.1, subspaces are guaranteed to intersect on the manifold. This is easy to visualize on a spherical manifold in the 2-sphere setting (e.g., longitudinal and latitudinal lines) but becomes challenging to intuit in higher dimensions. We rigorously argue that, just as in the 2-sphere case, the same principle holds in a D-sphere setting.

### **General Response 3:** Advantages of Online Learning on a Manifold

Provided the bipartite spherical map in Definition 2.1, when the agents map their joint actions to $\mathbf{\Phi}$ (or the D-sphere surface), the leader induces subspaces that simplify the follower’s best response region, both under perfect information and quantifiable uncertainty. This can be exploited to derive the high-probability worst-case outcome at each turn, based on the uncertainty of the parameter estimates for $\hat{\theta}_A$ and $\hat{\theta}_B$. This allows us to provide theoretical guarantees on the simple regret of the algorithm, which follows from Theorem 1.

---

### Author Response · Authors · 2024-11-19
**General Response**

### **General Response 1:** Linear Relationship of the Embedding Space $\mathbf{\Phi}$ to the Reward Space:

We affirm that there exists a linear relationship between the embedding space and the reward space. The linear bandit assumption—that the expected reward is the inner product between an embedding space and some linear parameters—is commonly accepted in the online learning literature [1, 2, 3, 10], as cited in our draft Section 2.2.2.  To demonstrate rigorously, let $Y = \mathbf{E}[\mu]$. Given a linear relation in the action space $X$, the reward can be expressed as:
$$
Y = \langle \tilde{\theta}, \tilde{\phi}(X) \rangle.
$$
This tool, $\tilde{\phi}(\cdot)$, is known as a feature map [3]. From one perspective, we can view the linear feature map as the final layer in a neural network output. Instead of learning the action-to-reward relation directly via a neural network, we learn the relation of the ambient space (or the joint action space) to the embedding space.

**Theoretical Argument:** Given $Y = \mathbf{E}[\mu]$ and a linear relation in the action space $X$, under some mild assumptions, we want to show that there always exist some alternative parameters $\theta$ and $\phi$ such that:
$$
\langle \tilde{\theta}, \tilde{\phi}(X) \rangle = \langle \theta, \phi(X) \rangle.
$$

Suppose we assume the existence of a bijective transformation $T: \tilde{\phi}(X) \mapsto \phi(X)$, and its inverse $T^{-1}: \phi(X) \mapsto \tilde{\phi}(X)$, which is smooth and differentiable with respect to $X$. We can then express:
$$
Y = \langle \tilde{\theta}, T^{-1}(\phi(X)) \rangle.
$$
To swap from $\tilde{\theta}$ to $\theta$, we select $\theta$ as:
$$
\theta = (J_T^\top)^{-1} \tilde{\theta},
$$
where $J_T$ is the Jacobian of $T(\cdot)$ with respect to $\tilde{Y}$, where $\tilde{Y} \in \text{Im}(\tilde{\phi})$. Therefore, we show the existence of $\theta$ such that:
$$
Y = \langle \theta, \phi(X) \rangle,
$$
where $\phi(X) \equiv T \circ \tilde{\phi}(X)$. This argument relies on the bijectiveness and smoothness of $T(\cdot)$, consequently being a diffeomorphism, a fair assumption as long as we consider $\tilde{\phi}(X)$ to be smooth and well-behaved. For non-smooth $\tilde{\phi}(X)$, we could settle for a continuous-to-discrete approximation, particularly useful for large discrete action spaces.

Since there are no limitations to what the learned feature map $\hat{\phi}$ can be, it offers great flexibility in constructing $\hat{\phi}$ to approximate $T \circ \tilde{\phi}(\cdot)$. For this purpose, we apply normalizing flows [4, 5, 6], a recent technology developed specifically for mapping ambient data to a desired manifold, $\mathbf{\Phi}$, of which multiple sufficient approximations for $\hat{\phi}$ could potentially exist. This enables the construction of $\mathbf{\Phi}$ such that the linear relation between the embedding space and reward space holds.

###  **General Response 1A:** Simulation Environments & Linear Structure of Reward Function
**Simulation Environments:** We present three games (R1, NPG, and SSG) where simulations were run to produce empirical results. The R1 and SSG games are both linear by design, as per the structure of Eq. G.1, G.2, and Eq. G.12 in Appendix G. This directly aligns with our reward function structure from (Eq. 2.16).

The NPG game is taken from the work of [8], representing a realistic supply chain application. Here, the reward function is multiplicative and additive, operating in 1D and 2D continuous space. In theory, any feature map can extend additional dimensions to the multiplicative factors to capture the multiplicative causal variables. This can be implicitly done via normalizing flows.

Nevertheless, knowing that there is an underlying non-linear to linear structure to the reward function defeats, at least in part, the motivation for our work. The key idea behind $\mathbf{\Phi}$ is to abstract away the need to know the exact reward structure, as there could be many embeddings in existence that could work under our proposed framework. We are able to select freely the architecture of the normalizing flow neural network, as well as increase the dimensionality of the spherical manifold (or the output dimension), to learn a sufficient embedding.

Given this freedom of architectural choice, we aim to learn an embedding that is well-behaved following the desiderata from D1–D6 in Table 1. The problem solver may not necessarily know the structure of the problem beforehand, and perhaps there exists only a set of offline data to begin with, where they will construct the best approximate model for the ideal Stackelberg embedding. As online data arrives, the model could, in principle, be retrained to produce a new feature map. This does not affect the procedure of Algorithm 1 (GISA).

---

### Author Response · Authors · 2024-11-19

We thank all of the reviewers deeply for their thoughtful and in-depth reviews of our draft. To efficiently address all concerns, we provide a General Response section, which serves as an additional discussion and clarification of points raised by multiple reviewers that overlap in terms of topic. We hope that the additional discussion will further improve the understanding of our work and resolve outstanding questions.

As suggested by some of the reviewers, we have also revised the manuscript with relatively minor updates addressing typos, minor adjustments to diagrams, adjustments to Definitions & Lemmas, and an update to the ICLR 2025 template.

---

---

### Author Response · Authors · 2024-12-03
**Authors' Response to Reviewer Feedback: Revisions and Key Contributions**

Once again, we deeply extend our gratitude to all the reviewers for their time and meaningful suggestions, many of which have been incorporated into this revision.

Our work addresses an inherently challenging problem, bridging multi-agent learning and manifold geometry. By introducing a novel connection between these domains using normalizing flows, we enable more efficient game-theoretic online learning in Stackelberg games compared to conventional solutions such as bi-level optimization.

In response to the feedback received, we have improved the clarity of our exposition to make the text accessible to a broader audience. We are especially grateful for the detailed feedback from positive reviewers, which has significantly enhanced the quality of our paper. Unfortunately, reviewer R8Sr, who raised strong initial concerns regarding our work, was unable to participate further during the discussion period, but we appreciate their useful input nonetheless. We believe our rebuttal and revisions have largely addressed those concerns, particularly those related to theoretical issues surrounding the manifold geometry.

We are pleased that the rebuttal and discussion have been very constructive, resulting in a strengthened assessment and an overall increase in the reviewers’ scores for our paper.

To highlight our core contributions again, we primarily focus on:

1. Providing a technique that enables the learning of a well-behaved manifold, $\mathbf{\Phi}$, via normalizing flows, as detailed in Section 2.2.2.

2. Establishing a rigorous theoretical foundation for our manifold learning technique. This learned diffeomorphism  reveals the linear relationship between the embedding space and the joint reward function, enabling straightforward Stackelberg regret guarantees.

Complementing our theory, we provide empirical results from realistic game-theoretic simulations, avoiding reliance on toy examples. Our extensible framework lays a foundation that can be adapted to future applications across diverse learning environments and alternative Riemannian manifolds. We consider our normalizing flow technique and theoretical guarantees in the context of multi-agent learning to be key contributions to the machine learning community.

We hope this summary effectively highlights the contributions and significance of our work. As always, we are grateful for the reviewers’ thoughtful ideas and suggestions, and we remain open to future refinements to our paper.

---

### Meta-Review · Area_Chair_qssT · 2024-12-19

**Metareview:**

Summary:
This paper investigates learning Stackelberg games with large, potentially infinite, action spaces. To address the challenge of large action spaces, the authors propose a framework that maps joint action spaces to a Riemannian manifold, making the problem more tractable. Assuming the reward functions are linear in terms of the learned mappings, the authors demonstrate that a no-regret algorithm can effectively learn optimal policies for both players. Experimental results suggest their approach outperforms baselines in certain game scenarios.

Strengths:

- The paper is well-written and clearly presented.
- Theoretical contributions and experimental results are sound.

Weaknesses:

- Motivation for the Riemannian manifold framework:
While the Riemannian manifold approach is mathematically elegant, the paper does not convincingly justify why such a complex framework is necessary compared to simpler alternatives, such as mappings in Euclidean space. This weakens the paper's overall motivation.
- Linearity assumption and efficiency:
The linearity assumption on the reward function is reasonable in theory, but its practicality diminishes when considering efficiency constraints, particularly when aiming for a low-dimensional representation. The authors do not adequately address how their framework balances this trade-off, leaving the motivation unclear.

Decision:
Reject. While the paper has clear strengths in presentation and theoretical grounding, the concerns about the motivation for the Riemannian manifold framework and the implications of the linearity assumption on efficiency prevent me from recommending acceptance.

**Additional Comments On Reviewer Discussion:**

During the discussion, the authors provided further explanations about their proposed framework, which partially addressed the concerns raised by the reviewers and myself. While their responses clarified some aspects of the framework and its motivations, I believe that a more thorough revision of the current draft is necessary to fully resolve these issues.

---

> ### Public Comment · ~Larkin_Liu1 · 2025-02-09
> **Response to AC's Final Notes from the Authors**
>
> We sincerely thank the AC and reviewers for their time, thoughtful feedback, and dedication to reviewing our paper. Your comments have been invaluable in helping us refine our work. Below, we would like to quickly address the AC's final notes:
>
> ---
> **Motivation for the Riemannian manifold framework:** We respond that its simply not possible to obtain a manifold that satisfies the properties needed for Stackelberg equilbrium learning via a simple Euclidean map, as this does not guarantee the properties over the manifold as enforced by the criteria from Table 1 (i.e., Lipschitzness, sensitivity, and even spreading). We discussed this briefly with reviewer D4pN in the rebuttal, however perhaps it was not made super clear in the paper. Based on this review, we see the need to revise the paper to provide a more explicit explanation of this item.
>
> **Efficiency concerns:** The Riemannian framework trades the complexity of the original bi-level optimization problem for the complexity of normalizing flows, resulting in a simpler optimization problem on the Stackelberg manifold. This trade-off is justified by the framework’s ability to greatly simplify the multi-agent learning process. We see that there is need to clarify this in an update to our paper.
>
> **Linearity assumption:** Perhaps this was misunderstood, but the linearity assumption is **not a limitation** but the **intention** of our framework. We assume a linear relationship between the feature space and the reward space, valid assumption previously established in online learning literature [2, 3]. We then impose the smoothness and biijectivity of the feature map, via representation learning with normalizing flows. Although this may to solve all problems in multi-agent learning, we firmly believe this subsumes a vast domain of Stackelberg online learning problems, as is the focus of our work and also complimented by our empirical results.
>
> Furthermore, the bijectivity and smoothness of the feature map, $\tilde{\phi}$, ensures that this linearity holds for any alternative feature map $\phi$. We provide theoretical proof for this through a change-of-variables argument in the rebuttal. However, we admit this was brief, and we agree that there is more of a need to clarifying this in our paper.
>
> ---
> We appreciate the AC and reviewers for highlighting the need several areas of clarification, and we intend to make an update to address this accordingly. We remain grateful for the AC and reviewers’ insightful comments and which serve only to improve our work in the future. Thank you for your time and thoughtful engagement.
>
> Kind regards,
> Authors

---

### Decision · Program_Chairs · 2025-01-22

Reject